# Mechanical forces across compartments coordinate cell shape and fate transitions to generate tissue architecture

Clémentine Villeneuve[1,2], Ali Hashmi[1,9], Irene Ylivinkka [1,9], Elizabeth Lawson-Keister[3,9], Yekaterina A. Miroshnikova[1], Carlos Pérez-González[4], Satu-Marja Myllymäki[1,5], Fabien Bertillot[1,2], Bhagwan Yadav [1], Tao Zhang [6], Danijela Matic Vignjevic [4], Marja L. Mikkola[5], M. Lisa Manning [3] ✉ & Sara A. Wickström [1,2,7,8] ✉

Morphogenesis and cell state transitions must be coordinated in time and space to produce a functional tissue. An excellent paradigm to understand the coupling of these processes is mammalian hair follicle development, which is initiated by the formation of an epithelial invagination—termed placode—that coincides with the emergence of a designated hair follicle stem cell population. The mechanisms directing the deformation of the epithelium, cell state transitions and physical compartmentalization of the placode are unknown. Here we identify a key role for coordinated mechanical forces stemming from contractile, proliferative and proteolytic activities across the epithelial and mesenchymal compartments in generating the placode structure. A ring of fibroblast cells gradually wraps around the placode cells to generate centripetal contractile forces, which, in collaboration with polarized epithelial myosin activity, promote elongation and local tissue thickening. These mechanical stresses further enhance compartmentalization of Sox9 expression to promote stem cell positioning. Subsequently, proteolytic remodelling locally softens the basement membrane to facilitate a release of pressure on the placode, enabling localized cell divisions, tissue fluidification and epithelial invagination into the underlying mesenchyme. Together, our experiments and modelling identify dynamic cell shape transformations and tissue-scale mechanical cooperation as key factors for orchestrating organ formation.

The structure of tissues is tightly linked to their function. During the formation of functional organs, large-scale changes in tissue elongation, stretching, compression, folding/buckling and budding impact the shape, position, packing and contractility state of cells. Conversely, changes in single-cell contractility, shape and position locally alter tissue organization and mechanics. Importantly, these tissue and cell-scale transformations need to be tightly coordinated in space and time with the acquisition of specialized cell states through mechanisms that are poorly understood[1,2]. The formation of the mammalian placode, a lens-shaped multilayered epithelial thickening that gives rise to the hair follicle, serves as an ideal model to study how cell state transitions and morphogenesis are coordinated to generate a specialized tissue structure.

In the mouse embryo, a first wave of the periodically spaced pattern of the hair follicle placode arises at late embryonic day 13 (E13) from

**Fig. 1 | Placode cells undergo isotropic elongation and subsequent basal surface expansion. a**, Schematic representation of the skin and the morphological parameters quantified. **b**, Phalloidin and Edar-stained sagittal cross-sections of mouse epidermis at E13.5–15-5. Scale bars, 20 μm. Images are representative of three mice per group. **c**, Quantification of cell axis lengths from E13.5 placode (marked by *Fgf20* expression) and epidermis from the images in **b**. No substantial differences in cell morphology between placode and epidermis are observed (*n* = 57 cells pooled across 11 placodes/surrounding epidermis from three mice; two-way analysis of variance (ANOVA)/Tukey's). **d**, Quantification of cell axis lengths from E13.5–15.5 placode basal cells. Note the increased height and reduced apical and basal cell surface lengths in E14.5 and E15.5 placodes,

and the increased basal surface length at E15.5 (*n* = 57 (E13.5), 74 (E14.5) and 77 (E15.5) cells pooled across three mice per timepoint; two-way ANOVA/Tukey's). **e**, Quantification of the basal/apical cell surface ratio in E13.5–15.5 placodes. Note the increased basal surface ratio at E15.5 (*n* = 11 (E13.5), 9 (E14.5) or 8 (E15.5) placodes pooled across three mice; Kruskal–Wallis/Dunn's). **f,g**, Quantification and representative images of 3D rendering of cell volumes and shapes from E14.5 placode and epidermis. Note the volume-preserving elongation of the placode cells. Scale bars, 5 μm (*n* = 392 (Epidermis) and 387 (Placode) cells pooled across three mice; *P* = 0.7 Mann–Whitney). Minimum-to-maximum box plots show 75th, 50th and 25th percentiles. NS, not significant.

a homogeneous epithelial sheet adhering to a basement membrane that overlays the dermis, a connective tissue containing dispersed mesenchymal cells. This is followed by a second and third wave of morphogenesis around E15.5 and E18, respectively, producing hair-follicle placodes that are morphologically identical and regularly interspersed with the first-wave follicle[3,4]. One of the key pathways regulating placode specification is Wnt signalling, which acts in a bidirectional, paracrine manner between the epidermis and the underlying dermis. This process commences at E12.5, when secreted epidermal Wnt activates broad mesenchymal β-catenin signalling to initiate the specification of the fibroblast population that later becomes the dermal condensate, an essential structure for hair follicle formation[5]. These fibroblasts, through an unknown signal, then induce hair follicle progenitor fate in the epidermis in patterned pre-placodes molecularly identified by

*Dkk4*, *Edar* and *Fgf20* expression[6–9]. Subsequently, progenitor cell rearrangement and compaction demarcate the physically identifiable placode (Fig. 1a). At the same time, mesenchymal cells migrate and cluster to form a dermal condensate directly underneath the placode[10–12]. Once this spatial pattern has been defined, progenitor cells specify to become follicle cells and activate the expression of additional genes to drive their development into the mature hair follicle, which harbours a set of adult stem cells responsible for constant self-renewal of the hair follicle throughout life. One of these genes is Sox9, the master transcription factor required for adult hair follicle stem cells, which initially exhibits broader expression but becomes restricted to the placode between E14 and E15, defining the transcriptional profile of future hair follicle stem cells[13,14]. The placodes—organized into concentric rings of stem and progenitor cell populations—subsequently grow

out as longitudinally aligned cylindrical compartments to form the hair follicle[13]. However, the physical mechanisms driving the establishment of a Sox9-positive stem cell population, coordinated with stereotypical morphological changes leading to placode invagination are still poorly understood.

In this Article we use the murine hair follicle placode as a paradigm to construct and genetically validate a biomechanical model for epithelial sheet transformation into a placode bud. Using whole-embryo live imaging, mechanical measurements, genetic manipulations and three-dimensional (3D) vertex modelling, we show that a combination of epithelial cell-intrinsic actomyosin contractility and extrinsic mechanical stresses from the underlying mesenchyme are required for placode thickening. Importantly, mechanical stress also contributes to enhancement and spatial restriction of Sox9 expression, coupling morphogenetic dynamics to cell type specification and tissue compartmentalization. Subsequent remodelling of the basement membrane facilitates a release of accumulated pressure within the placodes to induce the localized cell divisions necessary for placode budding. Collectively, this study identifies a critical role for coordinated tissue- and cell-scale forces across cell compartments in mammalian organogenesis.

## Results

### Placode cells undergo elongation and basal surface expansion

Hair follicle placode formation is initiated by epidermal cell rearrangement and compaction to form a physically identifiable thickened structure[10,15]. To investigate morphological changes that generate this specific epithelial deformation, we performed whole mount imaging of mouse epidermis and quantified the cell shapes of the basal layer by segmenting confocal z-stacks starting at E13.5. We first measured the basal cell size in three dimensions (Fig. 1a) using antibodies against Edar as well as *Fgf20-β-galactosidase* knock-in allele embryos (*Fgf20*[βGal]), which report the expression of *Fgf20*; both of these are among the earliest known marker genes of placode cells[7]. Analysis of morphological features revealed that at E13.5, Edar[hi] and *Fgf20*[βGal]-positive cells were still cuboidal in shape and morphologically indistinguishable from other basal cells (Fig. 1b,c and Extended Data Fig. 1a–c). We also compared cell elongation in the x–y plane, the nematic order of elongation, as well as the numbers of neighbouring cells within *Fgf20*[βGal]-positive cells and the surrounding epidermis, but these tissue-scale measurements also revealed no specific morphological differences between *Fgf20*[βGal]-positive cell clusters and their neighbours (Extended Data Fig. 1a–c), indicating that placode cell fate specification precedes any substantial cell shape or tissue mechanical changes.

In stark contrast, the next embryonic day (E14.5) revealed a clearly distinguishable placode characterized by a decrease in the cross-sectional area and a corresponding elongation of the longitudinal (lateral) axis of the basal cells (Fig. 1b,d). Interestingly, both apical and basal surfaces of the basal epidermal cells showed a proportional

decrease from their length at E13.5, indicating that the elongation was not achieved through apical or basal constriction (Fig. 1b–e and Extended Data Fig. 1a,b). Furthermore, 3D segmentation revealed that the elongation initiated between E13.5 and E14.5 occurred without an alteration in cell volume. This analysis also confirmed that the placode cells displayed cylindrical shapes with comparable apical and basal surface areas (Fig. 1f,g). At E15.5, the placode cells continued to further elongate (Fig. 1b,d). In addition, while the apical surface length remained comparable to that of E14.5 cells, the basal surface expanded (Fig. 1d,e), resulting in a conical shape of E15.5 placode cells. Collectively, these data indicate that the initial placode cell fate determination precedes morphological changes. The subsequent morphological changes are initiated by a volume-preserving elongation of basal cells along the longitudinal axis at E14.5, followed by expansion of the basal surface of the basal cells at E15.5.

### Cycles of deformation drive tissue elongation

To understand the mechanics of the two-step morphological transformation, we investigated cell- and tissue-scale forces that could act on the placode cells. For this we performed live imaging of intact mouse embryos with genetically labelled plasma membrane-targeted Tomato (R26R[mT/mG]; ref. 16). Visual inspection and particle image velocimetry (PIV) of live tissue dynamics to quantify cell movements indicated that cells within the E14.5 placodes displayed coordinated, collective centripetal fluctuations, as well as rotational tissue flows around the placode, as has been observed previously in skin explants[15] (Fig. 2a,b and Supplementary Video 1). Interestingly, the flows were no longer detected at E15.5, whereas the fluctuations continued and displayed overall negative divergence in the plane of the basal layer, indicative of tissue flows out of the basal cell layer plane (Fig. 2a,b and Supplementary Video 2). Indeed, examining tissue flows in 3D revealed in-plane contractions positioned at the placode neck, extensile tissue flows downwards to the dermis, and deformation of junctions, resulting in plastic deformation of the placode structure and net downward tissue elongation (Fig. 2c, Supplementary Video 3 and Extended Data Fig. 2a,b). To understand the forces that could generate these patterns of tissue deformation, we measured mechanical stresses around and within the E14.5 placode, as well as within the epidermis, using laser ablation[17]. Quantification of recoil and its radial displacement at the placode–epidermis boundary showed that recoil away from the cut occurred on both sides, confirming that this boundary was under tensile strain. Similar recoil magnitudes were detected within the epidermis (Extended Data Fig. 2c–e and Supplementary Video 4). In contrast, tension inside the placode itself was low (Fig. 2d and Supplementary Video 5).

To identify the source(s) of tension on the placode, we examined the patterns of actomyosin contractility. Phosphorylated myosin light chain-2 (pMLC2), which binds to the myosin-II heavy chain and generates contractile force, was higher along the apical domain than at the

**Fig. 2 | In-plane oscillatory deformation drives tissue elongation. a**, Live imaging snapshots of basal epidermis (left; dashed circles indicate placodes) analysed by PIV (middle) to extract flow vector mean divergence (right). Note the localized tissue flows at E14.5 and E15.5 and negative divergence at E15.5 (*n* = 4 mice per stage). **b**, Quantification of mean divergence over time (*n* = 3 (E14.5) and 4 (E15.5) mice per stage; mean ± s.e.m.). **c**, Snapshots (left), corresponding PIV (middle) and strain rates (right) of optical cross-sections from E15.5 3D time-lapse images. Note the progressive constriction of the placode neck and downward flow of tissue. **d**, Live imaging snapshots with regions of laser ablation highlighted (circles) and vectors showing the recoil magnitude and direction after ablation (arrows). The quantification of mean displacement over time shows larger displacements in the epidermis, indicative of lower tension inside the placode (*n* = 6 mice; Wilcoxon test). **e**, pMLC2, DAPI and Keratin-10-stained skin cross-sections at E14.5. The dashed line marks the boundary between epidermis and dermis. **f,g**, Quantifications of pMLC2 intensity (**f**) and the apical/

basal pMLC2 intensity ratio (**g**) at E14.5. Note the low pMLC2 within the placode and high pMLC2 on the apical/suprabasal surfaces (*n* = 12 placodes from three mice; Student's test). GV, gray values. **h**, Top-view dermis whole mount at E14.5 (top) and E15.5 (bottom) and quantifications show a ring of vimentin-positive fibroblasts (arrows) surrounding the placode (*n* = 12 (E14.5) and 10 (E15.5) placodes pooled across three mice; Student's *t*-test). **i**, Representative live imaging snapshots showing the bottom of the placode and the surrounding ring of fibroblasts (top). The region of laser ablation is indicated (rectangle), and the vectors (arrows) show the recoil magnitude and direction after ablation. Quantification of the mean displacement direction over time (bottom) shows fibroblast recoil away from the cut, while placode cells displace towards the cut (*n* = 92 (fibroblasts) or 45 (placode) from 45 positions pooled across six mice). Minimum-to-maximum box plots show 75th, 50th and 25th percentiles. Scale bars, 50 μm (**a**) or 20 μm (other panels). All images are representative of three mice per group.

basal surface in both placodes and epidermal cells at E14.5, and also particularly enriched in the suprabasal cells (Fig. 2e–g). Consistent with the laser ablation studies, pMLC2 was overall lower within the basal cells of the placode than in the surrounding basal cells of the epidermis

(Fig. 2e–g). At E15.5, pMLC2 remained low within the placode and high within the suprabasal layers. In contrast, high myosin activity was detected within the dermis, particularly within a ring-like structure around the placode (Extended Data Fig. 2f,g). Co-staining with vimentin

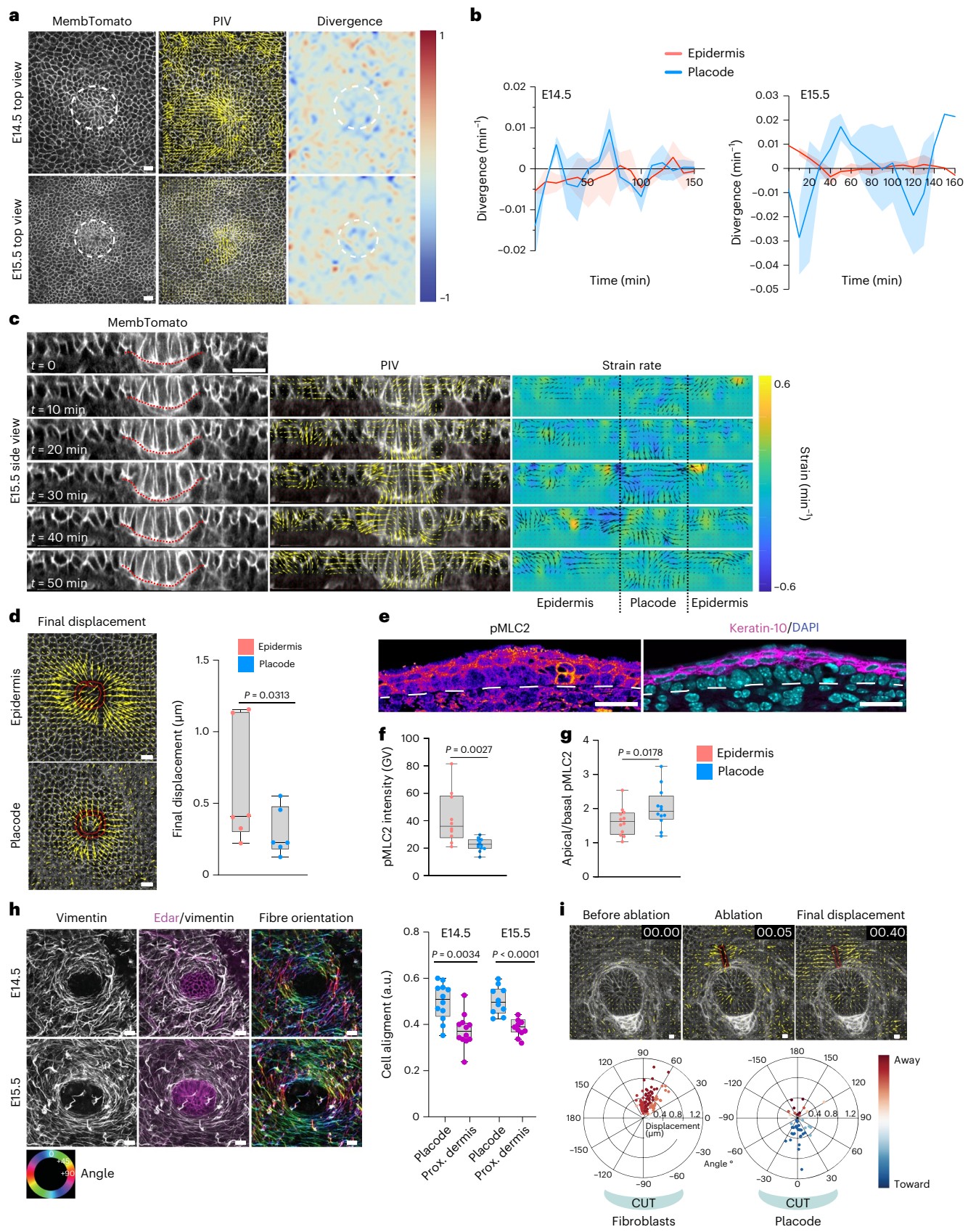

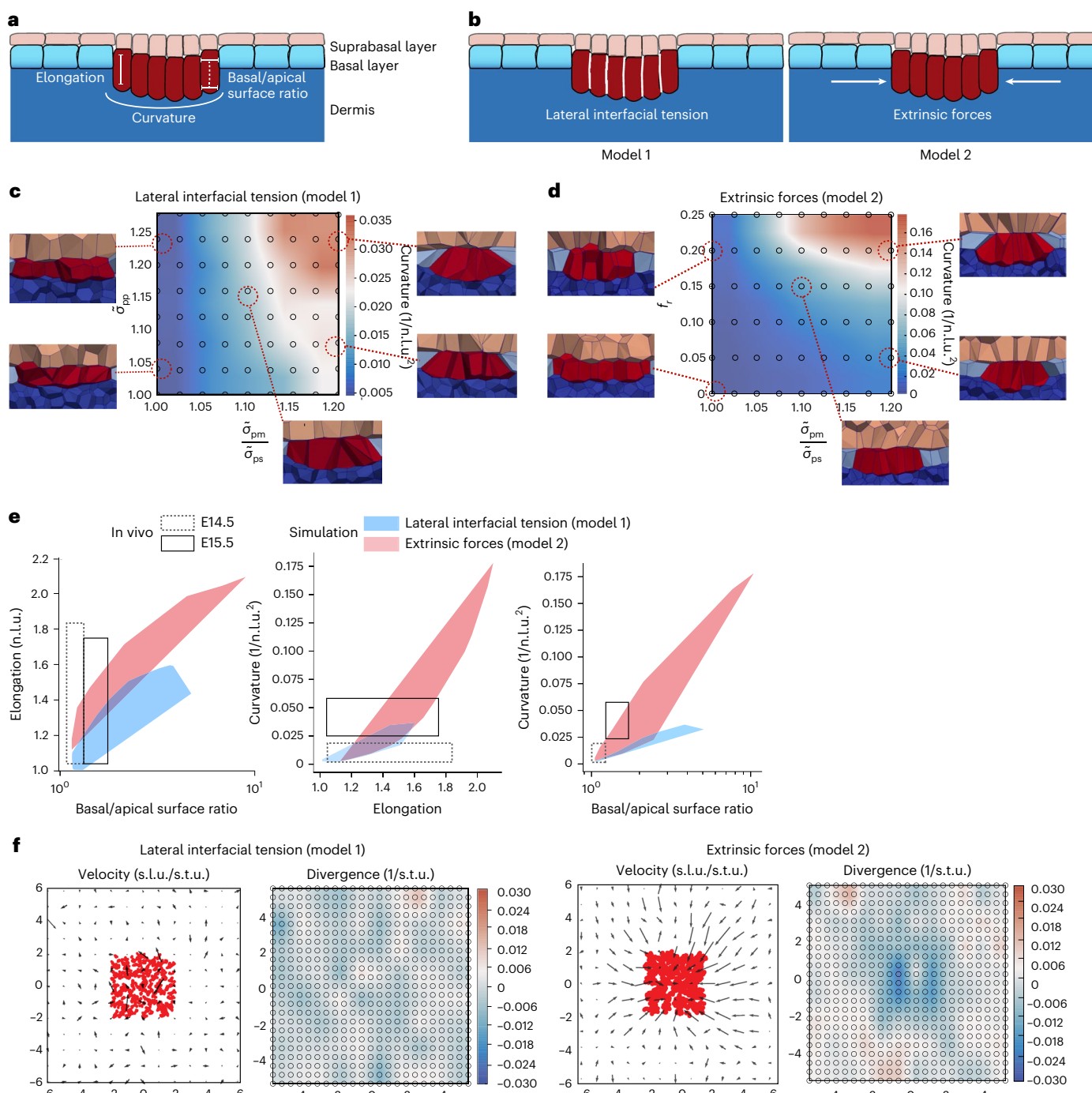

**Fig. 3 | Modelling suggests cooperative forces in cell shape transformations.** **a**, Schematics of the placode and parameters measured from embryos and simulations. **b**, Schematic of the two models where either the lateral wetting coefficient ($\bar{\sigma}_{pp}$; model 1) or the magnitude of extrinsic forces ($f_r$; model 2) was varied in combination with a changing basal-to-apical wetting coefficient $\left(\frac{\bar{\sigma}_{pm}}{\bar{\sigma}_{ps}}\right)$. **c**,**d**, Phase diagrams of curvatures induced by interaction of the lateral wetting coefficient (**c**) or extrinsic lateral forces (**d**) with varying changing basal-to-apical wetting coefficient. Note the generation of tissue curvature resembling the placode architecture with high lateral forces and a high basal-to-apical wetting coefficient. **e**, Comparison of morphological measurements from E14.5 and E15.5 embryos to simulations shows that model 2, with lateral cell non-autonomous forces, recapitulates in vivo morphological transformations across both developmental stages, whereas model 1 recapitulates cell shape changes at E14.5. **f**, In silico PIV analysis of tissue flow velocities and mean divergence of the flows from simulations with differential lateral wetting coefficient (model 1) or extrinsic lateral forces (model 2). n.l.u., normalized length unit; s.t.u., simulated time unit; s.l.u., simulated length unit.

and Edar confirmed this structure as a dense ring of fibroblasts aligned tangentially around the placode, which was detectable at E14.5 and became even more prominent at E15.5 (Fig. 2h). In contrast to an ablation in an adjacent region, specific laser ablation of the fibroblast ring revealed directional displacement of fibroblasts away from the cut, indicating that this structure was under tension (Fig. 2i, Extended Data Fig. 2h,i and Supplementary Video 6). In contrast, the placode cells showed less displacement, and this occurred toward the ablation, indicating that placode cells were being confined by the contractile fibroblasts (Fig. 2i).

Collectively these data confirmed that initial placode morphogenesis occurred in two stages: thickening/elongation at E14.5 was characterized by centripetal contractile tissue fluctuations, while at E15.5 the tissue underwent a transition towards downward tissue flows, constrained by in-plane contractions. Consistently, although contractility was low within the placode, a ring of contractile fibroblast developed to tightly wrap around the placode base, confining the placode cells.

## Modelling suggests cooperative forces in shape changes

To understand how the observed patterns of forces could generate the placode shape, we turned to quantitative mechanical modelling. We developed a multilayered 3D vertex model that consisted of basal epidermal and placode cells adhering to a basement membrane, as well as multiple layers of suprabasal cells (details are provided in the Methods, and Supplementary Table 1 presents the model parameters). We used the model to probe how three experimentally observed morphological features—cell elongation, basal and apical placode cell surface area, and deformation/curvature of the epithelium (Fig. 3a)—are generated by the observed patterns of epidermal myosin activity, tissue tension and fibroblast contractility. We explored two hypotheses in parallel: model 1, where cell-intrinsic changes in lateral interfacial tension in placode cells drive the morphological changes, and model 2, where extrinsic forces, such as those generated by the fibroblast ring around the placode, drive the changes (Fig. 3b). In both cases we assumed that the observed asymmetry in myosin activity generates asymmetry in the tensions/wetting coefficients for the apical and basal surfaces of the placode cells, which we describe in terms of a wetting coefficient at interfaces between the placode and suprabasal cells ($\tilde{\sigma}_{ps}$), and between placode cells and the basement membrane ($\tilde{\sigma}_{pm}$) (see Methods for more details). For model 1, we additionally altered the wetting coefficient $\tilde{\sigma}_{pp}$ describing tensions at lateral interfaces between the placode cells, while for model 2, we simulated external contractile forces by imposing forces directed towards the centre of the placode on all vertices of cells peripheral to the placode. The maximum force ($f_r$) occurs at the edge of the placode, and forces decay exponentially towards zero over a length scale of several cell diameters away from the placode edge. To explore the effect of a wide range of parameters on the morphology of the placode, we generated phase diagrams (Fig. 3c,d and Extended Data Fig. 3a,b) by varying the basal-to-apical wetting coefficient ($\frac{\tilde{\sigma}_{pm}}{\tilde{\sigma}_{ps}}$), lateral wetting coefficient $\tilde{\sigma}_{pp}$ and the magnitude of extrinsic forces ($f_r$) over the entire physiological range (beyond this range, the placode does not maintain integrity or produces abnormal curvatures).

We then measured cell elongation, basal-to-apical surface ratio and the curvature of the simulated placodes and compared them directly to the quantifications performed in embryos (Fig. 3c–e and

Extended Data Fig. 3a,b). The simulations showed that both wetting coefficients and extrinsic force magnitudes had a noticeable impact on the extent of placode cell elongation, basal surface expansion and placode curvature. Notably, direct comparisons with the experimental data indicated that cell-autonomous differences in the lateral wetting coefficient were sufficient to recapitulate the morphological transformations observed at E14.5, but were insufficient to fully explain the subsequent morphological changes at E15.5, as indicated by the degree of overlap between measurements from the embryo and from the simulations (Fig. 3e). In contrast, cooperation of extrinsic forces in combination with a high basal-to-apical wetting coefficient recapitulated the changes observed at both embryonic stages (Fig. 3e). Most importantly, only the extrinsic force model 2 could robustly recapitulate the pattern of negative divergence of tissue flows observed in E15.5 embryos (Fig. 3f and Extended Data Fig. 3c). Collectively, the simulations predicted that cell-autonomous changes in surface tension generated by a polarized myosin distribution could be sufficient to generate some degree of cell elongation and curvature, but were not able to explain the morphological transformation at E15.5, where cell extrinsic forces might be playing a more important role.

## Placode formation requires epidermal and dermal contractility

To challenge the model that predicted the collaboration of cell-autonomous and extrinsic contractile forces in placode formation, we proceeded to examine the role of myosin contractility in the epidermal and dermal compartments. First, we deleted myosin-IIA, the major myosin isoform responsible for keratinocyte actomyosin contractility in the epidermis (Keratin-14 Cre deletion of the *Myh9* gene; *Myh9*-eKO (ref.[18])). At E14.5, although placode development was initiated, the placodes of *Myh9*-eKO mice were less invaginated than in control mice (Fig. 4a,b). Also, the number of Sox9-positive cells and their local density were reduced within the *Myh9*-eKO placodes (Extended Data Fig. 4a–d). Interestingly, however, the cells still showed elongation comparable to cells in control mouse placodes (Fig. 4a,c). At E15.5, the placode showed a more pronounced phenotype as it failed to further invaginate. In addition, the dermal condensate, as marked by Sox2 expression[19,20], was found partially embedded within the placode cells upon loss of epidermal contractility (Extended Data Fig. 4e,f). These data suggested that epidermal myosin-IIA activity was required for placode invagination and to generate forces against the dermal condensate, but it was not sufficient to induce full placode cell elongation. To challenge this notion and to further challenge the vertex model, we queried the model on the importance of the high apical myosin activity observed in the epidermis by reducing the apical interfacial tension parameter, that is, the wetting coefficient at interfaces between placode and suprabasal cells ($\tilde{\sigma}_{ps}$). The simulations predicted that this would lead to evagination of the placode (Extended Data Fig. 4g). Indeed, analyses of the second morphogenetic wave of placode

**Fig. 4 | Placode development requires epidermal and dermal contractility.**
**a**, Phalloidin- and Edar-stained cross-sections of control (CNL) and *Myh9*-eKO epidermis at E14.5. Scale bars, 20 µm. **b,c**, Quantifications of placode invagination depth (**b**) and placode cell surface lengths (**c**) from CNL and *Myh9*-eKO mice at E14.5 show reduced placode depth but unchanged cell morphology in *Myh9*-eKO mice (*n* = 12 (CNL) and 13 (*Myh9*-eKO) placodes from three mice, Student's *t*-test (left) or 55 (CNL) and 53 (*Myh9*-eKO) cells pooled across three mice (right), two-way ANOVA/Tukey's). **d**, Edar-, vimentin-, phalloidin- and DAPI-stained whole mount top views from CNL and *Myh9*-dKO mice at E16.5. Note there are fewer placodes in *Myh9*-dKO. Scale bars, 50 µm. **e**, Quantification of vimentin-positive fibroblast ring thickness (left) and placode numbers (right) from **d**. Note the reduced ring width in *Myh9*-dKO. Scale bars, 50 µm (*n* = 70 (CNL) and 55 (*Myh9*-dKO) placodes from three mice per group (left) and *n* = 4 mice per group (right), Mann–Whitney). **f,g**, Sox9- and phalloidin-stained images (**f**) and quantification (**g**) of skin whole mounts from CNL and *Myh9*-dKO mouse epidermis at E14.5. A reduced local density of Sox9+ placode cells is

observed in *Myh9*-dKO. Scale bars, 20 µm (*n* = 5 mice per group; Mann–Whitney). **h,i**, Phalloidin- and Sox9-stained images (**h**) and quantifications (**i**) of blebbistatin-treated skin explants (10 nM for 24 h) starting at E13.5. Placodes were defined by 3D morphology and Sox9 positivity. Scale bars, 50 µm (*n* = 17 placodes from four mice; Mann–Whitney). **j,k**, Phalloidin- and Sox9-stained images (**j**) and quantification (**k**) of explants treated with blebbistatin for 24 h starting at E13.5. Circles mark placodes. Note the reduced placode/epidermis Sox9 intensity ratio in blebbistatin-treated explants. Scale bars, 20 µm (*n* = 11 (DMSO) and 13 (blebbistatin) placodes from three mice; Mann–Whitney). **l,m**, Schematic of the microfabrication design (**l**, top), representative images of a top view (**l**, middle) and a cross-section (**l**, bottom), and quantification (**m**) of epidermal progenitor cells cultured on microfabricated substrates and immunostained for Sox9 and DAPI. Note the increased Sox9 at the indentation boundary. Scale bars, 20 µm (*n* = 4 independent experiments; one-sample *t*-test). Minimum-to-maximum box plots show the 75th, 50th and 25th percentiles. All images are representative of three mice per group.

formation at E16.5 revealed inverted curvatures of placodes in *Myh9*-eKO mice (Extended Data Fig. 4h,i), further strengthening the conclusion that apical/suprabasal myosin activity in the epidermis is required to produce contractile forces against the dermis.

As the simulations predicted a critical role for extrinsic forces in placode formation, we proceeded to examine the role of the contractile fibroblasts around the developing placode. To this end, we deleted *Myh9* specifically in fibroblasts using Twist2-promoter driven Cre[21]

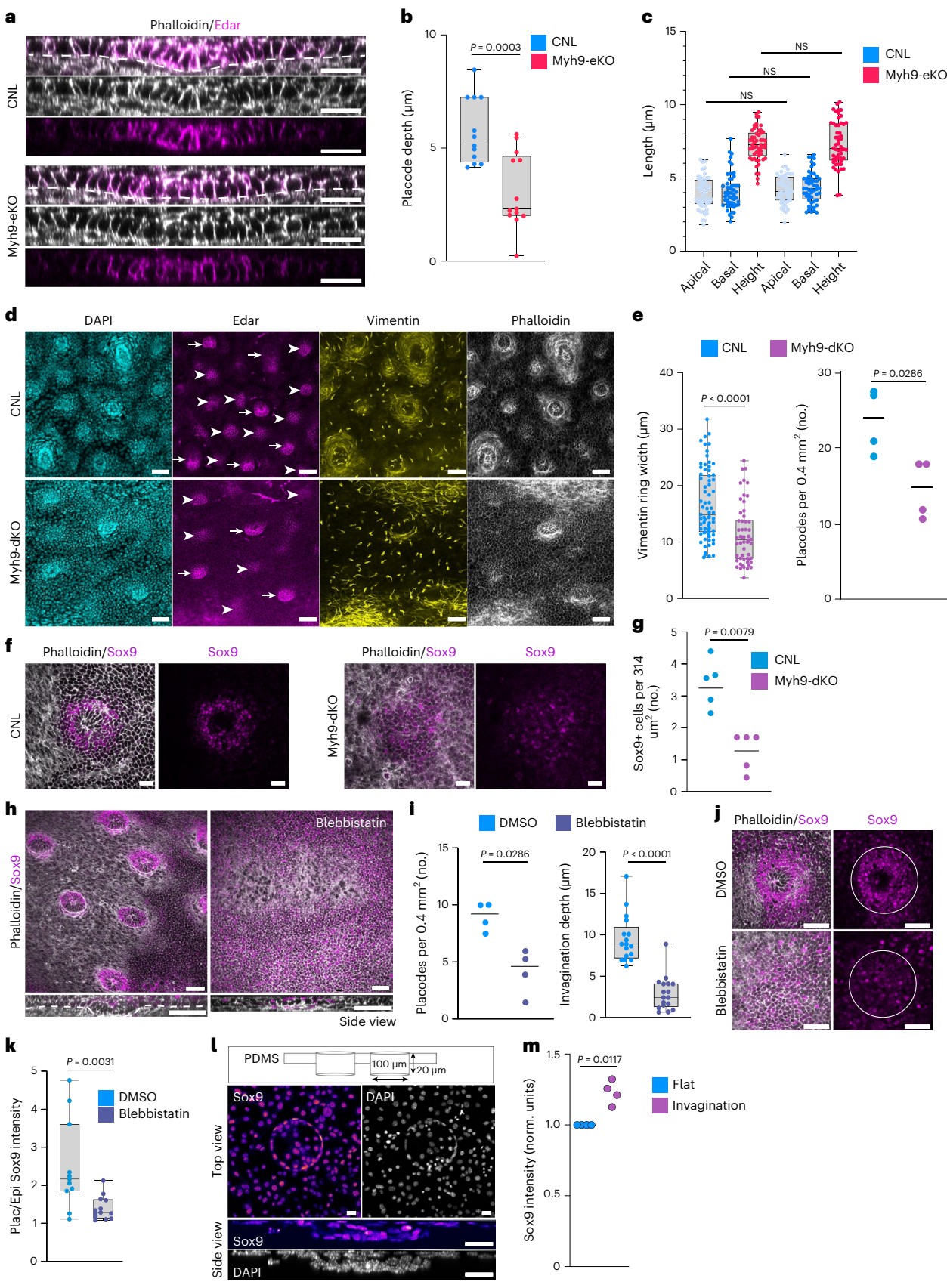

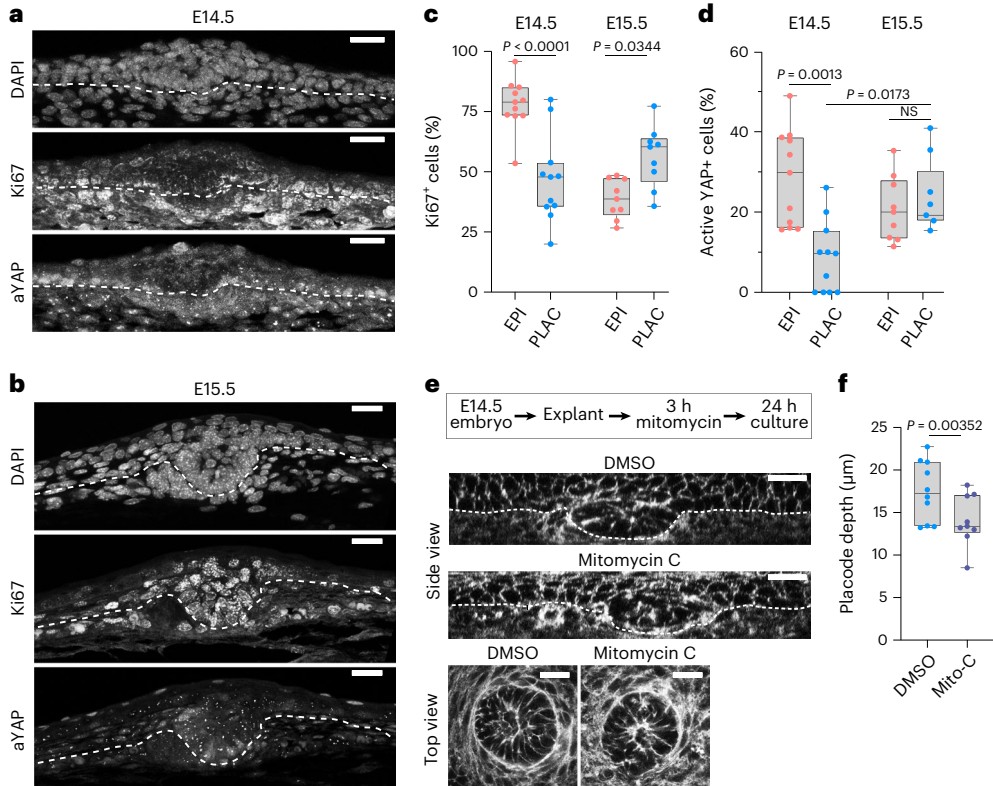

**Fig. 5 | Spatiotemporal coordination of cell divisions controls placode budding. a,b**, DAPI, Ki67 and active YAP (aYAP)-stained sagittal cross-sections of E14.5 (**a**) and E15.5 (**b**) epidermis. Scale bars, 20 µm. Images are representative of three mice per group. **c,d**, Quantification of Ki67 (**c**) and aYAP (**d**) from E14.5 and E15.5 placodes and epidermis. Note the low levels of Ki67 and aYAP in E14.5 placode and the return to levels comparable with epidermis in E15.5 placodes (n = 11 (E14.5) and 9 (E15.5) placodes or surrounding epidermis from three mice per group; one-way ANOVA (**c**) and Kruskal–Wallis (**d**)). **e**, Experimental outline and optical cross-sections and top views of E14.5 phalloidin-stained mitomycin C-treated and DMSO control skin explant cultures. Scale bars, 20 µm. Images are representative of three explants per group. **f**, Quantification of placode depth from the experiments shown in **e** (n = 10 (CNL) and 9 (Mito-C) placodes from three explants; Student's *t*-tests). Note the reduced placode depth in mitomycin-treated skin explants.

(*Myh9*-dKO; targeting dermal fibroblasts lining the epidermis and placodes high in Twist2 expression; Extended Data Fig. 5a). Analyses of the fibroblast ring showed reduced thickness in *Myh9*-dKO mice dermis (Fig. 4d,e and Extended Data Fig. 5b), confirming the hypothesis that myosin-IIA activity is required to assemble the ring of fibroblasts surrounding the placode. Importantly, much fewer placodes, as defined by Edar expression and the presence of a Sox2-positive dermal condensate, were observed in the *Myh9*-dKO epidermis at E14.5 (Extended Data Fig. 5c,d). Where placodes had formed, they displayed slightly reduced invagination, and cells within these *Myh9*-dKO placodes were less vertically elongated (Extended Data Fig. 5e,f). This severe defect in placode morphogenesis became even more obvious when the second wave of placode morphogenesis initiated at E16.5, quantified by reduced placode number and increased distance between individual placodes (Fig. 4d,e and Extended Data Fig. 5g). This indicated that myosin-IIA activity in the dermal fibroblasts and extrinsic mechanical forces are essential to promote placode formation, as also predicted by the simulations. Surprisingly, whereas in control mice Sox9-expressing cells were compartmentalized at the placode–epidermis interface, forming a ring-like structure, *Myh9*-dKO placodes showed an overall more dispersed Sox9 localization within the epidermis (Fig. 4f). Quantification revealed significant reductions in the number and density of Sox9-expressing cells, in particular around the prospective placode boundary defined by the presence of the dermal condensate and epithelial thickening (Fig. 4f,g and Extended Data Fig. 5h). Additionally, and in contrast to the *Myh9*-eKO, a significant increase of the average distance of Sox9-positive cells from the centre of the placode and in

the total area of tissue containing Sox9-positive cells was observed in the *Myh9*-dKO placodes, indicative of a failure of Sox9 compartmentalization (Extended Data Fig. 5h,i).

To investigate the potential collaborative role of myosin contractility across the compartments, we prepared full skin explants from E13.5 and E14.5 mice and treated them with blebbistatin for 24 h to block myosin activity in all cells. Strikingly, when myosin was inhibited in all cells at E13.5, the placodes, defined by cell shape changes associated with Sox9 expression and the presence of a Sox2-positive dermal condensate, almost completely failed to develop (Fig. 4h,i and Extended Data Fig. 5j), indicating a critical role of collective myosin activity in both compartments in placode thickening as also predicted by the model. Importantly, no substantial effects on cell proliferation were observed with the blebbistatin concentration used (Extended Data Fig. 5k). In addition to the morphogenetic defects, Sox9 expression failed to become compartmentalized into the placode cells at the boundary (Fig. 4j,k). Treating E14.5 embryos instead resulted in an impairment of placode invagination into the dermis, with the dermal condensate partially indenting the epithelial structure (Extended Data Fig. 5l,m), indicating that myosin-II activity is further required for placode invagination and curvature. The above genetic experiments implicate a major role for myosin-IIA in this process, but additional or compensatory effects from myosin-IIB are also possible[22].

As reducing the extrinsic forces on the placode by myosin-IIA inhibition resulted not only in reduced placode formation but also in reduced Sox9 levels and compartmentalization, we hypothesized that mechanical confinement of the placode cells by the contractile fibroblast ring

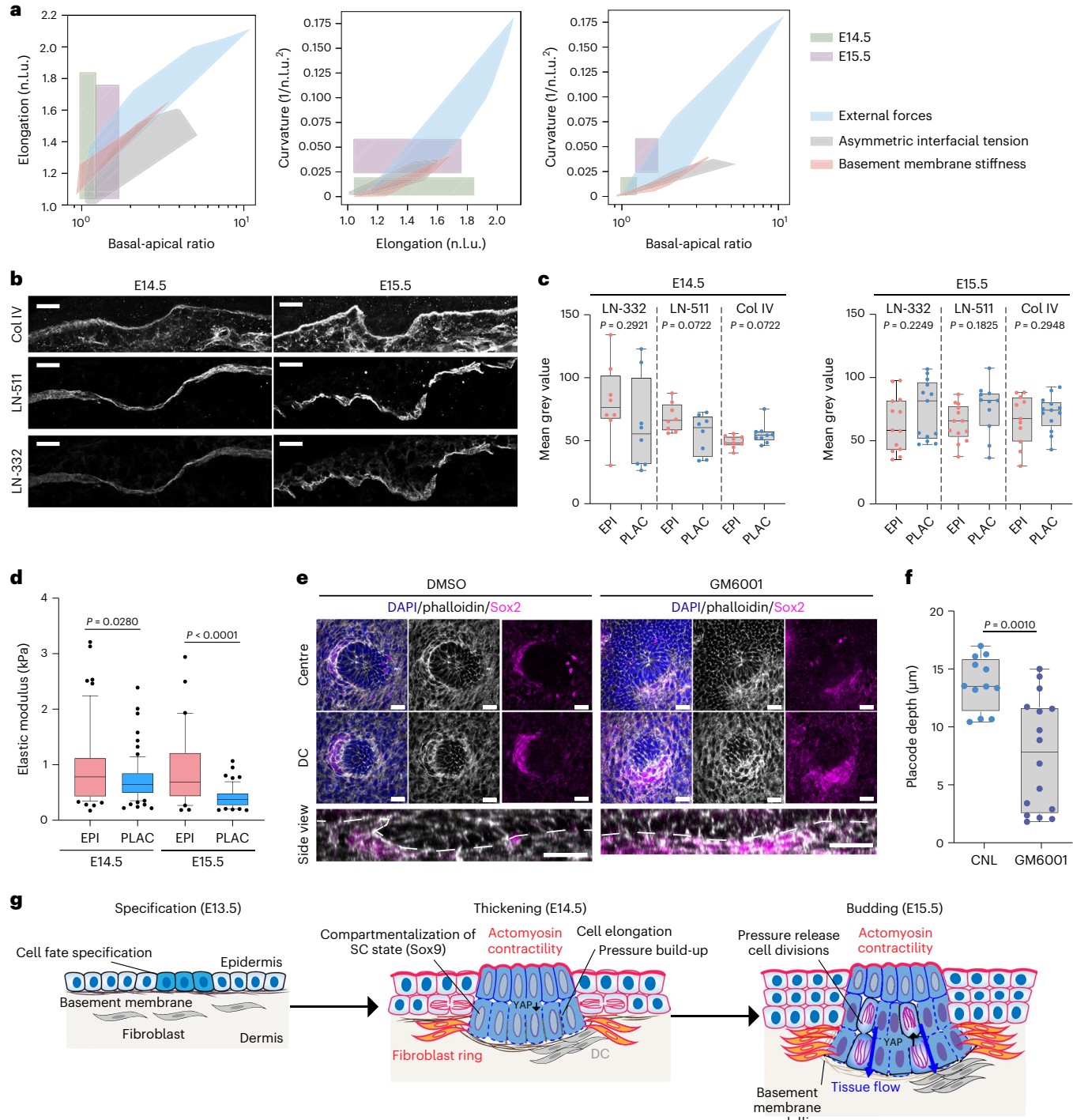

**Fig. 6 | Basement remodelling is required for placode downgrowth.**
**a**, Comparison of morphological measurements from E14.5 and E15.5 embryos to vertex model simulations shows that decreasing basement membrane stiffness below the placode is sufficient to induce placode cell elongation, invagination and curvature to that observed at E14.5. n.l.u., normalized length unit. **b**, Collagen IV (Col IV), laminin (LN)-511 and LN-332-stained cross-sections of E14.5 and E15.5 epidermis. Scale bars, 10 μm. Images are representative of three mice per group. **c**, Quantification of images from **b** show no substantial differences in Col IV or LN-332/511 intensity between epidermis (EPI) and placode (PLAC) (*n* = 13 (laminins) and 10 (Col IV) tissue sections pooled across three mice per group; Student's *t*-test). **d**, Quantification of elastic moduli of basement membranes using force indentation spectroscopy. Note the reduced elastic moduli of basement membranes below placodes (*n* = 55 (E14; EPI), 75 (E14; PLAC), 32 (E15; EPI) and 54 (E15; PLAC) force curves pooled across three mice per group; Kolmogorov–Smirnov).

**e**, Top views (top panels) and optical cross-sections (bottom panels) of DAPI, Sox2 and phalloidin-stained skin explant cultures isolated from E14.5 mice and treated with GM6001 (10 μM) or DMSO control for 48 h. Scale bars, 20 μm. Images are representative of three explants per group. DC, dermal condensate. **f**, Quantification of images shown in **e** show reduced placode depth in GM6001-treated explants (*n* = 12 (CNL) and 16 (GM6001) placodes from three explants per condition; Mann–Whitney). **g**, Model of placode formation, indicating early cell fate specification (E13.5) followed by polarized myosin activity and assembly of a peri-placode contractile fibroblast ring to generate tissue elongation and basal cell surface expansion (E14.5–E15.5). Tissue elongation builds up pressure, which is released by basement membrane remodelling, allowing YAP activation, localized cell divisions and placode budding (E15.5). SC, stem cell. Box plots show 10th to 90th percentile whiskers with 75th, 50th and 25th percentiles (**d**), or minimum-to-maximum with 75th, 50th and 25th percentiles (**c**,**f**).

could enhance Sox9 expression and compartmentalization to the outer ring of the placode. To test this, we isolated epidermal progenitor cells and placed them on microfabricated moulds that were designed to recapitulate the placode invagination (Fig. 4l). With a striking resemblance to the ring-like pattern of Sox9 expression observed in vivo, Sox9 expression was highest at the boundary of the microfabricated substrate invaginations (Fig. 4l,m). To specifically ask whether confinement is sufficient to promote Sox9 expression, we cultured epidermal progenitors in 3D collagen hydrogels and subjected them to extrinsic static compression (Extended Data Fig. 5n). Robust Sox9 expression was already observed after 3 h of compression, and the effect was sustained after 16 h of compression (Extended Data Fig. 5o,p). Collectively, these data, in line with the model predictions, indicated that changes in epidermal interfacial tension through altered myosin-IIA-mediated epidermal contractility are required to produce the contractile force for the placode to invaginate and to push down the dermal condensate. However, the full morphogenetic transformation of cell elongation and tissue deformation require additional extrinsic forces generated by a contractile fibroblast ring wrapping the placode. Surprisingly, these extrinsic forces are also essential for promoting compartmentalization of the tissue by directing patterns of Sox9 expression, similar to what has previously been shown in mouse digit development[23,24]. Whether this effect is direct or indirect remains open for future studies.

## Coordinated cell divisions controls placode downgrowth

We next asked what facilitated the profound transition of the placode from an initial epithelial thickening at E14.5 to a downward flowing, budded structure at E15.5. To this end we investigated patterns of cell divisions that have been shown to promote tissue fluidification as well as *Drosophila* tracheal budding and epithelial buckling[25–28]. Using both embryo live imaging of Histone2B-mCherry/membrane-enhanced green fluorescent protein (EGFP) mice to visualize dividing cells (R26R-RG mice[29]) and quantitative imaging of Ki67-stained fixed tissues, we examined the patterns of proliferation at E14.5 and E15.5. These analyses showed that cell divisions were substantially more frequent within the epidermis at E14.5, compared to the placode (Fig. 5a,c and Supplementary Video 7). This low mitotic frequency in the placode was associated with decreased levels of active nuclear YAP (Fig. 5a,d), the mechanosensitive transcription factor that acts within the Hippo pathway to control growth and tissue size[30,31]. In stark contrast, at E15.5, the most invaginated lower surfaces of the placodes showed higher levels of cell division (Fig. 5b,c and Supplementary Video 8), accompanied by restoration of nuclear active YAP (Fig. 5b,d), consistent with previous work showing that YAP promotes proliferation in the epidermis and hair follicle, and that cell compression promotes nuclear exit of YAP to inhibit cell division[32,33]. In line with this, inhibition of YAP in freshly isolated epidermal progenitors showed decreased levels of cell division (Extended Data Fig. 6a,b). Collectively, this indicated that mechanical confinement of E14.5 placode cells contributed to the repression of cell divisions through nuclear exclusion of YAP.

The enrichment of cell divisions outside of the placode at E14.5 and subsequently broadly within the basal cell layer of the placode at E15.5 led us to hypothesize that high levels of divisions at the placode base at E15.5 could accelerate budding. To test this hypothesis, we prepared skin explants from E14.5 mice and treated them with mitomycin C for 3 h to halt cell divisions, followed by 24 h of culture to allow further morphogenesis (Fig. 5e and Extended Data Fig. 6c,d). Indeed, blocking cell divisions at E14.5 did not substantially affect placode diameter, but strongly attenuated budding (Fig. 5e,f and Extended Data Fig. 6e).

Collectively, these data indicated that confinement of the placode at E14.5 promotes nuclear exclusion of YAP, contributing to reducing cell divisions, and the subsequent folding/buckling of the epithelium at E15.5 releases compressive stress on the placode, facilitating YAP activation, cell-cycle re-entry and mitoses, which then enhance efficient budding of the placode.

## Basement membrane remodelling facilitates placode budding

Finally, we asked what facilitated release of mechanical confinement, allowing cell-cycle re-entry and downward budding of the placode. Previous work using oncogenic mutations to model epidermal tumours showed that active basement membrane remodelling, triggering its local softening, drove the formation of tumour buds[34]. Indeed, simulations using our vertex model predicted that basement membrane softening would result in placode invagination (Fig. 6a). To test whether placode budding was associated with basement membrane softening, we first analysed the basement membrane composition using immunofluorescence, but did not observe substantial changes in the presence of the two major laminins, LN-511 and LN-332, or collagen IV, at the sites of placode compared to the surrounding epidermis, and the basement membrane appeared continuous across the placode (Fig. 6b,c). However, measurements of the stiffness of the basement membrane using atomic force microscopy revealed a slightly softer basement membrane beneath the placode at E14.5. This local softening became significantly more pronounced at E15.5 (Fig. 6d).

To address the mechanism of local basement membrane softening in the absence of substantial changes in molecular composition, we asked if proteolysis-driven turnover could play a role. We first examined the expression of extracellular matrix (ECM)-degrading enzymes by re-analysing single-cell sequencing data from the E14.5 mouse skin[35]. We analysed protease expression in the various skin cell populations (Supplementary Table 2) and found the key proteolytic enzyme matrix metalloproteinase (MMP)-2 highly expressed by the dermal fibroblasts, and the membrane bound MMP-14 specifically enriched in the placode as well as in the dermal condensate (Extended Data Fig. 6f). Other MMPs were not expressed at high levels (Supplementary Table 2). To test whether MMP-mediated matrix remodelling was required for placode invagination, we treated E13.5 skin explants for 48 h with the broad-spectrum MMP inhibitor GM6001 to block proteolytic degradation of the ECM and basement membrane. Strikingly, inhibiting matrix remodelling strongly attenuated placode budding, as visualized by both the decreased diameter and depth of the developing placodes within the explants (Fig. 6e,f and Extended Data Fig. 6g). Collectively, these data indicated that ECM remodelling triggered by proteolytic degradation is essential for placode growth by allowing pressure release and subsequent downgrowth.

## Discussion

By combining 3D vertex modelling with ex vivo biophysical measurements and genetic manipulations, we have dissected a two-step mechanism of placode morphogenesis and compartmentalization that depends on the coordination of mechanical forces across compartments. In the first step, polarized myosin activity in the epithelium, together with contractile activity from fibroblasts, generate oscillatory contractions to produce a local, mechanically confined tissue thickening (Fig. 6g). In the second step, release of pressure through local ECM remodelling facilitates tissue fluidification, which, due to the continuing oscillatory contraction, produces a plastic deformation and downward budding of the tissue (Fig. 6g). The precisely timed interactions between cell types and the forces they generate, and the transmission of these forces through the ECM and tissue mechanical changes, highlight the complexity of orchestrating morphogenesis and cell state transitions in multicompartment mammalian tissues. Interestingly, in vitro studies in experimentally confined epithelial cell lines have revealed a similar interplay between pressure, growth control, and buckling[32,36], suggesting that the mechanisms described here could be applicable to developmental processes in other epithelial organs. Furthermore, the effect of confinement on stem cell differentiation is consistent with previous work linking tension to cell fate progression through changes in chromatin organization[18,37].

Our findings are further interesting in light of recent reports showing that mesenchymal dynamics regulate feather follicle patterning in avian skin[38,39]. Collectively, these studies highlight how mechanical crosstalk

across compartments is essential to propagate symmetry-breaking and the emergence of tissue patterns. Our work also identifies an interesting distinction where the murine placode early fate specification seems to precede mechanical changes, which subsequently are essential for further cell type specification and, in particular, compartmentalization of cell fates. Interestingly, the distinct role of fibroblasts in generating forces, and adjusting pressure through remodelling the ECM, recapitulate features observed between cancer-associated fibroblasts and tumour cells during cancer progression[34,40]. These common features between placode development/budding and cancer progression are consistent with observations on cancer cells hijacking embryonal pathways to promote aggression[41–43]. This suggest that the mechanisms discovered in this study, as well as the modelling approaches developed, could be of broad relevance for various biological processes. In particular, it will be interesting to understand how stromal fibroblasts and changing patterns of epithelial contractility, pressure and tissue fluidity play roles in the initiation of cancer aggression.

## Online content

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

[1]Stem Cells and Metabolism Research Program, Faculty of Medicine, University of Helsinki, Helsinki, Finland. [2]Department of Cell and Tissue Dynamics, Max Planck Institute for Molecular Biomedicine, Münster, Germany. [3]Department of Physics and BioInspired Institute, Syracuse University, Syracuse, NY, USA. [4]Cell Biology and Cancer Unit, Institut Curie, PSL Research University, CNRS, Paris, France. [5]Cell and Tissue Dynamics Research Program, Institute of Biotechnology, Helsinki Institute of Life Science (HiLIFE), University of Helsinki, Helsinki, Finland. [6]School of Chemistry and Chemical Engineering, Shanghai Jiao Tong University, Shanghai, China. [7]Helsinki Institute of Life Science, Biomedicum Helsinki, University of Helsinki, Helsinki, Finland. [8]Wihuri Research Institute, Biomedicum Helsinki, University of Helsinki, Helsinki, Finland. [9]These authors contributed equally: Ali Hashmi, Irene Ylivinkka, Elizabeth Lawson-Keister. ✉e-mail: mmanning@syr.edu; sara.wickstrom@mpi-muenster.mpg.de

## Methods

### Mice

All mouse studies were approved and carried out in accordance with the guidelines of the Finnish national animal experimentation board (ELLA) or the Ministry for Environment, Agriculture, Conservation and Consumer Protection of the State of North Rhine-Westphalia (LANUW), Germany. Mice were maintained under a 12-h/12-h light–dark cycle, under controlled temperature and humidity (18–23 °C and 40–60%, respectively) and with constant access to food and water. *Myh9* floxed mice were obtained from the European Mouse Mutant Archive (EM:02572)[44] and crossed with the K14-Cre line[18,45] or the Twist2-Cre line obtained from JAX laboratories (stock #008712)[21]. Membrane-targeted Tomato reporter mice (R26R^mT/mG)[16] were from JAX laboratories (stock #007676)[16]. Histone2B-mCherry/membrane-EGFP (R26R-RG) mice[29] were from Riken Laboratories for Animal Resources and Genetic Engineering (LARGE) and were crossed with the K14-Cre line to obtain epidermis-specific expression. *FGF20*-LacZ mice have been previously described[7]. Embryonic ages 13.5, 14.5, 15.5 and 16.6 were used and defined by the presence of vaginal plug (embryonic day 0), limb morphology and other external criteria[46].

### Immunofluorescence and confocal microscopy

For whole mounts, embryos were collected and directly fixed in 37 °C pre-warmed 4% paraformaldehyde (PFA) at room temperature (r.t.) for 3 h. After multiple phosphate-buffered saline (PBS) washes, the skin was peeled and placed in blocking solution (0.5% Triton-X, 5% bovine serum albumin (BSA), 3% normal goat serum (NGS)) for 1 h at r.t.

For paraffin sections, tissues were fixed in 4% PFA, embedded in paraffin and sectioned. Sections were de-paraffinized using a graded alcohol series, and antigen retrieval was carried out using target retrieval solution (Dako) at pH 6 or 9 in a pressure cooker.

For cryosections, tissue samples were embedded unfixed in Tissue Tek O.C.T. (opimal cutting temperature) Compound (Sakura) and frozen, then 6–8-μm-thick sections were cut and fixed with 4% PFA. For basement membrane stainings, unfixed cryosections were stained. All samples were blocked in 5% BSA, 3% NGS.

Organoids and a 2D cell culture of primary epidermal stem cells were fixed in 4% PFA for 1 h at r.t. After PBS washes, the samples were placed in blocking solution (0.5% Triton-X, 5% BSA, 3% NGS) for 1 h at r.t.

The samples were incubated with primary antibodies diluted in blocking solution overnight at 4 °C (organoids and 2D cell culture) or at r.t. (whole mounts). For paraffin sections, antibodies were diluted in Dako antibody diluent (Agilent) overnight at 4 °C. The following primary antibodies were used: goat anti-Edar (R&D Systems, AF745; 1:200) rabbit anti-p-MLC2 (p-Ser20; Abcam, 2480; 1:100) rabbit anti-vimentin (Abcam, 185030; 1:100), rabbit anti-Sox9 (Cell Signaling, 82630; 1:100), rabbit anti-Sox2 (Millipore Sigma, AB5603; 1:200), mouse anti-Twist2 (Abcam, 50887; 1:200), mouse anti-Ki67 (Cell Signaling, 9449; 1:300), rabbit anti-active YAP1 (Abcam, 205270; 1:100), guinea-pig anti-keratin 14 (Progen, GP-CK14; 1:300), rabbit anti-collagen IV (Abcam, 6586; 1:200), rabbit anti-laminin 332 (gift from R. E. Burgeson[47]; 1:20,000), rat anti-laminin α5 (504 (ref. [48]); gift from L. Sorokin; 1:20,000) and chicken anti-β-galactosidase (Abcam, 9361; 1:500). F-actin was detected using Alexa647-conjugated phalloidin (1:500). The following secondaries (all from Invitrogen) were used to visualize primary antibodies and nuclei: anti-rabbit immunoglobulin-G (IgG) Alexa Fluor 488/568 (A11008/A10042; 1/500); anti-mouse IgG1/IgG2A Alexa Fluor 546 (A21123/A21134; 1/500); anti-rat IgG-rat Alexa Fluor 647 (A21247; 1/500); anti-goat IgG Alexa Fluor 488 (A21467); anti-chicken IgG Alexa Fluor 488 (A21449; 1/500); 4′,6-diamidino-2-phenylindole (DAPI, Invitrogen). Slides were mounted in Elvanol (Mowiol (Sigma; #81381), Dabco (Sigma; #D27802), glycerol (ThermoFisher; #BP229-1)).

All fluorescence images were collected by laser scanning confocal microscopy (SP8, Leica or LSM80, Zeiss) with the Leica Application Suite (LAS X version 2.0.0.14332) or Zeiss ZEN (Zeiss ZEN version 3.5) software. We used ×20 dry and ×40/×63 water-immersion objectives. Images were acquired at r.t. using sequential scanning of frames of 1-μm-thick confocal planes (pinhole 1). Images were collected with the same settings for all samples within an experiment.

### Ex vivo embryo imaging

Imaging was performed essentially as described previously[37]. Briefly, embryos were dissected at E14.5 or E15.5 and immobilized on imaging chambers engineered on top of Lumox-Teflon imaging plates (Sarstedt). Whole embryos were immersed in growth medium (CnT-PR, CELLnTEC Advanced Cell Systems) supplemented with calcium-depleted 10% FBS (HyClone/Thermo Fisher), 20 U ml$^{-1}$ penicillin-streptomycin (Thermo Fisher), 20 mM Hepes, 1.8 mM CaCl$_2$ and 10 ng ml$^{-1}$ human recombinant Epithelial growth factor (EGF) (Novus; NBP2-34952). Imaging was performed using an Andor Dragonfly 505 high-speed spinning disk confocal microscope (Oxford Instruments) equipped with 488-nm and 546-nm lasers, an Andor Zyla 4.2 sCMOS camera and an environmental chamber set at 37 °C, 5% CO$_2$. Acquisition was carried out with ×25, ×40 dry or ×40 water-immersion objectives using the Fusion 2.0 software. After acquisition, the image series was 3D drift-corrected using manual registration on ImageJ together with the descriptor-based series registration ImageJ plug-in, before analysis. Image analyses are described in detail in the following. Optical cross-section live image sequences were generated using the oblique slicer tool on Imaris.

### Skin explants

Embryonic back skins were dissected from E13.5 or E14.5 embryos, placed at 37 °C, 5% CO$_2$ on a 3-μm-pore-size cell culture insert (Corning) in Dulbecco's modified Eagle medium (DMEM) supplemented with Glutamax, 10% FBS, 1% penicillin-streptomycin at an air–liquid interface, as previously described[49]. Skins were dissected into two pieces, and one half was treated with DMSO as control and the other half with mitomycin C (1 μg ml$^{-1}$; Sigma), blebbistatin (10 nM; Sigma) or GM6001 (10 μM; Sigma) for 3, 24 or 48 h as indicated. After culture, tissues were fixed and processed for immunofluorescence as described above for whole mounts.

### Cell culture, microfabrication and hydrogels

Epidermal stem cells were isolated from E15.5 embryos (organoids) or P0 mice (2D culture). After dissection, skins were treated with antibiotic/antimycotic solution (Sigma A5955) for 5 min, and epidermal single cell suspensions were generated by incubating skin pieces in 5 U ml$^{-1}$ Dispase II (D4693) for 90 min at 4 °C (15.5 embryonic skins) to separate the epidermis from the dermis; followed by 0.8% trypsin for 10 min (E15.5 skins) or 30 min (P0 skins) at 37 °C. Organoid cultures were performed as described previously[50]. Briefly, cells were suspended in growth factor-reduced Matrigel (Corning) and grown in in 3C medium (MEM Spinner's modification (Sigma), 5 μg ml$^{-1}$ insulin (Sigma), 10 ng ml$^{-1}$ EGF (Sigma), 10 μg ml$^{-1}$ transferrin (Sigma), 10 μM phosphoethanolamine (Sigma), 10 μM ethanolamine (Sigma), 0.36 μg ml$^{-1}$ hydrocortisone (Calbiochem), 2 mM glutamine (Gibco), 100 U ml$^{-1}$ penicillin, 100 μg ml$^{-1}$ streptomycin (Gibco), 10% chelated fetal calf serum (Gibco), 5 μM Y27632, 20 ng ml$^{-1}$ mouse recombinant vascular endothelial growth factor (VEGF) and 20 ng ml$^{-1}$ human recombinant fibroblast growth factor-2 (FGF-2) (all from Miltenyi Biotec)). For 2D cultures, cells were seeded on glass coverslips coated with 30 μg ml$^{-1}$ collagen I (Millipore) and 10 μg ml$^{-1}$ fibronectin (Millipore) for 1 h at 37 °C. Cells were cultured in 3C medium, incubated with DMSO or verteporfin (10 μM; Stem Cell technologies) for 6 h before fixation.

For microfabrication, circular patterns (100 μm in diameter) were projected on an SU8 spin-coated silicon wafer (height, 20 μm) using the Primo UV-lithography system controlled by Leonardo software (v4.21) and mounted on a Nikon Eclipse Ti2 microscope controlled by Nikon software (NIS Elements AR.5.41.01). After UV illumination (7 mJ with 100% laser power), the wafers were baked on a hot plate at 95 °C for 4 min and transferred into propylene glycol methyl ether acetate

(PGMEA) for 4 min. After three PGMEA washes and three washes with isopropanol, the dry wafers were silanized in a vacuum chamber for 2 h. Desiccated polydimethylsiloxane (PDMS) was poured on the wafers and they were then baked overnight at 65 °C. Freshly isolated epidermal cells were seeded on $O_2$ plasma-cleaned PDMS gels in 3C medium, cultured for 24 h, then 300 μM $CaCl_2$ was added for 8 h before fixation with PFA, followed by immunofluorescence.

For compression studies, silicon cylinders (weight, 500 g) were placed on top of hydrogels at 37 °C, 5% $CO_2$ for 3 h or 16 h. After compression, samples were directly fixed and immunofluorescence were performed as described above. The uncompressed control sample was fixed together with the 3-h compression sample.

### Laser ablation

E14.5 and E15.5 embryos were collected and immobilized on imaging chambers as described above. Immediately after collection, imaging and photo-ablation were performed using a Zeiss LSM880 confocal microscope equipped with a 561-nm laser, a two-photon Ti:sapphire laser (Mai Tai DeepSee, Spectra Physics) and an incubation system set at 37 °C, 5% $CO_2$. For ablation, 800 nm and 75% laser power were used. Images were acquired with a ×40/1.30 numerical aperture oil objective. The region to be ablated was drawn as indicated in the respective figure panels, and the same thickness was used for all ablations within an experiment. Images were captured at 1-s intervals for 40 s, starting 5 s before ablation. Quantification of displacements after ablation was performed as previously described[36]. Briefly, PIV was performed on the entire field of view using a window size of 64 × 64 pixels and an overlap of 0.75. Each timepoint was compared to the last timepoint before ablation to obtain tissue displacement. For radial displacement profiles, the local normal direction with respect to the ablated region was computed as described in ref. 36. This normal direction was used to calculate the radial component of the displacement, which was then averaged as a function of distance to the ablated region. The radial displacement profile was used to calculate the total tissue displacement as a function of time to the distance displaced by the tissue at both sides of the cut. For this, the inner peak of negative displacement was subtracted from the outer peak of positive displacement.

To quantify the recoil of the placode cells and the fibroblast ring after radial ablation, regions of interest (ROIs) were defined to average the displacements obtained by PIV. These regions were obtained by computationally dilating the mask of the ablated region with 75 pixels, and then detecting the overlap between this dilated region and the placode/fibroblast ring. Displacements were averaged in each of these ROIs, and the magnitude and direction of this mean displacement in time was computed. The direction of the cut was defined by fitting its mask to a straight line. The angle difference between the mean displacements and the cut was then calculated (−90° towards the cut, 0 or 180° as parallel to the cut, and 90° away from the cut for fibroblasts and 0° towards the cut, +90 or −90° as perpendicular to the cut, and 180° away from the cut for the placode cells). Graphs in Fig. 2i show the direction of the mean displacement 40 s after ablation.

### Image analyses

For tissue-scale analysis, 2D segmentation and morphometry were performed using Tissue Analyzer (v2.3)[51]. Hair follicle placodes and dermal condensates were identified based on the expression of specific markers (FGF20, Edar, Sox9 and Sox2) and morphology. Placode morphogenesis and cell shape were quantified manually using Fiji[52] as follows. Placode cells were identified using FGF20 β-gal or Edar, after which phalloidin staining was used to determine cell contours. Only cells in contact with the basement membrane (basal cells), with clear contours, were quantified. Images were taken with high z-resolution (0.5 μm), resliced in ImageJ, and the appropriate stacks were selected individually for each cell. The measurements (apical/basal lengths; z-axis length) were then carried out manually for each single cell.

To quantify the placode depth, the distance between the epidermal basement membrane and the deepest point of the placode cells was measured. The closest distance between placodes was measured manually from the centre of one placode to the centre of the nearest neighbour placode.

For pMLC2 intensity, ROIs restricted to the placode cells and the surrounding interfollicular epidermal cells were determined, and mean fluorescence intensities were quantified. To quantify the ratio of apical/basal PMLC2 mean intensity, ROIs corresponding the apical and basal surfaces were generated, and the mean fluorescence intensities were quantified.

Fibroblast alignment around the placode was quantified using the OrientationJ plug-in (v2.0.5) from Fiji. The local window sigma was set at 5 and a cubic spline gradient was applied. The colour map represents the orientation of the fibres in degrees. A ROI defining the placode was drawn, and the fibre alignment to the closest tangent from the ROI and a function of the distance to the ROI were computed using a custom-built Python script. Fibres up to 30 μm away from the ROI were defined as 'placode' and fibres between 90 and 120 μm away from the ROI were defined as 'proximal dermis'.

Sox9-positive cells were quantified manually. The coordinates of Sox9-positive cells were extracted using Fiji. To quantify the pattern of Sox9 expression, four different parameters were measured using custom-built Python scripts: (1) the total number of Sox9-positive cells in a placode (defined by the presence of a dermal condensate and a thickening of the epithelium), (2) the local density of Sox9-positive cells within a circular area within a 10-μm radius of the placode (defined as above); (3) the average distance of Sox9-positive cells from the centre of the placode (determined as above); (4) the convex Hull area corresponding to the minimal area necessary to include all the Sox9-positive cells. To measure Sox9 mean intensities in organoids, a nuclear ROI was generated using automated thresholding of the DAPI staining, after which the mean fluorescence intensities of Sox9 were quantified within the nuclear area of interest from maximum projection images of 15 stacks. Sox9 mean nuclear intensities were normalized to the DAPI intensity of the corresponding individual nucleus. Images were collected with the same settings for all samples within an experiment.

Evaginations into the suprabasal layers from E16.6 CNL and *Myh9*-eKO embryos were quantified from paraffin sections. A straight line was drawn at the level of the epidermis/dermis boundary, and the length of the invagination was measured from this straight line to the top of the Krt14-positive cell population.

Ki67- and YAP1-positive cells were determined manually. The number of mitoses (from metaphase to telophase) from DMSO- or mitomycin-treated skin explants was identified using DAPI staining and manually quantified.

### 3D tissue rendering and 3D cell shape

Images were acquired from freshly isolated unfixed mTmG E14.5 embryos with the Andor Dragonfly 505 high-speed spinning disk confocal microscope. Images were acquired as z-stacks of 1-μm-thick optical sections, and cells were segmented and tracked across every z-stack using Tissue Analyzer, after which x−y cell dimensions were measured and treated as pseudo-time. To ensure accurate segmentation of the cell height, a nuclear bounding box was determined in parallel from skin whole mount stained with DAPI, and imaged with higher resolution using a laser scanning confocal microscopy (SP8X; Leica) with a ×63 glycerol immersion objective. 3D rendering was then performed using a custom-built code in Mathematica as follows. The cell junctions were manually corrected with a combination of Tissue Analyzer and Mathematica (v12.3), and tracking was performed to stitch the cells across the stacks. Cells that appeared in fewer than three frames were discarded from the analysis. ROIs were then converted from image objects to mesh regions in Mathematica by first applying a 3D Gaussian filter

over a radius of two pixels to smooth the roughness and then applying a 'Dual Marching Cubes' algorithm using the 'ImageMesh' function. The cellular mesh regions were then rendered appropriately in a box with the scaling parameters of the image stack. The computations for cellular volume and the associated parameters were performed on the mesh objects and on point clouds derived from the mesh.

## PIV

PIV was performed using PIVLab version 2.59. Live imaging movies of embryos were first imported into PIVLab, where a multiple-pass windowed fast Fourier transform (FFT) based approach was utilized to track tissue-scale flows for both sagittal and axial cross-sections. To correct for variations in intensities and prevent tracking errors, a CLAHE (contrast limited adaptive histogram equilization) was performed with a box of size 10 pixels. A denoising was also performed on the resulting images using Wiener noise filter with a maximum pixel count of 3. The tracking was initially done on test images by varying the window sizes of the first and second pass and, where applicable, the third pass. Below a certain threshold, performing PIV resulted in patterns of vector fields that were determined to be insensitive to the choice of window size. Based on this, a constant size of 100 pixels was selected for the first pass, with the second pass set as 50 pixels with an overlap between first-pass windows set to 50 pixels. For PIV on frontal cross-sections from time-lapse images, masks were manually drawn over regions of the epithelium, and a three-pass windowed FFT was performed with the first window size set to the maximum allowable value determined by the algorithm. The resulting vector fields were visually inspected, and any spurious vectors were manually rejected from the analysis. As the maximum allowable window size for a sagittal cross-section was relatively smaller than the spacing between lateral junctions, the strength of the vector field was smoothed within PIVLab to account for the noise. The vector fields were utilized to compute divergence maps and strain rates. To compute the mean divergence within the placode, a circular ROI covering the placode was generated and imported into PIVLab, where the mask was overlaid on the divergence field. To compute the mean divergence of the interfollicular epidermis, placodes were masked out. The values were subsequently integrated within the regions to yield a mean value.

To plot the mean divergences from multiple embryos, the divergence of each placode and its surrounding interfollicular epidermis were aligned according to the first observed peak of negative divergence.

## 3D vertex model and simulations

The 3D vertex model code was adapted from ref. 53, based on previous work[54,55]. A multilayered 3D vertex model tissue structure was generated to capture the interactions between the basement membrane, basal cells and suprabasal cells. A subset of the basal cells was labelled as placode cells, with potentially distinct mechanical properties. Each cell in the network was represented as a polyhedron with shared vertices and edges such that the tissue was completely confluent, with no gaps or holes in the tissue. Although the basement membrane was not composed of biological cells, we used vertex cells with a special energy functional, described below, to represent its mechanics. The multilayered structure was initialized from a Voronoi tessellation of a uniform 3D point pattern using the Voro++ library[56], and we minimized the vertex model energy function (equation (1) below) before the various cell types were specified. Periodic boundary conditions were maintained at the edges of the simulation domain in all directions. Neighbour exchanges (which are called T1 transitions in 2D, but are generally more complicated in 3D) were allowed via I–H and H–I transitions as described in ref. 54. Each cell was labelled as one of four: suprabasal, basal, placode or basement membrane.

We explored a standard vertex model energy functional in which each cell has a characteristic surface area $S_0$ and volume $V_0$. The preferred surface area is generated from a combination of cell adhesion, which seeks to elongate the surface area between the cells, and contractility, which shrinks these interfaces, while the characteristic volume is a consequence of cell or volume incompressibility. This gives rise to the standard vertex energy functional ($E$) of the tissue in 3D, $E = \sum_i K_V (V_i - V_0)^2 + K_S (S_i - S_0)^2$, where we sum over the energy contribution from each cell $i$. The $K_V$ and $K_S$ terms represent the moduli corresponding to bulk volumetric and interfacial surface tension perturbations, respectively. Although some versions of vertex models only include linear terms in the surface area, we include higher-order terms here, as they are required to observe fluid-like behaviour in the limit of small fluctuations, and we wish to allow for that possibility.

The dynamics of the $i$th vertex in the network is given by

$$\frac{dr_i}{dt} = -\mu \frac{\partial E}{\partial r_i} + \langle v_0 \hat{n}_j \rangle \quad (1)$$

where $r$ is the position vector associated with the vertex and $\mu$ is the inverse of a frictional drag. The last term captures active fluctuations in tension, by allowing each vertex $i$ in cell $j$ to fluctuate in a random direction given by unit vector $\hat{n}_j$ at every time step, as Gaussian white noise on a unit sphere:

$$\frac{d\hat{n}_j}{dt} = \sqrt{2D_r} \left( E - \hat{n}_j \hat{n}_j \right) \xi_j \quad (2)$$

where $E$ is the identity tensor, and $\xi_j$ is the Gaussian white noise with $\langle \xi_j \rangle = 0$ and $\langle \xi_j \rangle (t) \langle \xi_j \rangle (t') = \delta_{ij} \delta(t - t') E$. The magnitude of the fluctuations is controlled by $v_0$.

Forces are computed on the vertices at each time step, and a simple forward Euler integration scheme is used to integrate equations (1) and (2).

The mechanics of this model—for homogeneous cell types—is largely controlled by a dimensionless shape coefficient, $s_0 = S_0/V_0^{2/3}$. If cells are able to achieve their preferred shape $s_0$, the tissue will behave as a fluid where cells flow and rearrange. However, if cells cannot achieve their preferred shape due to geometric incompatibility, the tissue will be solid-like and the observed cell shapes will be pinned at a value of $s_0$ close to 5.4 in 3D (refs. 57,58). This rigidity transition has been studied extensively in both 2D and 3D models for both Voronoi and vertex model simulations[57–62].

Additionally, to capture heterotypic interactions between different cell types in the stratified epithelium, we also allow cells of different types to regulate the tension between interfaces they share, leading to the following energy functional for heterogeneous tissues:

$$E_{cell} = \sum_i K_V (V_i - V_0)^2 + K_S (S_i - S_0)^2 + \sum_j \sigma_{\alpha\beta} S_{ij} \quad (3)$$

Here, $\sigma_{\alpha\beta}$ represents the additional interfacial tension between cells of type $\alpha$ and $\beta$, and $S_{ij}$ is the surface area joint between cells $i$ and $j$. This mechanism has been shown to create robust sorting and sharp boundaries between cells of different types[63,64]. In practice, standard values for 3D vertex model parameters—$K$ of -1, $S_0$ of -5.4 (refs. 57,58) and $\sigma_{\alpha\beta}$ of -1 (refs. 63,64)—generate an energy functional where the terms that are linear in surface area $S_i$ have negative coefficients. This is not problematic due to the second-order restoring forces proportional to $S_i^2$. Therefore, we refer to the magnitude of these coefficients as 'wetting coefficients' instead of tensions. Although we use standard parameters for vertex models here (Supplementary Table 1), we have confirmed that our results are similar if we study positive linear coefficients for the surface areas.

We represent the basement membrane as small cells that interact via the same vertex energy functional (equation (3)), with a small $s_0$ that is deep in the solid regime so that the region behaves as an elastic solid.

In addition, we explicitly introduce a spring network to the interface between the basement membrane and the surrounding tissue to better mimic the mechanics of the ECM. This takes form as an additional term added to equation (3), where the sum is along all of the edges that are on the interface between the basement membrane and the basal layer:

$$E_{\text{spring}} = \frac{1}{2} k_m \sum_e (l_e - l_0)^2 \qquad (4)$$

where $k_m$ is a spring stiffness, and $l_0$ is the rest length of the spring along an edge labelled by index $e$. This additional term creates an additional energetic penalty for interfaces between the placode and basal cells and the basement membrane.

Although this term is nonlinear, we next demonstrate that we can approximate the behaviour of the springs as an extra contribution to the effective interfacial tension $\sigma_k$, when there are small perturbations to the interface between the basement membrane and the basal cells. We examine a simple three-layer system of just basal and suprabasal cells with a spring network between the basal cells and the basement membrane. Once this system reaches a steady state in the presence of active fluctuations, we extract a typical spring network configuration from the vertex model by saving the locations and neighbours of all the vertices. We then extract the spring network only, and volumetrically dilate the network isotropically so that the box size $L_i \to L_i(1 + \varepsilon)$, where $\varepsilon$ is the volumetric strain. Next we calculate the total surface area of the stretched spring network and its new energy for various different strains. From the slope of these data, we can extract the effective surface tension for stretching the network, which we call $\sigma_k$. This additional effective tension is directly controlled by the rest length $l_0$. We choose the rest length such that the interface between the basal cells and the basement membrane is twice as stiff as the interfaces between cells, to capture data from atomic force microscopy (AFM) experiments demonstrating that the basement membrane is much stiffer than the cell cortices[65].

With this model in hand, we sought to explore whether the isotropic elongation of the placode cells and the invagination phenomenon is a result of the intrinsic forces generating the placode cells or the result of the extrinsic forces applied on the placode cells via cells outside the placode. To this end, we utilized the 3D vertex model to examine both cases.

**Model 1—deformations under intrinsic forces.** To capture the ability of cells to alter their own intrinsic properties, we investigated the case where the placode cells can modulate their adhesion and interfacial tension with the neighbouring cells. In the vertex model, there are two ways for cells to adjust their interactions with their neighbours—either by changing their preferred surface area, which represents its contractility and homotypic adhesion, or by changing $\sigma$ (which appears in the last term in equation (3)), representing an additional interfacial tension driven by heterotypic interactions with other cell types. To quantify the relative levels of adhesion and tension at any interface, we define a total wetting coefficient $\tilde{\sigma}$ on each interface as

$$\tilde{\sigma}_{ij} = \frac{\sigma_{ij} - 2K_s S_0 + (\delta_{im} + \delta_{jm})\sigma_k}{\sigma_{bb} - 2K_s S_{0,b}} \qquad (5)$$

The third term takes into account the extra effective tension generated by the spring network if the interface is touching the basement membrane. As one can see from equation (5), we have normalized all total wetting coefficients (for the placode lateral interfaces $\sigma_{pp}$, placode–suprabasal interfaces $\sigma_{ps}$ and placode–basement membrane interfaces $\sigma_{pm}$) by the total wetting coefficient on the lateral interfaces between basal cells, $\sigma_{bb} - 2K_s S_{0,b}$. In general, increasing $S_0$ will cause the placode cells to elongate. However, this will also change the effective wetting coefficient defined by equation (5). For simplicity, we examined the parameter space in terms of total wetting coefficient on

each interface. In practice, this means that when we change $S_0$, we also adjust the heterotypic interfacial tension on each interface to ensure the relative ratio of total wetting coefficients across each cell interface is constant. Thus, we performed simulations over a broad range of different values of the wetting coefficients. We found that the morphology of the placode depends on two values: the wetting coefficient between lateral interfaces of placode cells ($\tilde{\sigma}_{pp}$) and the ratio of the placode–basement membrane wetting to placode–suprabasal coefficients, $\frac{\tilde{\sigma}_{pm}}{\tilde{\sigma}_{ps}}$.

**Model 2—deformation under extrinsic forces.** Next, we examine contractile forces such as those generated by the fibroblast ring around the placode cells. To capture this effect in a minimal way, we impose an inward-directed, compressive body force on basal cells outside the placode, which takes the form

$$\mathbf{f}_{\text{rad}} = f_r e^{-\frac{|\mathbf{r} - \mathbf{r}_c| - \mathbf{r}_{\text{plac}}}{V_0^{1/3}}} \frac{\mathbf{r} - \mathbf{r}_c}{|\mathbf{r} - \mathbf{r}_c|} \qquad (6)$$

where $f_r$ is the maximum magnitude of the external force on the basal cells just outside the placode, and where the force falls off exponentially away from the placode's centre, $\mathbf{r}_c$, over a small, characteristic length scale $V_0^{1/3}$. Note that a constant placode shape $s_0$ and tensions are maintained between the placode cells in this model. Also note the caveat that no cell divisions or dermal condensate (dermal papilla) structure were incorporated into the model. Details of the parameter selection are provided in Supplementary Table 1.

## Computing curvature and cell elongation in simulations and experiments

The curvature of the simulated tissue was determined in a two-step process. A paraboloid was first fitted to the basal vertices shared between the placode cells and those representing the basement membrane. Subsequently, a Gaussian curvature was determined at the deepest point for the fitted surface. In the embryonic tissue whole mounts, we manually identified points on the interface between the placode cells and the membrane in 2D cross-sectional views. These points were fitted to a parabola, assuming that the associated parabola was symmetric along the missing axis, and generated the associated paraboloid. The same procedure as applied for the simulation data was then followed to extract the curvature. To quantify the cell elongation in simulations, the depth of the paraboloid was extracted and combined with the average height of the cells in the basal layer.

## Single-cell RNA-sequencing analysis

Raw sequence data for E14.5 mouse skin[35] was obtained from NCBI GEO (accession no. GSE122043). The quality of the data was checked using FastQC (v0.11.9). STARsolo aligner (v2.7.7a)[66] was used to align sequenced reads to the mouse reference genome (GRCm38), followed by nUMI and barcode counting, constructing the nUMI count matrices. The quality of the sequence alignment was checked using MultiQC (v1.9)[67]. nUMI matrices were filtered, centred, normalized, clustered, and marker genes extracted using Scanpy (v1.8.1)[68]. Owing to the strong batch effects observed between the biological replicates, replicates were not merged, but were analysed independently. Data for 'control replicate 2' from E14.5 mice are shown. The Leiden algorithm was used to cluster cells at different resolutions between 0.1 and 1 with intervals of 0.1. Resolution 0.8 is displayed. Marker genes were obtained with default parameter of the Scanpy's find-marker function, which uses the $t$-test_overestim_var method for differential expression analysis and the Benjamini–Hochberg method for $P$-value correction. Marker gene lists were automatically selected using rank gene groups (--rgg option) in scanpy functions. Gene Ontology enrichment was performed using a hypergeometric test implemented in gseapy (v0.10.5)[69].

## AFM

AFM measurements of hair follicle basement membranes were performed on freshly cut 16-µm-thick sagittal cryosections of embryonic skin using a JPK NanoWizard 2 (Bruker Nano) atomic force microscope mounted on an Olympus IX73 inverted fluorescent microscope (Olympus) and operated with JPK SPM Control Software v.5. Cryosections were equilibrated in PBS supplemented with protease inhibitors, and measurements were performed within 20 min of thawing the samples. Triangular non-conductive silicon nitride cantilevers (MLCT, Bruker Daltonics) with a nominal spring constant of 0.07 Nm$^{-1}$ were used for the nanoindentation experiments, which were performed in combination with optical microscopy to identify the epidermal–dermal boundary. Basement membrane was identified by its characteristic height and stiffness properties by generating high-resolution indentation maps starting from the basal surface of epidermal cells and towards the dermis. For all indentation experiments, forces of up to 3 nN were applied, and the velocities of the cantilever approach and retraction were kept constant at 2 µm s$^{-1}$, ensuring an indentation depth of 500 nm. All analyses were performed with JPK Data Processing Software (Bruker Nano). Before fitting the Hertz model corrected by the tip geometry to obtain Young's modulus (Poisson's ratio of 0.5), the offset was removed from the baseline, the contact point was identified, and cantilever bending was subtracted from all force curves.

## Statistics and reproducibility

Statistical analyses were performed using GraphPad Prism software (GraphPad, version 9). Statistical significance was determined by the specific tests indicated in the corresponding figure legends. Only two-tailed tests were used. In all cases where a test for normally distributed data was used, a normal distribution was confirmed with the Kolmogorov–Smirnov test ($\alpha = 0.05$). All experiments presented in the Article were repeated in three independent biological replicates at least. Experimental groups were based on mouse phenotypes, and drug treatments were performed on split tissues from the same mouse. No statistical methods were used to predetermine sample sizes; they were based on experience with the methodology and are similar to those reported in previous publications[11–15]. Data collection and analysis were not performed blinded, as the phenotypes were visibly obvious to the researchers and image analyses were automated. For AFM measurements, outlier identification was carried out to remove rare individual measurements that represented apparent artefacts. No other data points were removed from the experiments.

## Reporting summary

Further information on research design is available in the Nature Portfolio Reporting Summary linked to this Article.

## Data availability

Data supporting the findings of this study are available from the corresponding author on request. Previously published single-cell sequencing data from the E14.5 mouse skin data that were re-analysed here are available under accession code GSE122043. Source data are provided with this paper.

## Code availability

All scripts and custom code are available at WickstromLab and Zhang-Tao GitHub.

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

## Acknowledgements

We thank C. M. Niessen and L. C. Biggs for advice and critical reading of the manuscript, A. M. Luoto, H. Ahola and C. Ortmeier for expert technical assistance, M. M. Nava for help with the AFM analyses, the Max Planck Institute BioOptics and Biomedicum Helsinki Imaging Unit for imaging support and HiLIFE Laboratory Animal Centre Core Facility, University of Helsinki, for support with animal experiments. This work was supported by the Sigrid Juselius Foundation, Helsinki Institute of Life Science, Wihuri Research Institute, European Research Council (ERC) under the European Union's Horizon 2020 research and innovation programme (grant agreement 770877 – STEMpop), and the Academy of Finland Center of Excellence BarrierForce (all to S.A.W.). This project received funding from the European Union's Horizon 2020 research and innovation programme under Marie Skłodowska-Curie grant agreement no. 101032331 to C.V. Y.A.M. was the recipient of the EMBO Long-Term fellowship ALTF 728-2017 and Human Frontier Science Program fellowship LT000861/2018. M. L. Manning and E.L.-K. acknowledge support from NIH R01HD099031 and the Simons Foundation (grants nos. 454947 and 446222).

## Author contributions

C.V. designed and performed most of the experiments and analysed data. A.H. and E.L.-K. developed and performed image analyses and vertex model simulations. I.Y. and S.-M.M. performed AFM and imaging experiments. Y.A.M. designed and supervised AFM and compression measurements. B.Y. analysed sequencing data. F.B. analysed imaging data. C.P.-G. and D.M.V. performed and analysed laser ablation experiments. T.Z. generated the initial vertex model code. M. L. Mikkola provided critical mouse models, conceptual advice and designed experiments. M. L. Manning provided conceptual advice, analysed data and designed and supervised the modelling. S.A.W. conceived and supervised the study, designed experiments, analysed data and wrote the paper. All authors commented and edited the manuscript.

## Funding

## Competing interests

The authors declare no competing interests.

## Additional information

**Extended data** is available for this paper at https://doi.org/10.1038/s41556-023-01332-4.

**Correspondence and requests for materials** should be addressed to M. Lisa Manning or Sara A. Wickström.

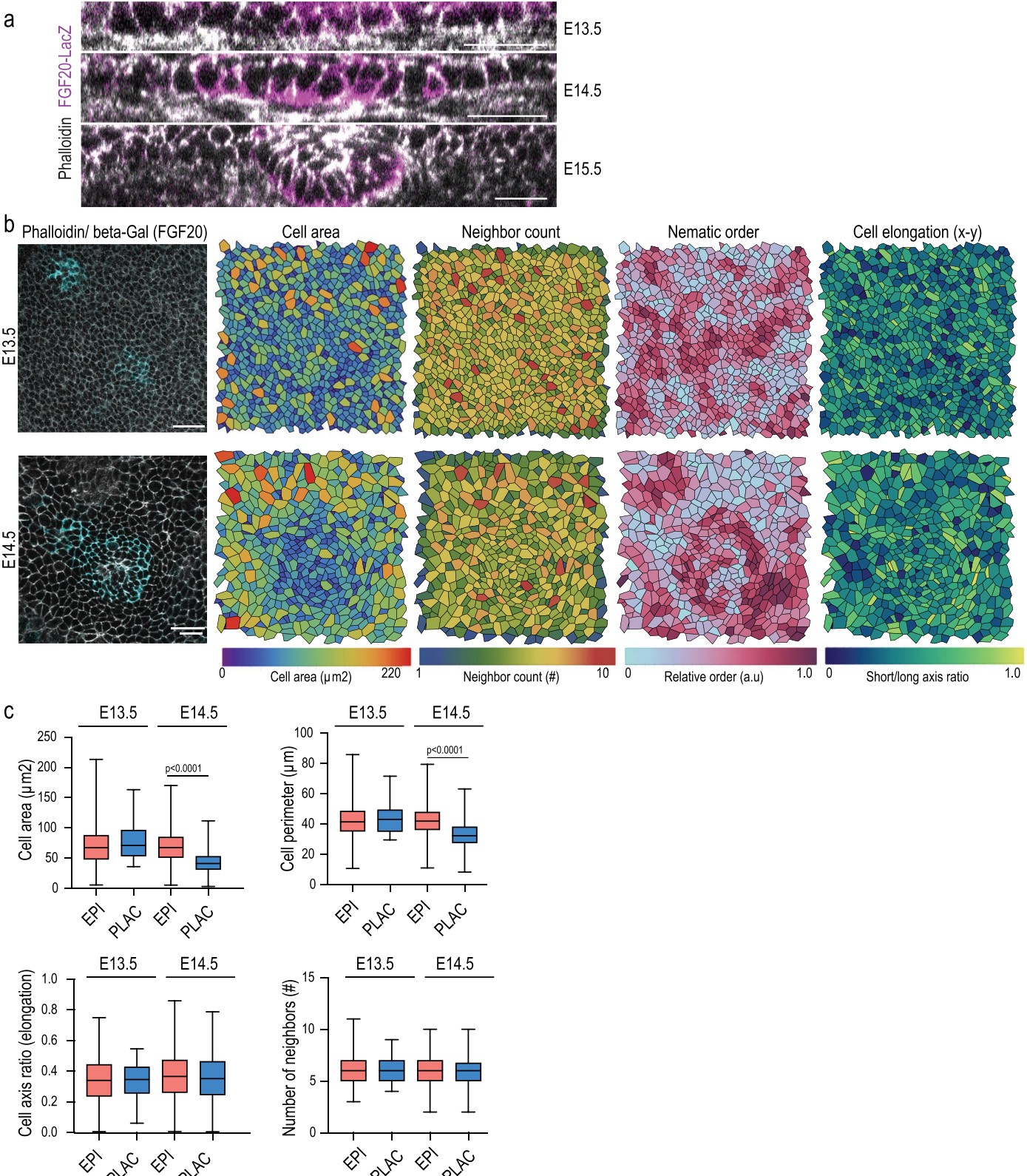

**Extended Data Fig. 1 | Early cell fate specification precedes morphological changes. (a, b)** Representative images of E13.5 - E15.5 cell cortexes and FGF20-expressing (β-gal-stained) placode cells from sagittal (a) and planar panels (b, left panels). Cell area, neighbor count, nematic order and cell elongation (in x-y) were quantified. Scale bars 50 μm. Images representative of 3 mice/group. **(c)** Quantification of morphological parameters from images in (b). No substantial differences in cell morphology between placode and epidermis are observed at E13.5 (n = 2635 (E13.5 EPI); 54 (E13.5 PLAC); 2657 (E14.5 EPI); 493 (E14 PLAC); 2982 (E15.5 EPI) and 422 (E15.5 PLAC) cells pooled across 3 mice /group). Minimum-to-maximum box plots show the 75th, 50th, 25th percentiles.

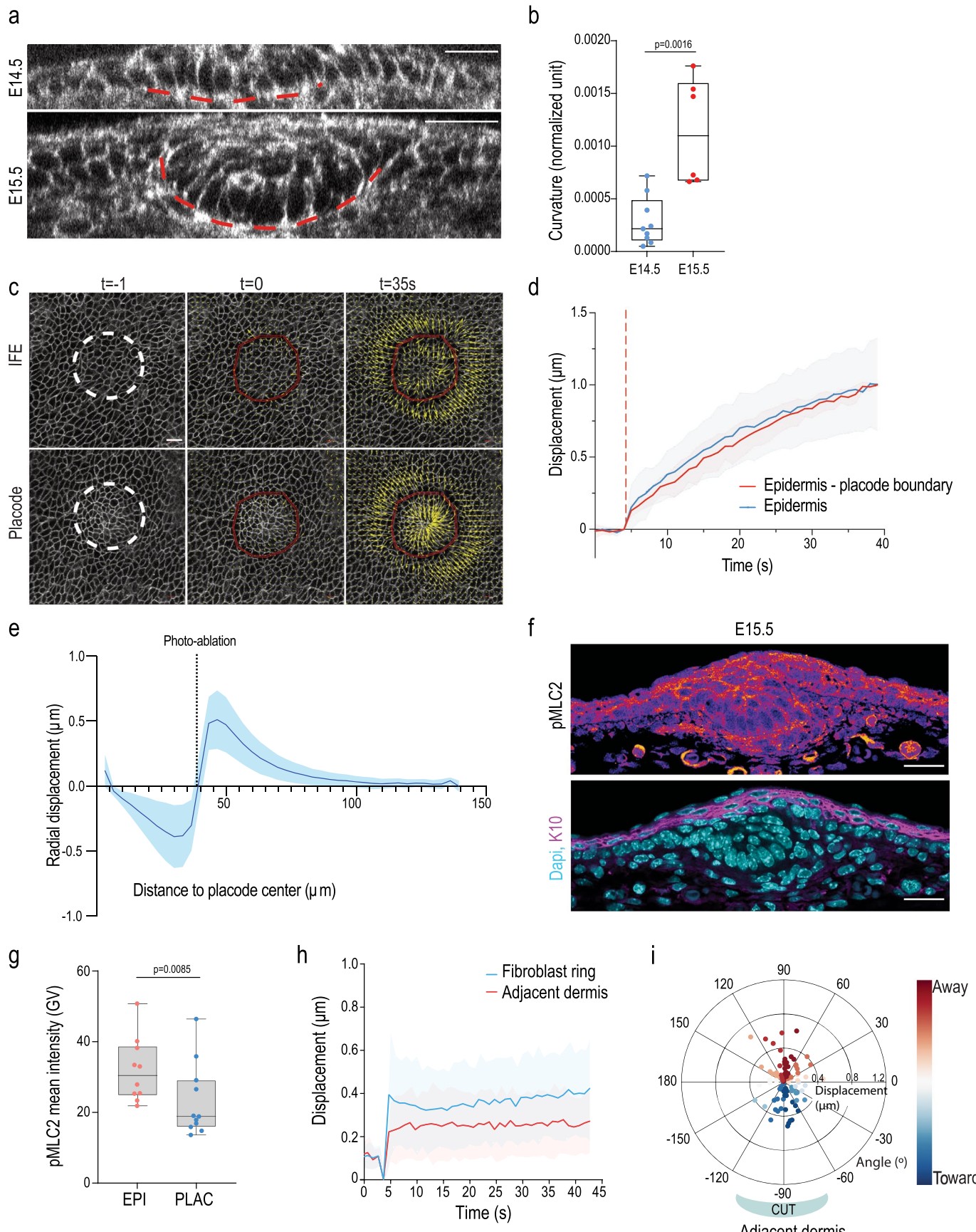

**Extended Data Fig. 2 | See next page for caption.**

**Extended Data Fig. 2 | Quantification of forces during tissue deformation.**
(**a, b**) Representative images (a) and quantification (b) of placode curvature
at E14.5 and E15.5 (n = 9 (E14.5) or 6 (E15.5) placodes pooled across 3 embryos/
stage; Mann-Whitney). Scale bars 20 µm. (**c**) Representative snapshots of whole
embryo live imaging with regions of laser ablation highlighted (dotted circle)
and vectors showing recoil magnitude and direction after ablation (arrows).
Scale bars 20 µm. (**d**) Quantification of mean displacement after laser ablation
shows similar displacements in the epidermis and at the placode/epidermis
boundary (n = 7 mice; Mean +/- SD). (**e**) Radial recoil as a function of distance to
placode center from cuts between placode and epidermis (n = 7 mice with 6–10
placodes/mouse). (**f**) Representative DAPI, Keratin-10 and pMLC2-stained images
from E15.5 mouse skin sections from 3 independent experiments. Scale bars
20 µm. (**g**) Quantification shows a decrease in pMLC2 intensity in placode at E15.5
(n = 10 placodes from 3 explants/condition; paired *t*-test). (**h**) Quantification
of mean displacement after laser ablation shows larger displacement in the
fibroblast ring than in the adjacent dermis (n = 8 mice with 4–10 placodes and
adjacent dermis/mouse). (**i**) Quantification of the direction and magnitude of
the mean displacement over time shows no preferential direction of recoil in the
fibroblasts adjacent to the ring (n = 97 adjacent dermis from 8 mice). Minimum-
to-maximum box plots show the 75th, 50th, 25th percentiles.

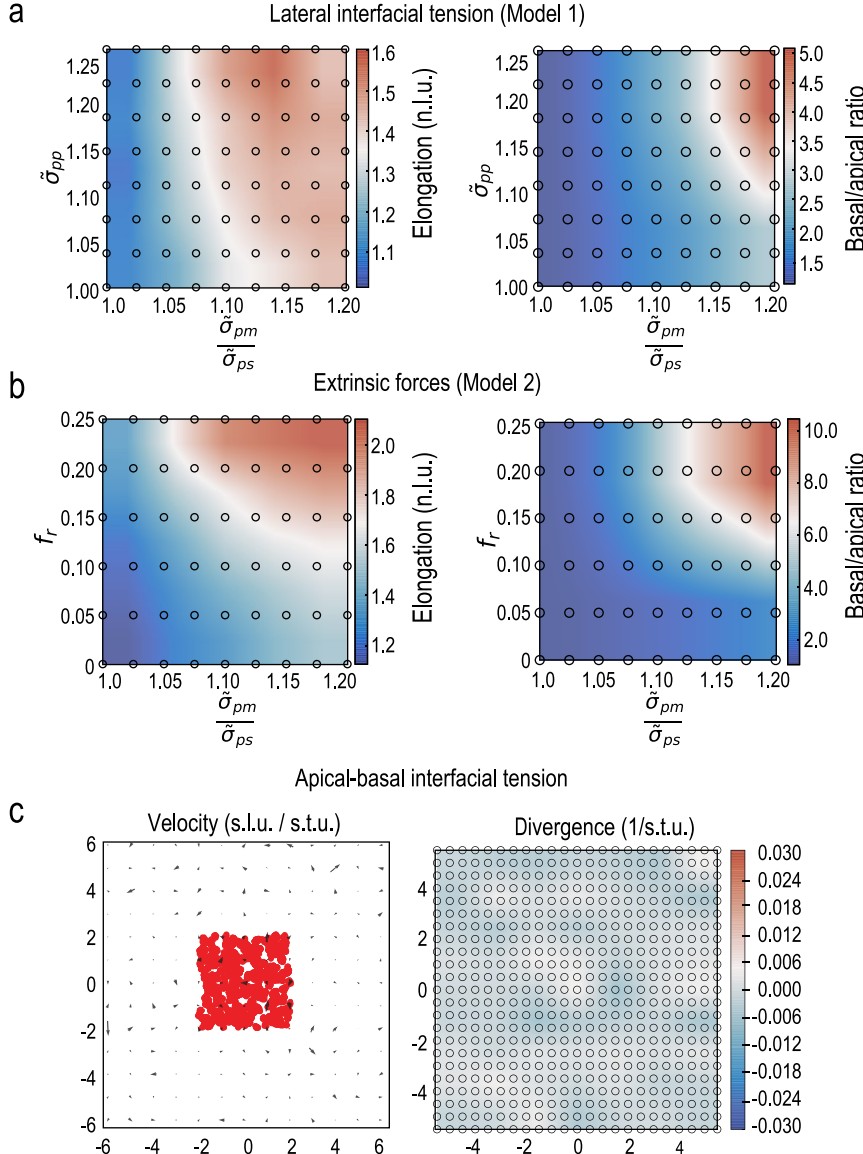

**Extended Data Fig. 3 | 3D vertex modelling to predict roles of lateral interfacial tension and extrinsic forces in placode formation. (a, b)** Phase diagrams of cell elongation or changes in apical-to-basal surface length induced by interaction of lateral wetting coefficient (a) or extrinsic lateral forces (b).

(**c**) Velocity map and mean divergence heatmap of model simulations applying high basal to apical wetting coefficient. n.l.u.= normalized length unit; s.t.u.= simulated time unit; s.l.u.= simulated length unit.

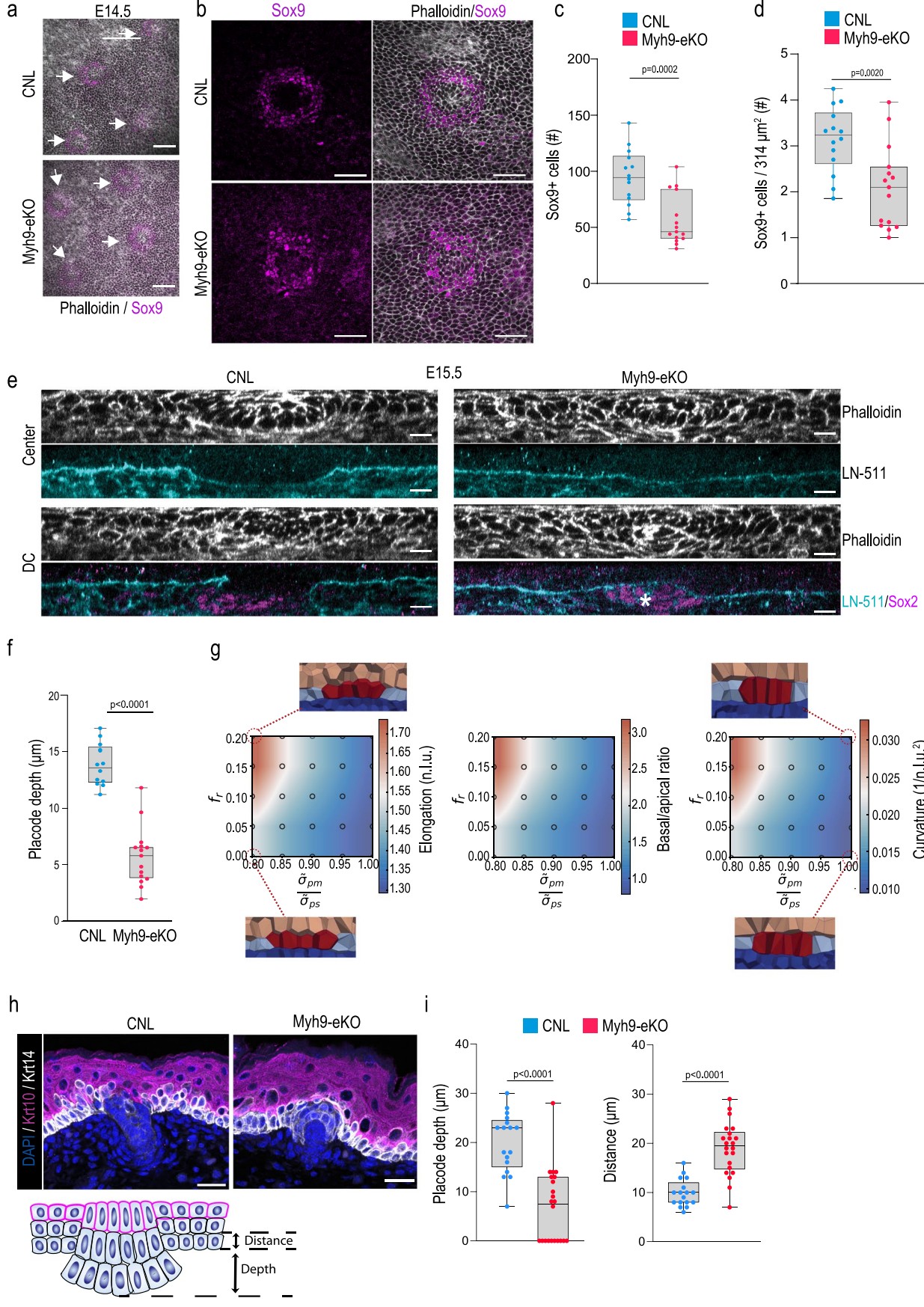

**Extended Data Fig. 4 | See next page for caption.**

**Extended Data Fig. 4 | Epidermal contractility is required for placode downward budding. (a)** Representative top views from basal layer of E14.5 CNL and *Myh9*-eKO mouse epidermis. Arrows mark Sox9-positive placodes. Scale bars 50 μm. Images representative of 4 mice/group. (**b-d**) Representative images (b) and quantification (c, d) of Sox9 pattern of expression from E15.5 CNL and *Myh9*-eKO mouse epidermis. Note reduced number of Sox9-positive cells (c) and local density (d) in *Myh9*-eKO placodes. Scale bars 50 μm. (n = 14 (CNL) and 15 (*Myh9*-eKO) placodes from 3 mice/group; (c) Mann-Whitney; (d) Student's *t*-test). (**e**) Phalloidin, LN-511 and Sox2-stained sagittal cross sections of CNL and *Myh9*-eKO mouse epidermis at E15.5. Sections from placode center and dermal condensate (DC; red asterisk) are shown. Note position of DC within the placode in *Myh9*-eKO. Scale bars 10 μm. Images representative of 3 mice/group. (**f**) Quantifications of placode depth from CNL and *Myh9*-eKO mice at E15.5 shows reduced placode depth in *Myh9*-eKO mice (n = 12 (CNL) and 15 (*Myh9*-eKO) placodes from 3 mice (left); Student's test). (**g**) Phase diagrams of cell elongation, placode curvature or changes in apical-to-basal surface length induced by varying basal to apical wetting coefficient. The model predicts that if apical interfacial tension is reduced the placode will evaginate into the suprabasal layer instead of invaginate into the basement membrane. n.l.u = normalized length unit. (**h, i**) Cross sections (h) and quantification (i) of second-wave placodes in control (CNL) or *Myh9*-eKO embryos at E16.5 showing placode depth (left) and the evagination (right). Schematic shows how measurements were performed. Scale bars, 20 μm (n = 17 (CNL) and 22 (*Myh9*-eKO) placodes from 3 mice; Mann-Whitney (I, left) and Student's test (I, right)). Minimum-to-maximum box plots show the 75th, 50th, 25th percentiles.

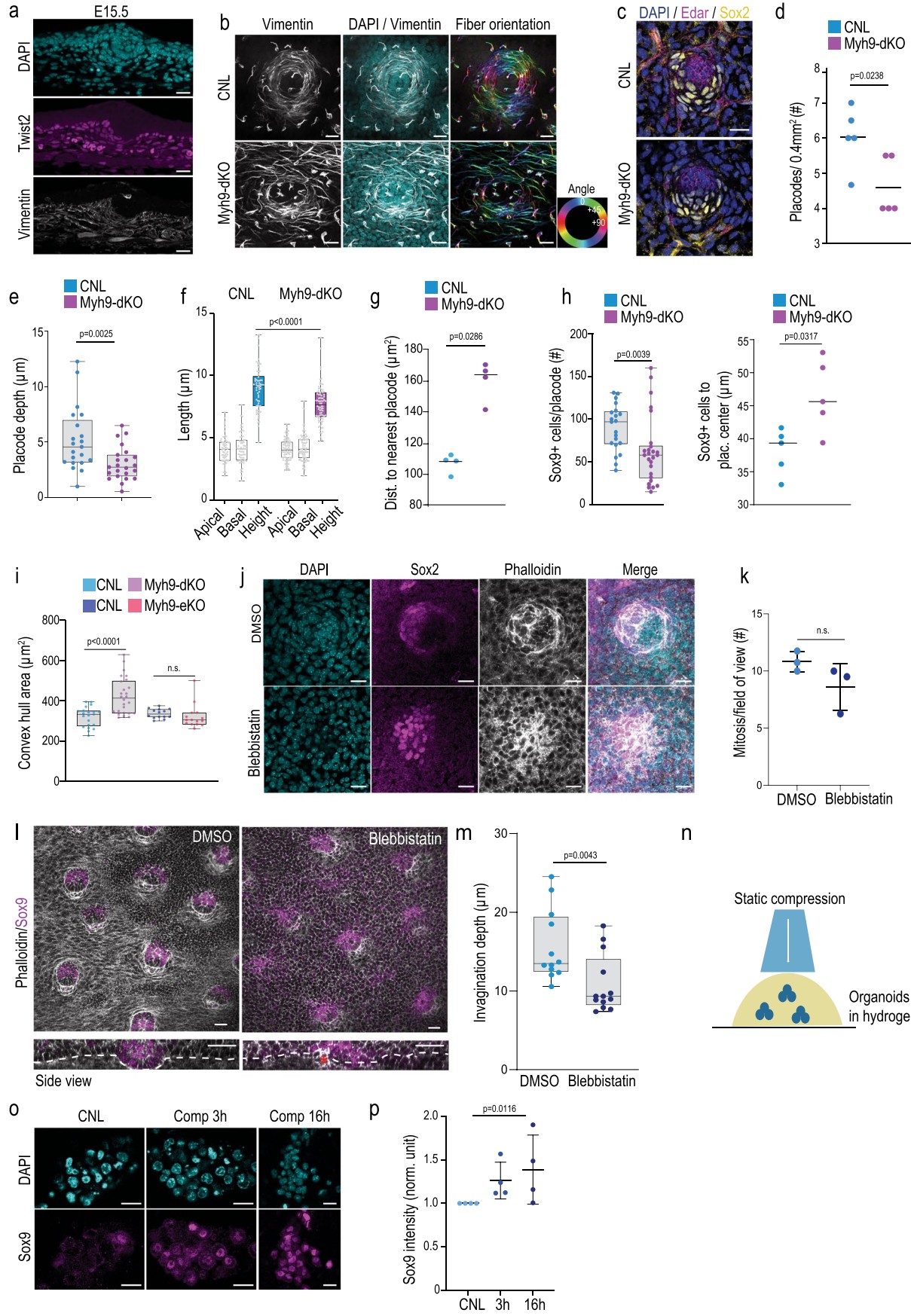

**Extended Data Fig. 5 | See next page for caption.**

**Article**

**Extended Data Fig. 5 | Contractility is required for placode development.**
(**a**) E15.5 epidermis stained for DAPI, Vimentin and Twist2. (**b**) Whole mount
dermal level top views of E16.5 skin stained for Vimentin. Fiber orientation (right)
is shown in degrees. Note attenuated alignment of the fibroblasts around the
placode in *Myh9*-dKO compared to control (CNL). (**c**) Whole mount top views
stained for Sox2, Edar and DAPI show dermal condensate in E14.5 *Myh9*-dKO.
(**d, e**) Quantifications of placode numbers (d) and depth (e) from CNL and *Myh9*-
dKO at E14.5 (n = 5 mice/group (d) or n = 21 (CNL) and 23 (*Myh9*-dKO) placodes
from 5 mice/group (e); Mann-Whitney). (**f**) Quantifications of placode cell
surface lengths at E14.5. (n = 113 (CNL) and 126 (*Myh9*-dKO) cells pooled across
5 mice/group; 2-way ANOVA/Tukey's. (**g**) Quantifications of the distance
between placodes at E16.5 (n = 4 mice/group; Mann-Whitney). (**h**) Quantification
of Sox9-postive cell number (left) and mean distance from placode center
(right) at E14.5. (n = 21 (CNL) and 26 (*Myh9*-dKO) placodes from 5 mice/group;
Mann-Whitney; left and n = 5 mice/group; Mann-Whitney; right) (**i**)
Quantification of minimum area of Sox9-positive cells (convex hull) shows a
wider distribution of Sox9 expression in *Myh9*-dKO (n = 21 (CNL), 26 (*Myh9*-dKO),

14 (CNL) and 15 (*Myh9*-eKO) placodes pooled across 3 mice (*Myh9*-eKO / CNL)
or 5 mice (*Myh9*-dKO / CNL); one-way ANOVA/Tukey's). (**j**) Top views of skin
explants treated with Blebbistatin for 24 h starting at E14.5, stained with DAPI,
Phalloidin and Sox2 to identify dermal condensate. (**k**) Quantification of mitoses
in DMSO- or Blebbistatin -treated explants for 24 h starting at E14.5 (Mann-
Whitney; n = 3 explants/condition; mean +/- SD). (**l, m**) Representative images
(l) and quantification (m) of phalloidin-stained Blebbistatin-treated explants
(10 nM; for 24 h starting at E14.5). Note attenuated placode invagination and
indentation of placode by dermal condensate (asterisk) in blebbistatin-treated
explants (n = 12 (DMSO) and 13 (Blebbistatin) placodes from 3 mice; Mann-
Whitney). (**n**) Schematic representation of organoid compression experiments.
(**o, p**) Representative images (o) and quantification (p) of Sox9-stained control
and compressed organoids (n = 50 (CNL), 56 (3 h) and 40 (16 h) organoids
pooled across 4 independent experiments; Kruskal-Wallis/Dunn's; mean +/- SD).
Minimum-to-maximum box plots show 75th, 50th, 25th percentiles. Scale bars
30 µm (l) or 30 µm (all other panels).

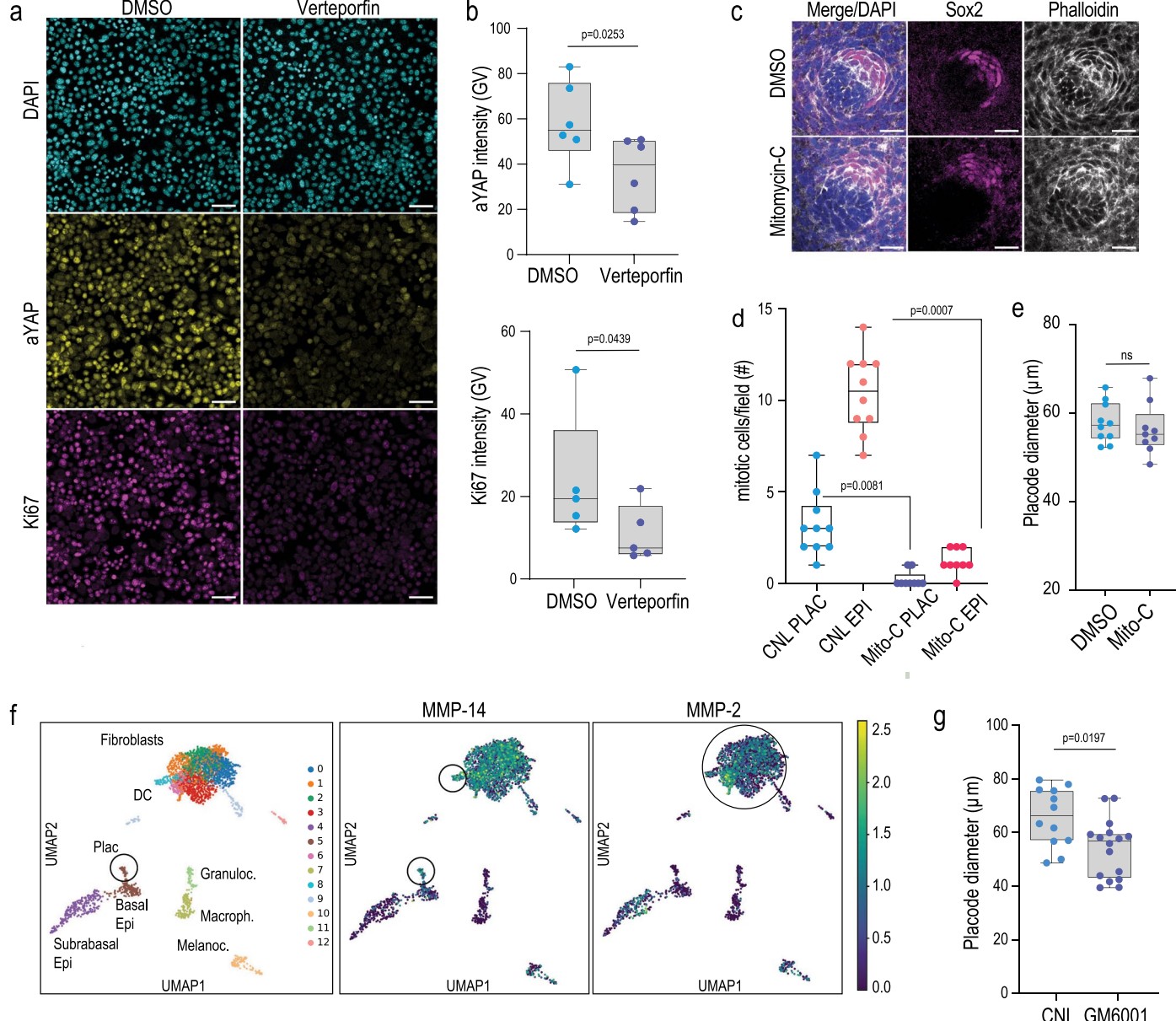

**Extended Data Fig. 6 | Yap-dependent regulation of cell proliferation and basement membrane remodeling is required for placode development.**
(**a, b**) Representative images (a) and quantification (b) of active YAP (aYAP; top) and proliferation (Ki67; bottom) in isolated epidermal stem cells treated with YAP inhibitor Verteporfin (10 μM, 6 h) (n = 6 independent experiments, paired *t*-test). (**c**) Top views of Phalloidin/Sox2-stained Mito-C- and DMSO-treated control explants isolated from E14.5 mice. Images representative of 3 mice/group. (**d**) Quantification of mitotic cells in placodes (PLAC) and interfollicular epidermis (EPI) of skin explants from E14.5 mice treated Mito-C. Note reduced mitoses in Mito-C-treated explants (n = 10 (CNL), 9 (Mito-C) placodes (PLAC)

or surrounding epidermis (EPI) from 3 mice/group; Kruskal-Wallis/Dunn's). (**e**) Quantification of placode diameter in explants from E14.5 mice treated Mito-C (n = 10 (CNL), 9 (Mito-C) placodes from 3 mice/group; Student's *t*-test). (**f**) UMAP projections of single cell RNA seq data from E14.5 skin. Note MMP-14 expression in placode (black circle) and fibroblast cells and MMP-2 expression in fibroblasts. (**g**) Quantification of images shown in (Fig. 6f) shows reduced placode diameter in GM6001-treated explants (n = 12 (CNL) and 16 (GM6001) placodes from 3 explants/condition; Mann-Whitney). Minimum-to-maximum box plots show the 75th, 50th, 25th percentiles. Scale bars 50 μm.

# Reporting Summary

## Statistics

For all statistical analyses, confirm that the following items are present in the figure legend, table legend, main text, or Methods section.

| n/a | Confirmed | |
|---|---|---|
| ☐ | ☒ | The exact sample size (*n*) for each experimental group/condition, given as a discrete number and unit of measurement |
| ☐ | ☒ | A statement on whether measurements were taken from distinct samples or whether the same sample was measured repeatedly |
| ☐ | ☒ | The statistical test(s) used AND whether they are one- or two-sided *Only common tests should be described solely by name; describe more complex techniques in the Methods section.* |
| ☒ | ☐ | A description of all covariates tested |
| ☐ | ☒ | A description of any assumptions or corrections, such as tests of normality and adjustment for multiple comparisons |
| ☐ | ☒ | A full description of the statistical parameters including central tendency (e.g. means) or other basic estimates (e.g. regression coefficient) AND variation (e.g. standard deviation) or associated estimates of uncertainty (e.g. confidence intervals) |
| ☐ | ☒ | For null hypothesis testing, the test statistic (e.g. *F*, *t*, *r*) with confidence intervals, effect sizes, degrees of freedom and *P* value noted *Give P values as exact values whenever suitable.* |
| ☒ | ☐ | For Bayesian analysis, information on the choice of priors and Markov chain Monte Carlo settings |
| ☒ | ☐ | For hierarchical and complex designs, identification of the appropriate level for tests and full reporting of outcomes |
| ☒ | ☐ | Estimates of effect sizes (e.g. Cohen's *d*, Pearson's *r*), indicating how they were calculated |

*Our web collection on statistics for biologists contains articles on many of the points above.*

## Software and code

Policy information about availability of computer code

| Data collection | Leica Application Suite X (confocal microscopy, version 2.0.0.14332)<br>Zeiss Zen software (2.3 SPI)<br>Zeiss ZEN software (blue 3.5)<br>Andor Fusion software (spinning disc confocal microscopy, version 2.3.0.44 )<br>JPK SPM Control Software (version 5)<br>Nikon Software (NIS-Elements AR 5.41.01) |
|---|---|

| Data analysis | JPK Data Processing Software (Bruker Nano, version 5)<br>GraphPad Prism software (GraphPad, version 8 and 9)<br>Fiji (version 2.0.0)<br>FastQC (v0.11.9)<br>STARsolo aligner (v2.7.7a)<br>MultiQC (v1.9)<br>gseapy (v0.10.5)<br>PIVLab (v2.59)<br>Python (v3.9)<br>Tissue Analyzer (v2.3 beta2)<br>Mathematica (v12.3)<br>Cellpose (v2.0)<br>Orientation J(v2.0.5) from Fiji |
|---|---|

For manuscripts utilizing custom algorithms or software that are central to the research but not yet described in published literature, software must be made available to editors and reviewers. We strongly encourage code deposition in a community repository (e.g. GitHub). See the Nature Portfolio guidelines for submitting code & software for further information.

## Data

Policy information about availability of data

All manuscripts must include a data availability statement. This statement should provide the following information, where applicable:

- Accession codes, unique identifiers, or web links for publicly available datasets
- A description of any restrictions on data availability
- For clinical datasets or third party data, please ensure that the statement adheres to our policy

Previously published sequencing data that were re-analysed here are available under accession code GSE122043.
The vertex model is available under the following link:  https://github.com/ZhangTao-SJTU/tvm
Custom-built Python and Mathemathica scripts used in the manuscript are available under the following link:https://github.com/WickstromLab
All other data supporting the findings of this study are available from the corresponding author on reasonable request.

## Human research participants

Policy information about studies involving human research participants and Sex and Gender in Research.

| Reporting on sex and gender | N/A |
|---|---|
| Population characteristics | N/A |
| Recruitment | N/A |
| Ethics oversight | N/A |

Note that full information on the approval of the study protocol must also be provided in the manuscript.

# Field-specific reporting

Please select the one below that is the best fit for your research. If you are not sure, read the appropriate sections before making your selection.

☒ Life sciences          ☐ Behavioural & social sciences          ☐ Ecological, evolutionary & environmental sciences

For a reference copy of the document with all sections, see nature.com/documents/nr-reporting-summary-flat.pdf

# Life sciences study design

All studies must disclose on these points even when the disclosure is negative.

| Sample size | Sample size was determined based on previous experience, published literature ( PMID: 30063206, 28700594, 29662173) . Sample size for each experiment is indicated in figure legends. |
|---|---|
| Data exclusions | For AFM measurements outlier identification was carried out to remove rare individual measurements that represented apparent artefacts, no other data points were removed from the experiments |
| Replication | All experiments were performed using at least three biological replicates. Number of replicates for each experiment is indicated in the corresponding figure legend. Several steps were taken to ensure the reproducibility of experimental findings and key results were confirmed using complementary experimental approaches. |

| Randomization | Samples were not randomized, randomization was not relevant as samples were grouped according to genotype or treatment. |
| --- | --- |
| Blinding | Blinding was used where relevant (AFM analysis). It was not relevant in the in vivo studies as phenotypes of mice were clear to experimenters. Whenever possible automated software algorithms were used for unbiased quantification of cell volumes (Fig. 1g-h), cellular and tissue morphology (Extended Data Figure 1b-c; Extended Data Figure 2a, b), PIV analysis and strain rate extraction (Fig.2a-d and i; Extended Data Figure 2c-e, h, i) staining intensities (Fig. 2e-g; Fig.4f, g, j-m; Fig. 6b, c; Extended Data Figure 2f,g; Extended Data Figure 4b-d; Extended Data Figure 5h,i o, p; Extended Data Figure 6a,b) , fiber alignment (Fig.2h) and sequencing data (Extended Data Figure 6f). |

# Reporting for specific materials, systems and methods

We require information from authors about some types of materials, experimental systems and methods used in many studies. Here, indicate whether each material, system or method listed is relevant to your study. If you are not sure if a list item applies to your research, read the appropriate section before selecting a response.

### Materials & experimental systems

| n/a | Involved in the study |
| --- | --- |
| ☐ | ☒ Antibodies |
| ☒ | ☐ Eukaryotic cell lines |
| ☒ | ☐ Palaeontology and archaeology |
| ☐ | ☒ Animals and other organisms |
| ☒ | ☐ Clinical data |
| ☒ | ☐ Dual use research of concern |

### Methods

| n/a | Involved in the study |
| --- | --- |
| ☒ | ☐ ChIP-seq |
| ☒ | ☐ Flow cytometry |
| ☒ | ☐ MRI-based neuroimaging |

## Antibodies

| Antibodies used | goat anti Edar (R&D Systems AF745; 1:200), rabbit anti-p-MLC2 (p-Ser20; Abcam 2480; 1:100; 1:100), rabbit anti-Vimentin (Abcam 185030; 1:100), rabbit anti-Sox9 (Cell Signalling, 82630; 1:100), rabbit anti-Sox2 (Millipore Sigma, AB5603; 1:200), mouse anti-Twist2 (Abcam, 50887; 1:200), mouse anti-Ki67 (Cell Signaling, 9449; 1:300), rabbit anti-active YAP1 (Abcam, 205270; 1:100), guinea-pig anti-Keratin 14 (Progen, GP-CK14; 1:300), rabbit anti-Collagen IV (Abcam, 6586; 1:200), rabbit anti-Laminin 332 (gift from R.E Burgeson; 43; 1:20 000), rat anti-Laminin a5 (504 44; self-produced; gift from L. Sorokin; 1:20 000), chicken anti-beta-galactosidase (Abcam, 9361; 1:500). F-actin was detected using Alexa647-conjugated phalloidin (1:500). The following secondaries (all from Invitrogen): anti-rabbit IgG Alexa Fluor 488 / 568 (A11008 / A10042; 1/500); anti-mouse IgG1 / IgG2A Alexa Fluor 546 (A21123 / A21134; 1/500); anti-rat IgG-rat Alexa Fluor 647 (A21247; 1/500); anti-goat IgG Alexa Fluor 488 (A21467) and anti-chicken IgG Alexa Fluor 488 (A21449; 1/500) were used to visualize primary antibodies. |
| --- | --- |
| Validation | All antibodies are well characterized and widely used in the literature. They were applied according to datasheet instructions or previously published protocols. Antibodies were additionally validated as follows:<br>- Inhibitor studies followed by immunofluorescence on primary mouse keratinocytes (YAP, Extended Data Figure 6a, b) or immunofluorescence on mouse tissues  (Ki67, Extended Data Figure 6c, d), or zebrafish tissues (pMLC2; PMID: 33208950).<br>- Immunofluorescence of mouse tissue from knockout or transgenic mouse models (Laminin alpha5 (PMID: 27234307), beta galactosidase (Extended Data Figure 1a, b and PMID 30063206)).<br>- Immunofluorescence on mouse tissue followed by observing expected pattern of histological staining (as described in the literature for example PMID: 26256211, PMC6361530 and 30887615) and colocalization with additional markers of same state or process or specific morphology: (Keratin 14, localization in the basal layer of the epidermis, Extended Data Figure 4h ; Twist2 and Vimentin, specific expression in fibroblasts, Extended Data Figure 5a; Collagen IV and Laminin 332, localisation specifically at the basement membrane Fig.6b;  Sox9, localisation specifically  in placode cells, Fig.4f and PMID29662173;   Sox2, localisation specifically  in the dermal condensate, Extended Data Figures 4e and 5c and PMID 30063206; Edar, enrichment specifically  in placode cells, Fig.1b and Fig.2h and PMID: 30887615).<br>. |

## Animals and other research organisms

Policy information about <u>studies involving animals</u>; <u>ARRIVE guidelines</u> recommended for reporting animal research, and <u>Sex and Gender in Research</u>

| Laboratory animals | Wild-type C57Bl6J mice, Myh9 floxed mice (obtained from the European Mouse Mutant Archive; EM:02572) were crossed with the K14-Cre line (Hafner et al.,  or the Twist2-Cre line obtained from JAX laboratories (stock #008712). Membrane-targeted Tomato reporter mice (R26RmT/mG) were from JAX laboratories (stock #007676). Histone2B -mCherry/membrane-EGFP (R26R-RG mice; were from Riken Laboratories for Animal Resources and Genetic Engineering (LARGE) and were crossed with K14-Cre line to obtain epidermis-specific expression. FGF20-LacZ reporter mice have been previously described in Huh et al., Genes Dev 2013. |
| --- | --- |
| Wild animals | Study did not involve wild animals |
| Reporting on sex | Studies were carried out on embryos of both sexes as no sex-specific difference relevant for the processes studied here were noted. |

| Field-collected samples | Study did not involve samples collected from the field |
|---|---|
| Ethics oversight | All mouse studies were approved and carried out in accordance with the guidelines of the Finnish national animal experimentation board (ELLA) or the Ministry for Environment, Agriculture, Conservation and Consumer Protection of the State of North Rhine-Westphalia (LANUW), Germany. |

Note that full information on the approval of the study protocol must also be provided in the manuscript.

