## [Peer Review File · Nature Cell Biology]

Peer Review Information

Journal: Nature Cell Biology

Manuscript Title: Mechanical forces across compartments coordinate cell shape and fate transitions to generate tissue architecture

Corresponding author name(s): Professor Sara Wickström

Editorial Notes:

Reviewer Comments & Decisions:

Decision Letter, initial version:
--

*Please delete the link to your author homepage if you wish to forward this email to co-authors.

Dear Professor Wickström, dear Sara

I hope you are well, and I'm sorry once again for the delay.

Your manuscript, "Mechanical forces across compartments coordinate cell shape and fate transitions to generate tissue architecture", has now been seen by 3 referees, who are experts in mechanobiology (referee 1); hair follicles (referee 2); and hair follicle and scRNAseq (referee 3). As you will see from their comments (attached below) they find this work of potential interest, but have raised substantial concerns, which in our view would need to be addressed with considerable revisions before we can consider publication in Nature Cell Biology.

Nature Cell Biology editors discuss the referee reports in detail within the editorial team, including the chief editor, to identify key referee points that should be addressed with priority, and requests that are overruled as being beyond the scope of the current study. To guide the scope of the revisions, I have listed these points below. We are committed to providing a fair and constructive peer-review process, so please feel free to contact me if you would like to discuss any of the referee comments further.

I should stress that the referees' concerns point to unclear causal and mechanistic links which would need to be addressed with experiments and data, and reconsideration of the study for this journal and re-engagement of referees would depend on strength of these revisions.

In particular, it would be essential to:

A) Experimentally assess the contribution of forces from different tissues and orientations (all Reviewers, see particularly comments from Reviewer #1)

B) Provide further data on the effects of myosin perturbations and mechanical perturbations on Sox9 and Sox2 expression (all Reviewers)

C) Generate and experimentally test hypotheses as predicted by the theoretical modelling (Reviewers #1 and #3)

D) All other referee concerns pertaining to strengthening existing data, providing controls, methodological details, clarifications and textual changes, should also be addressed.

E) Finally please pay close attention to our guidelines on statistical and methodological reporting (listed below) as failure to do so may delay the reconsideration of the revised manuscript. In particular please provide:

We would be happy to consider a revised manuscript that would satisfactorily address these points, unless a similar paper is published elsewhere, or is accepted for publication in Nature Cell Biology in the meantime.

- ensure that it conforms to our format instructions and publication policies (see below and <https://www.nature.com/nature/for-authors>).

- provide a point-by-point rebuttal to the full referee reports verbatim, as provided at the end of this letter.

- provide the completed Reporting Summary (found here <https://www.nature.com/documents/nr-reporting-summary.pdf>). This is essential for reconsideration of the manuscript will be available to editors and referees in the event of peer review. For more information see <http://www.nature.com/authors/policies/availability.html> or contact me.

When submitting the revised version of your manuscript, please pay close attention to our [href="https://www.nature.com/nature-portfolio/editorial-policies/image-integrity">Digital Image Integrity Guidelines](https://www.nature.com/nature-portfolio/editorial-policies/image-integrity). and to the following points below:

- that unprocessed scans are clearly labelled and match the gels and western blots presented in figures.
- that control panels for gels and western blots are appropriately described as loading on sample

processing controls

-- all images in the paper are checked for duplication of panels and for splicing of gel lanes.

Nature Cell Biology is committed to improving transparency in authorship. As part of our efforts in this direction, we are now requesting that all authors identified as 'corresponding author' on published papers create and link their Open Researcher and Contributor Identifier (ORCID) with their account on the Manuscript Tracking System (MTS), prior to acceptance. ORCID helps the scientific community achieve unambiguous attribution of all scholarly contributions. You can create and link your ORCID from the home page of the MTS by clicking on 'Modify my Springer Nature account'. For more information please visit www.springernature.com/orcid.

This journal strongly supports public availability of data. Please place the data used in your paper into a public data repository, or alternatively, present the data as Supplementary Information. If data can only be shared on request, please explain why in your Data Availability Statement, and also in the correspondence with your editor. Please note that for some data types, deposition in a public repository is mandatory - more information on our data deposition policies and available repositories appears below.

[Redacted]

We would like to receive a revised submission within six months.

We hope that you will find our referees' comments, and editorial guidance helpful. Please do not hesitate to contact me if there is anything you would like to discuss.

Best wishes,

Daryl

Daryl J.V. David, PhD

Senior Editor, Nature Cell Biology
Nature Portfolio

Heidelberger Platz 3, 14197 Berlin, Germany
Email: daryl.david@nature.com
ORCID: <https://orcid.org/0000-0002-9253-4805>

Reviewers' Comments:

Reviewer #1:

Remarks to the Author:

In this manuscript, Villeneuve et al., study the biophysical mechanisms of hair follicle placode formation and its coordination with Sox9 expression. The authors initially characterize the morphological changes in the tissue during placode formation. After that, they attempt to understand how these morphological changes occur from a biophysical perspective. They suggest that apical/basal changes in myosin in placode cells partially explain their morphological changes, and they claim that a fibroblast ring generates radial stresses that drive placode formation and compartmentalization of Sox9. After that, the authors study the role of cell divisions and claim that tissue confinement explains the observed spatiotemporal changes in divisions in the tissue, as well as morphological changes in placode formation. Finally, they study the role of basement membrane mechanical changes in placode formation. They conclude that a combination of all the above-mentioned effects drives the formation of the placode and Sox9 expression.

The manuscript studies an important question in embryogenesis, namely the relation between tissue dynamics, biophysics and cell state transitions. While some of the experiments are very interesting, a most of the author's claims that are fundamental for the proposed mechanism are not supported by the data provided. Specifically, all the first part of the study, in which they claim that there are centripetal oscillations and a fibroblast ring that applies contractile radial forces is largely unsupported by the data provided. Beyond the biophysical mechanism, the authors' claims relating the biophysical mechanism and Sox9 expression also lack support. The theoretical model is too basic and, in my opinion, is disproved by the authors themselves with their beautiful experiments on cell division patterns and basement membrane mechanics. In contrast to the first part, the experiments on cell division patterns and basement membrane are very interesting. While the conclusions of these experiments are not fully justified either (it is unclear how some experiments were done; see comments below), the observed phenotypes upon perturbation of cell divisions and MMPs are strong and clear, suggesting that these processes are the important ones (rather than myosin forces or potential fibroblasts rings).

Overall, I think the manuscript touches on an interesting topic, but many of the claims regarding the proposed mechanisms are not supported by the data provided at this point. Therefore, I cannot recommend publication of the manuscript in this form.

General comments:

- Authors claim the existence of oscillatory deformations driving tissue elongation (L141). The data provided to support these claims is very unclear though. Fig. 2B seems to show the divergence going above and below zero, but it is unclear whether these are fluctuations or actual oscillations. To claim the existence of oscillations, it is necessary to provide a clear measure of them, such as peak in a Fourier analysis of the signal or similar. Moreover, oscillations are characterized by multiple cycles, which I failed to see in the data. The only thing I can see in the Supplementary movies and in the analysis is a single contraction in the placode, but not oscillatory behavior. Perhaps there are repeated contractions, but this is not apparent in the data. Can the authors please provide a proof of such oscillatory behavior?

- The laser ablation experiments around the placode (Fig. 2D) indicate that the tissue is under tension. The recoil and radial displacement observed are precisely the same than previously observed in

circular ablations of a completely different epithelial tissue that had not placodes or any specific structure. It is true that the placode seems to be under tension. However, is there any difference between this observation around the placode and in any other location in the tissue? It seems that the authors are suggesting that the placode being under tension is somehow a specific feature of the mechanics of placode formation. I am not convinced that this is the case. Can the authors please perform circular ablations away from the placode and see if the mechanics is different in any way? It would also help to have data for circular cuts of different radii both inside and outside the placode, as these could tell if there are really differences in tension inside and outside the placode.

- L171: The authors claim that there is a "dense ring of fibroblasts" surrounding the placode. Unfortunately, I failed to see this in the data provided (Fig. 2I). In Fig. 2I, I see that there is no vimentin in the placode and that there is vimentin around, with the fibers oriented along the border of the placode. This is very interesting. However, I do not see the fibroblasts in any image or a quantification of their density or ring formation in any way. Can the authors please quantify the density and alignment of fibroblasts around the placode to verify if there is a "dense ring of fibroblasts"?

- Following on the previous comment, in L178-179, the authors claim "a ring of contractile fibroblasts developed to tightly wrap around the placode base, coinciding with the centripetal oscillations in this area". I do not see almost any data supporting such a strong claim. Since this is one of the main claims of the manuscript and central to the proposed mechanism, the authors should provide strong evidence for it. Currently, there is no data proving that there is a ring of fibroblasts, let alone that they are contractile along a ring surrounding the placode, or that there exist oscillations (as mentioned above). It seems that the authors believe that a ring of fibroblasts wraps around the placode and contracts, driving a radial contraction. It is a very nice hypothesis, but the data does not support many parts of it as the paper stands now. I suggest that the authors label the fibroblasts (immunostaining should be enough, but live imaging would provide more information) and check that there is indeed a higher concentration of them around the placode and forming a ring. I suggest also that the authors perform radially-oriented laser ablation cuts in the region of the dermis where there presumably is a contractile ring, as well as radially-oriented cuts in the dermis further away from the placode. These experiments would directly test the authors hypothesis that there is a fibroblast ring that is contractile, thereby generating an inward (radial) force on the placode.

- Following in my previous point, the circular vimentin arrangement around the placode could arise from the expansion forces generated as the placode invaginates into the dermis, providing a passive resistance rather than being the driver of a contraction by fibroblasts. Looking at all the data provided, I think this model is equally plausible at this point and the one the authors suggest.

- L181: The authors propose a 3D vertex model of the tissue with multiple layers. The model predicts that a high basal to apical wetting coefficient could explain some of the observed shape changes, but the radial forces are also needed to predict the observed tissue divergence. However, the experimental data (Fig. 2F,G) does not show a large change in myosin between basal and apical cell surfaces in the placode, so it is unclear to me how this observation fits with the predictions of the model. More generally, I am not sure how the model is helping understand the data. It is unclear that this model (either 1 or 2) represents what is really happening – there are many other qualitatively different models that could also explain the observed data. Theoretical models are useful when, in addition to roughly fitting some data, provide a clear prediction that can be then tested experimentally. Can the authors generate a prediction (for instance, if the wetting coefficient changes, then the cell shapes change in a particular way) and test it experimentally? It would be useful to make a prediction that could specifically differentiate models 1 and 2 and then test it. However, I think it is important to consider the possibility that neither of these models properly accounts for the experimental system, as it does not incorporate in any details of the dermis, it is unclear if the

mechanics of the real boundaries is the one assumed in the model, etc. For instance, there could be a localized softening of the basement membrane, as observed in lung branching morphogenesis or in tooth formation, that could explain the localized thickening of the placode.

- The authors report that in myosin knockout mice, the placode is less deep than in controls (Fig. 4B), but that the shape of cells is not different in controls and in myosin knockout mice (Fig. 4c). How is this possible? The authors seem to conclude from the previously presented data and the simulations, that changes in myosin (wetting coefficient) would change cell shapes, which in turn would cause changes in the placode depth. It is unclear to me how to make these observations consistent with each other. Can the authors please explain it?

- The myosin inhibition experiments in fibroblasts, as well as global chemical inhibition with blebbistatin are very interesting, and also puzzling. Specific inhibition of Myh9 in fibroblasts is a natural test for some of the predictions of the model 2, as the authors point out. There seems to be some changes associated with the loss of Myh9 in fibroblasts, but the perturbation cannot stop placode formation or Sox9 expression, indicating that likely there is some other mechanical driver of placode formation. Can the authors perform radially-oriented laser ablation cuts in the observed vimentin ring to directly remove tension in case there is some there? The global chemical inhibition of myosin with blebbistatin is very interesting because it is the perturbation that leads to a stronger phenotype, with placodes not forming. The fact that this inhibition really perturbs placode formation suggests that other myosin-dependent processes not considered in the model can be causing the forces behind placode formation. Myosin inhibition makes Sox9 expression more diffuse in the tissue, but cells keep expressing Sox9. Since in the presence of myosin the placode cells seem to converge and define a higher density region (Fig. 4I), the compartmentalization of Sox9 seems to be more associated with the changes in placode morphogenesis than with any effect of force on Sox9 expression. Can the authors please check Sox9 expression after performing laser ablations to release tension in the tissue?

- Following up on the previous point, the authors claim that Sox9 increases upon application of pressure to epidermal progenitor cells in culture conditions. While there seems to be an effect in Fig. 4N, the quantification in Fig. 4O shows a very small change in expression. Because of the small change, I am not convinced by these experiments. However, performing laser cuts in the tissues and measuring Sox9 expression levels should give a clear answer to this question.

- The observations presented on the pattern of cell divisions are very interesting. The complete switch in the proliferation pattern between E14.5 and E15.5 is remarkable. I think the fact that the authors saw a near completely disruption of placode formation in the presence of blebbistatin is not because of the mechanism they proposed, but rather because blebbistatin is affecting cell division. Can the authors measure the cell division pattern in the presence of blebbistatin to test this possibility?

- The experiments to determine spatial changes in the mechanics of the basement membrane are also very interesting. However, I do not understand the AFM measurements. It is notoriously difficult to measure the mechanics of basement membranes *in vivo*. Did the authors isolate basement membranes or the entire tissue? If they measure the stiffness of the tissue, including the cells, the measurements do not reflect changes in the mechanics of the basement membrane because the sample is only indented by 0.5 microns, which would test the mechanics of the cells in contact with the AFM bead, but not the basement membrane. It is unclear from the methods section if the authors used isolated basement membranes and how such experiments were done. Can the authors please provide more details on these experiments? What was the size of the bead used for indentation?

- The experiments inhibiting MMPs are beautiful. The observed phenotype upon chemical inhibition of MMPs shows that clearly the ECM (perhaps not only basement membrane) plays an important role in

placode formation.

Specific comments:

- The sketch in Figure 1A should be improved to include many of the terms and cell types that are described in the text. For instance, the mesenchymal cells are missing in the sketch and many steps of cell state changes are not described. I would strongly suggest to make a sketch for the different steps that defines the different cell states and main morphological features.

- The term "isotropic elongation" (line 107) is rather unusual. Elongation is anisotropic, so it is quite unclear what is meant by isotropic elongation. Perhaps the authors can find a different term?

- It seems quite difficult to identify FGF20 positive cells from the images in Fig. 1B, especially at E13.5. Can the authors please provide supplementary figures showing the good representative images and how they are analyzed? Also, the definition of cell geometry with basal and apical lengths can obscure the results because cell lengths on both apical and basal polygonal cell shapes are different in different directions by about 30% or so. Perhaps a proper definition of what length is measured would help identify relevant morphological changes.

- FGF20 positive cells at E14.4 and E15.5 (Fig. 1b) look morphologically very different to me, but the morphological quantification in Fig. 1D shows essentially no significant differences in basal and apical length between these two stages. Why? Can the authors please benchmark their analysis to be sure that the quantification is adequate? Can they provide the error associated with these measurements (not the ensemble error of different replicates, but rather the error in the segmentation and shape analysis) to be sure that the quantification is meaningful?

Reviewer #2:

Remarks to the Author:

Authors demonstrated (i) an elongation in cell morphology between E13.5-E15.5 (i.e., when Sox9 expression is observed in future HFSCs) and (ii) contractile oscillations around placode neck on E14 (placode stage) being driven by a ring of fibroblasts. The authors laid out the argument that this fibroblast-driven process is mediated by myosin IIA. Furthermore, Sox9 expression was more robust w/ increased compression. The ideas are novel, and the experiments seem appropriate. However, there are some major issues that may be attributing to significant biases in the study.

1. Lack of an objective placode marker:

Fgf20 β Gal embryos were used to generate the data regarding placode cell morphology. Subsequent live imaging experiments utilized membrane-targeted Tomato reporter mice. Based on the figures in the paper, it appears that placodes were identified based on morphology. As a result, the lines drawn to demarcate the placode seem quite heterogeneous with some lines containing too much and some containing too little. This represents a significant source of bias for the measurement of placode depth and length. I recommend that the authors provide data using immunostaining for FGF20 for at least some of these experiments to validate their findings.

2. Myosin IIA data:

Page 7, line 232 describes myosin IIA as "the major myosin isoform responsible for keratinocyte actomyosin contractility". The authors claim that myosin IIA is essential for contraction by the ring of fibroblasts.

Was there a predetermined fluorescence threshold used by the authors for the vimentin staining

analysis? The ring width of vimentin for Figure 2d looks almost identical between the control group and the Myh9-dKO group (maybe just a little blurrier in the Myh9 group). However, the arrows are smaller for the dKO group. Again, the methods need to be clarified or the data needs to be analyzed in a more objective manner.

Furthermore, there is a lot of functional redundancy between NMII isoforms, especially NMIIA and IIB. This is supported by the lower # of placodes in the blebbistatin-treated group (inhibition of NMIIA and IIB) vs. Myh9-dKO group (inhibition of NMIIA only). It also looks like placodes still form w/ slightly reduced invagination. The claims regarding the "essential" role of myosin IIA either needs to be attenuated or be revised to state that it is possible that the contractions are being mediated by other NMII isoforms.

3. AFM data:

AFM was performed to measure the stiffness of the basement membrane. Prior studies that performed a similar experiment removed the surrounding cells in a systematic manner (J Struct Biol. 2009 Jul;167(1):19-24.) to ensure that the measurements were taken accurately. Was this performed? If not, how can you be certain that measurements were not taken from surrounding tissue?

Some additional questions:

Initially there was more cell proliferation in the surrounding epidermis than in the placodes. Is it possible that the expansion of the surrounding epidermis helped "push" the less proliferative placodes downwards?

If Sox9 expression is dependent on mechanical tension, then why does the blebbistatin treated tissue still have high levels of Sox9 expression?

Any thoughts on what caused the initial difference in apical/basal cell properties in the very early stage of placode formation?

Neither Myh9-eKO or dKO abolished the initial placode pattern formation (the KO mostly just reduced the placode down-growth, or maybe reduced the density of the placodes). So, which myosin component was involved in the initial pattern formation, or compensated for Myh9 after its deletion?

Minor edits:

1. For the "side view" photos, a more widely cropped or a more zoomed out image would be more appropriate. For example, in Figure 1b, it is difficult to appreciate the cell morphology for the top panel (E13.5) since the top portions of the cells are not visible. It is also impossible to see the dermis in these images.

Reviewer #3:

Remarks to the Author:

In this study, Villeneuve et al. aims to characterize the mechanical forces and morphogenetic behaviors that placode cells encounter during embryonic hair follicle development. Specifically, the authors used biophysical approaches to look at placode stage (E14.5) and hair germ stage (E15.5) and mapped out a two-step process of cellular elongation followed by downgrowth of the budding placode. They concluded that both cell-intrinsic contractility and extrinsic forces from the dermis contributed to the morphological changes in the placode. With a combination of genetic perturbation experiments and drug treatment, they disrupted the contractility in the epidermal and dermal compartment and showed both resulted in defects in placode formation and/or invagination. Mechanistically, they linked placode invagination at E15.5 with changes in extrinsic mechanical stress

and resultant cell cycle reentry experienced by placode cells. They further attributed the release of mechanical force to the softening of the basement membrane during placode invagination. The biophysical experiments are methodical and provided new insights into the biomechanical forces that regulate epithelial morphogenesis. However, the manuscript can be made stronger if the authors could reconcile their findings and models with previous knowledge in the hair follicle biology field.

Major comments:

1. It appears that the definition and the biological meaning of a 'fibroblast ring' is not very clear. The vimentin-rich status in the fibroblast ring does not necessarily mean that those fibroblasts exert extrinsic force on the placode that causes the placode invagination. It could be that the placode or other nearby epidermal cells exert forces on the cells in the fibroblast ring as it invaginates down. For example, it could be that epidermal cells neighboring the placode push the placode inward and downward and then the placode pushes on the fibroblasts, which would be in line with their laser ablation experiment. Therefore, the extrinsic force within the authors' model could also come from neighboring epidermal cells. Additionally, within their model, the authors could also take into account the potential forces exerted by the DC, which lies beneath the placode. The authors should incorporate this into their 3D vertex modeling. In this study, the center of the placode is assumed to not experience any force, even though it is in contact with the DC.
2. The authors showed that Sox2+ DCs exist in the Myh9-eKO, which suggested that the defect in placode invagination mostly resulted from loss of contractility from the epidermis instead of other defects (e.g. loss of DC). The authors do not show the presence or absence of Sox2 DCs in the Myh9-dKO mutant (Figure 4a), the Blebbistatin-treated explants (Figure 4j), or the MMP-inhibitor-treated explants (Figure 5h). It would be helpful if the authors could also show the DC status in these experiments to show that the defects are not due to potential effects on other dermal populations (e.g. DC) that led to absence of HF induction.
3. The authors concluded that E15.5 placodes undergo downgrowth due to increased nuclear YAP signaling and the resulting cell cycle reentry in basal placode cells, suggested by Figure 5b/c. 1) what is the DC status in Mitomycin-C treated explants vs control? 2) Can the authors comment on where the proliferation and YAP signaling occurs in the hair germ? For example Ouspenskaia et al. showed that Wnt-high basal cells are slow-cycling, so it would be surprising to see similar proliferation in the basal cells and suprabasal cells of the hair germ. Also, the authors could incorporate work published by Zhang et al 2010 PNAS, which examined the role of YAP in embryonic epithelial/hair follicle development.
4. Although this paper is timely for the field, many of the findings are largely descriptive in nature. Although possibly beyond the scope of this paper, it would be nice if there were more in vivo functional experiments to test their model.

Minor comments:

1. What was the control in Figure 4n? Was the control no compression at 0h, 3h or 16h?

READABILITY OF MANUSCRIPTS – Nature Cell Biology is read by cell biologists from diverse

backgrounds, many of whom are not native English speakers. Authors should aim to communicate their findings clearly, explaining technical jargon that might be unfamiliar to non-specialists, and avoiding non-standard abbreviations. Titles and abstracts should concisely communicate the main findings of the study, and the background, rationale, results and conclusions should be clearly explained in the manuscript in a manner accessible to a broad cell biology audience. Nature Cell Biology uses British spelling.

Methods should be written concisely, but should contain all elements necessary to allow interpretation and replication of the results. As a guideline, Methods sections typically do not exceed 3,000 words. The Methods should be divided into subsections listing reagents and techniques. When citing previous methods, accurate references should be provided and any alterations should be noted. Information must be provided about: antibody dilutions, company names, catalogue numbers and clone numbers for monoclonal antibodies; sequences of RNAi and cDNA probes/primers or company names and catalogue numbers if reagents are commercial; cell line names, sources and information on cell line identity and authentication. Animal studies and experiments involving human subjects must be reported in detail, identifying the committees approving the protocols. For studies involving human subjects/samples, a statement must be included confirming that informed consent was obtained. Statistical analyses and information on the reproducibility of experimental results should be provided in a section titled "Statistics and Reproducibility".

All Nature Cell Biology manuscripts submitted on or after March 21 2016 must include a Data availability statement as a separate section after Methods but before references, under the heading "Data Availability". For Springer Nature policies on data availability see <http://www.nature.com/authors/policies/availability.html>; for more information on this particular policy see <http://www.nature.com/authors/policies/data/data-availability-statements-data-citations.pdf>. The Data availability statement should include:

- Accession codes for primary datasets (generated during the study under consideration and designated as "primary accessions") and secondary datasets (published datasets reanalysed during the study under consideration, designated as "referenced accessions"). For primary accessions data should be made public to coincide with publication of the manuscript. A list of data types for which submission to community-endorsed public repositories is mandated (including sequence, structure, microarray, deep sequencing data) can be found here <http://www.nature.com/authors/policies/availability.html#data>.
- Unique identifiers (accession codes, DOIs or other unique persistent identifier) and hyperlinks for datasets deposited in an approved repository, but for which data deposition is not mandated (see here for details <http://www.nature.com/sdata/data-policies/repositories>).
- At a minimum, please include a statement confirming that all relevant data are available from the authors, and/or are included with the manuscript (e.g. as source data or supplementary information), listing which data are included (e.g. by figure panels and data types) and mentioning any restrictions on availability.
- If a dataset has a Digital Object Identifier (DOI) as its unique identifier, we strongly encourage including this in the Reference list and citing the dataset in the Methods.

We recommend that you upload the step-by-step protocols used in this manuscript to the Protocol Exchange. More details can be found at www.nature.com/protocolexchange/about.

All imaging data should be accompanied by scale bars, which should be defined in the legend. Cropped images of gels/blots are acceptable, but need to be accompanied by size markers, and to retain visible background signal within the linear range (i.e. should not be saturated). The boundaries of panels with low background have to be demarked with black lines. Splicing of panels should only be considered if unavoidable, and must be clearly marked on the figure, and noted in the legend with a statement on whether the samples were obtained and processed simultaneously. Quantitative comparisons between samples on different gels/blots are discouraged; if this is unavoidable, it should only be performed for samples derived from the same experiment with gels/blots were processed in parallel, which needs to be stated in the legend.

The total number of Supplementary Figures (not including the “unprocessed scans” Supplementary Figure) should not exceed the number of main display items (figures and/or tables (see our Guide to Authors and March 2012 editorial <http://www.nature.com/ncb/authors/submit/index.html#suppinfo>; <http://www.nature.com/ncb/journal/v14/n3/index.html#ed>). No restrictions apply to Supplementary Tables or Videos, but we advise authors to be selective in including supplemental data.

GUIDELINES FOR EXPERIMENTAL AND STATISTICAL REPORTING

REPORTING REQUIREMENTS – We are trying to improve the quality of methods and statistics

reporting in our papers. To that end, we are now asking authors to complete a reporting summary that collects information on experimental design and reagents. The Reporting Summary can be found here <https://www.nature.com/documents/nr-reporting-summary.pdf> If you would like to reference the guidance text as you complete the template, please access these flattened versions at <http://www.nature.com/authors/policies/availability.html>.

Author Rebuttal to Initial comments

Point-by-point response to reviewer comments

Reviewer #1:

The manuscript studies an important question in embryogenesis, namely the relation between tissue dynamics, biophysics and cell state transitions. While some of the experiments are very interesting, a most of the author's claims that are fundamental for the proposed mechanism are not supported by the data provided. Specifically, all the first part of the study, in which they claim that there are centripetal oscillations and a fibroblast ring that applies contractile radial forces is largely unsupported by the data provided. Beyond the biophysical mechanism, the authors' claims relating the biophysical mechanism and Sox9 expression also lack support. The theoretical model is too basic and, in my opinion, is

disproved by the authors themselves with their beautiful experiments on cell division patterns and basement membrane mechanics. In contrast to the first part, the experiments on cell division patterns and basement membrane are very interesting. While the conclusions of these experiments are not fully justified either (it is unclear how some experiments were done; see comments below), the observed phenotypes upon perturbation of cell divisions and MMPs are strong and clear, suggesting that these processes are the important ones (rather than myosin forces or potential fibroblasts rings).

Overall, I think the manuscript touches on an interesting topic, but many of the claims regarding the proposed mechanisms are not supported by the data provided at this point.

We thank the reviewer for the expert assessment and constructive comments that greatly helped us to improve the manuscript. Based on these comments we have performed substantial new experimentation and carefully edited the manuscript to further strengthen the evidence for the mechanisms identified.

Specifically, we have:

1. Performed additional imaging, quantifications, and laser ablation experiments to better demonstrate the contractile forces exerted by the fibroblast ring
2. Performed additional confinement assays and imaging to demonstrate that compressive forces enhance Sox9 expression to compartmentalize placode cells
3. Strengthened the modeling by adding dynamic regulation of basement membrane mechanics and generating additional predictions from the model that we experimentally test

We believe these new experiments now demonstrate the collaborative role of fibroblasts and epidermal contractility in producing the initial epidermal thickening and basement membrane remodeling together with cell divisions facilitating the downward budding.

Although we truly appreciate the constructive criticism on the modeling, which we address in detail in our responses to the specific points, we respectfully disagree with the reviewer that the model is “too basic”. First, we should emphasize that the base simulation we are using is both state-of-the-art and significantly constrained by experimental data. To our knowledge, there are very few 3D models published that capture both mechanics and cell shapes, and none that capture multiple tissue types interacting with the mechanics of extracellular matrix. Therefore, the development of this modeling platform in and of itself represents a major advance that we have not seen in any other publication.

General comments:

- Authors claim the existence of oscillatory deformations driving tissue elongation (L141). The data provided to support these claims is very unclear though. Fig. 2B seems to show the divergence going above and below zero, but it is unclear whether these are fluctuations or actual oscillations. To claim the

existence of oscillations, it is necessary to provide a clear measure of them, such as peak in a Fourier analysis of the signal or similar. Moreover, oscillations are characterized by multiple cycles, which I failed to see in the data. The only thing I can see in the Supplementary movies and in the analysis is a single contraction in the placode, but not oscillatory behavior. Perhaps there are repeated contractions, but this is not apparent in the data. Can the authors please provide a proof of such oscillatory behavior?

We appreciate this comment and realize that the movie clips provided were too short to see full cycles of oscillation. We have now replaced these movies and associated figure panels (New Fig. 2a-b and New Supplementary Movies 1, 2) with longer imaging experiments (longer imaging is unfortunately not possible due to embryo viability), where we now observe multiple cycles of contraction and relaxation, more clearly demonstrating the earlier conclusions.

In addition, to avoid any overstatement, we have edited the text and refer to fluctuations instead of oscillations throughout the manuscript.

- The laser ablation experiments around the placode (Fig. 2D) indicate that the tissue is under tension. The recoil and radial displacement observed are precisely the same than previously observed in circular ablations of a completely different epithelial tissue that had not placodes or any specific structure. It is true that the placode seems to be under tension. However, is there any difference between this observation around the placode and in any other location in the tissue? It seems that the authors are suggesting that the placode being under tension is somehow a specific feature of the mechanics of placode formation. I am not convinced that this is the case. Can the authors please perform circular ablations away from the placode and see if the mechanics is different in any way? It would also help to have data for circular cuts of different radii both inside and outside the placode, as these could tell if there are really differences in tension inside and outside the placode.

We realize that we were not clear enough on the description of this result, as we did not intend to convey that the placode is under tension whereas the interfollicular epidermis is not. The point we tried to make is that the placode-epidermis boundary is under high tension. We thank the reviewer for raising this point that we have now clarified in the text.

More importantly, as suggested by the reviewer, we have included data for additional ablations from both outside and inside the placode as suggested to provide more detailed quantifications of the tension distribution. These new experiments demonstrate that while tension at the interfollicular epidermis (IFE) and IFE-placode boundary is high and comparable to tension within the epidermis, tension within the placode itself is lower (new Fig. 2d; new Supplementary Fig.2c-d and new Supplementary movies 4, 5). These new data are fully consistent with the previous data on distribution of myosin activity which showed low myosin activity inside the placode (Fig. 2e-g). We thank the reviewer for this very helpful criticism.

- L171: The authors claim that there is a “dense ring of fibroblasts” surrounding the placode. Unfortunately, I failed to see this in the data provided (Fig. 2I). In Fig. 2I, I see that there is no vimentin in the placode and that there is vimentin around, with the fibers oriented along the border of the placode. This is very interesting. However, I do not see the fibroblasts in any image or a quantification of their density or ring formation in any way. Can the authors please quantify the density and alignment of fibroblasts around the placode to verify if there is a “dense ring of fibroblasts”?

This is an important point and we have taken several steps to strengthen this observation.

1. We now provide higher resolution images and careful quantifications of the arrangement of the fibroblasts to strengthen this point (new Fig. 2h).
2. We perform laser ablations to show that the ring is contractile and exerts tension on the placode as demonstrated by the recoil of the placode towards the ablated area in the ring (new Fig. 2i and new Supplementary Movie 6).

- Following on the previous comment, in L178-179, the authors claim “a ring of contractile fibroblasts developed to tightly wrap around the placode base, coinciding with the centripetal oscillations in this area”. I do not see almost any data supporting such a strong claim. Since this is one of the main claims of the manuscript and central to the proposed mechanism, the authors should provide strong evidence for it. Currently, there is no data proving that there is a ring of fibroblasts, let alone that they are contractile along a ring surrounding the placode, or that there exist oscillations (as mentioned above). It seems that the authors believe that a ring of fibroblasts wraps around the placode and contracts, driving a radial contraction. It is a very nice hypothesis, but the data does not support many parts of it as the paper stands now. I suggest that the authors label the fibroblasts (immunostaining should be enough, but live imaging would provide more information) and check that there is indeed a higher concentration of them around the placode and forming a ring. I suggest also that the authors perform radially-oriented laser ablation cuts in the region of the dermis where there presumably is a contractile ring, as well as radially-oriented cuts in the dermis further away from the placode. These experiments would directly test the authors hypothesis that there is a fibroblast ring that is contractile, thereby generating an inward (radial) force on the placode.

As discussed in the previous response, we agree that this is a critical point and have we now carried out the suggested experiments:

1. We provide higher resolution images and careful quantifications of the alignment of the fibroblasts to strengthen this point (new Fig. 2h).
2. We perform laser ablations to show that the ring is contractile and exerts tension on the placode as demonstrated by the recoil of the placode towards the ablated area in the ring (new Fig. 2i and new Supplementary Movie 6).

We thank the reviewer for the constructive criticism that helped us to strengthen this conclusion.

- Following in my previous point, the circular vimentin arrangement around the placode could arise from the expansion forces generated as the placode invaginates into the dermis, providing a passive resistance rather than being the driver of a contraction by fibroblasts. Looking at all the data provided, I think this model is equally plausible at this point and the one the authors suggest.

We agree that our previous data did not fully exclude this possibility. To do this we have performed a panel of laser ablation experiments that show that

- 1) the placode itself is under low tension (new Fig. 2d)
- 2) the fibroblast ring is contractile and under tension, thus contradicting a passive resistance state (new Fig. 2i and new Supplementary Movie 6)
- 3) whereas the placode itself is under low tension, upon ablation of the fibroblast ring it shows displacement towards the fibroblast cut indicative of contractile forces exerted by the fibroblast on the placode (new Fig. 2i and new Supplementary Movie 6).

- L181: The authors propose a 3D vertex model of the tissue with multiple layers. The model predicts that a high basal to apical wetting coefficient could explain some of the observed shape changes, but the radial forces are also needed to predict the observed tissue divergence. However, the experimental data (Fig. 2F,G) does not show a large change in myosin between basal and apical cell surfaces in the placode, so it is unclear to me how this observation fits with the predictions of the model. More generally, I am not sure how the model is helping understand the data. It is unclear that this model (either 1 or 2) represents what is really happening – there are many other qualitatively different models that could also explain the observed data. Theoretical models are useful when, in addition to roughly fitting some data, provide a clear prediction that can be then tested experimentally. Can the authors generate a prediction (for instance, if the wetting coefficient changes, then the cell shapes change in a particular way) and test it experimentally? It would be useful to make a prediction that could specifically differentiate models 1 and 2 and then test it. However, I think it is important to consider the possibility that neither of these models properly accounts for the experimental system, as it does not incorporate in any details of the dermis, it is unclear if the mechanics of the real boundaries is the one assumed in the model, etc. For instance, there could be a localized softening of the basement membrane, as observed in lung branching morphogenesis or in tooth formation, that could explain the localized thickening of the placode.

We thank the reviewer for this comment, as it emphasizes that we need to be clearer both about the hypotheses we are testing with the model and how we are comparing model predictions to experiment.

First, we should emphasize that the base simulation we are using is both state-of-the-art and already significantly constrained by a panel of experimental data. To capture placode shape deformations, we require a fully 3D model with multiple cell types, as there are deformations both in the plane perpendicular to the basement membrane and out of the plane in the placode that must be predicted. To our knowledge, there are very few 3D models published that capture both mechanics and cell shapes, and none that capture multiple tissue types interacting with the mechanics of extracellular matrix. Therefore, the development of this modeling platform in and of itself represents a major advance that we have not seen in any other publication.

Specifically, we want to point out that we see a strong difference between experimentally quantified apical and basal myosin as shown in Fig. 2e and quantified in Fig. 2g, which we then used as one of the parameters in the model.

The reviewer states that our model “does not incorporate details of the dermis”. It is important to emphasize that we have incorporated a spring network at the interface between basal cells and the basement membrane. Such spring networks have been shown in previous work to capture elasticity, stiffness, and strain stiffening of extracellular matrix networks. Moreover, 2D vertex models are extremely well-validated and have been shown to be *quantitatively* predictive of mechanics and cell shape in epithelial monolayers in vivo (e.g. Wang PNAS 2020), and also cell sorting between cell types with adhesion knockdowns in culture (Sahu Soft Matter 2021). Although careful quantitative 3D reconstruction of cell shapes is painstakingly difficult even in fixed cell culture experiments, recent work by some of us has demonstrated that the 3D vertex model predictions of correlation between cell shape and tissue fluidity in simple systems with one cell type are also realized in cancer cell aggregates (Grosser PRX 2022). With all of this validation, we expect that 3D vertex models are a good starting point for testing mechanical hypotheses.

We agree with the reviewer that we have to fix the model parameters to make predictions. In order to fix parameters for 3D vertex models in the mouse epithelium, we have reconstructed 3D shapes in mosaically labeled cells in the basal and suprabasal layers, fit them to ellipsoids to quantify the major and minor axes dimensions, which fixes the vertex model parameters for the cell shape indices in those layers. The remaining relevant model parameters are: 1) stiffness of the basement membrane (both within and outside the placode), 2) interfacial tensions at the heterotypic interface between basal and suprabasal cells, as well as basal cells and the basement membrane, 3) shape index of placode cells, and any 4) additional forces generated by cells or tissues that are not explicitly included in our model. As mentioned above, experimental observations of myosin additionally constrain interfacial tensions. They show a higher myosin intensity on the apical side compared to the basal side of placode cells. In addition, the apical interfaces in both the placode and surrounding basal cells appear to have similar myosin intensities, and the same is also true for the basal interfaces. This indicates that in our model, the tension on the apical

interface can be parameterized by a single value that is larger than that on the bottom interface, though the ratio is difficult to constrain quantitatively.

Finally, there are forces generated by cells or tissues outside the basal and suprabasal layers. One obvious candidate is the fibroblast cells that begin to surround the placode in the dermis. Although the original version of the paper simply assumed that they were generating contractile forces in a ring that push the placode radially inward, we have now generated additional experimental laser ablation data to confirm that this is in fact the case. This is included in our model as a set of forces external to the placode acting inward, but again, the magnitude of these forces is difficult to constrain quantitatively in vivo.

Given the difficulty in constraining these parameters quantitatively experimentally, the utility of the simulations/modeling is that we can study *all possible* values of those unconstrained model parameters that are consistent with the qualitative data. The model then makes quantitative **predictions** about the **range** of possible cell shapes that are possible given each mechanism, given our experimental uncertainty. The grey and blue boxes in Fig 2 are therefore predictions of the model that can be compared to experiments. This comparison then allows us to state that the **entire range** of possible changes to autonomous cell shapes in the placode are not sufficient by themselves to generate the curvature and elongation seen in the experiments in E15.5 mouse embryos (in other words, the experimentally observed range of shapes given by the red box does not overlap the shapes that can be induced by this mechanism, e.g. the blue region). We have altered the text of the manuscript to emphasize these points.

In addition, as the reviewer points out, there was one remaining under constrained parameter in our model, which is the stiffness of the basement membrane both inside and outside the placode. We thank the reviewer for this helpful comment, as we have now performed a complete set of simulations to look at the placode structure induced by both global softening of the basement membrane and localized softening of the basement membrane, consistent with qualitative experimental observations. Interestingly, softening under the placode vs. softening of the entire region generate very similar placode morphologies. The new simulation data for softening under the placode is now shown in updated figures in the main text, and generally indicates that softening generates morphologies quite similar to cell-autonomous changes to the placode.

New Supplementary Figure 6f. Lower basement membrane stiffness shows higher invagination, basal-apical interfacial area ratio, and curvature into the basement membrane in a similar magnitude to that of changing lateral interfacial tension. Still, this mechanism alone isn't enough to recover the morphological changes seen at E15.5

In case the reviewer is concerned that the underlying vertex model is flawed, we also make additional predictions that can be tested. One *prediction* of our model is that if the system is perturbed so that the interfacial tension on the apical side is significantly reduced, the placode will invaginate upward (**new Supplementary Fig. 4g**). We now show that this model prediction is realized in experiments by analyzing a later developmental stage of the epidermal myosin IIA knockout that shows this phenotype (**new Supplementary Fig. 4h, i**).

New Supplementary Fig 4g. Modeling predicts that if apical interfacial tension is significantly reduced the placode will invaginate into the suprabasal layer instead of the basement membrane. Curvature in these phase diagrams is measured into the suprabasal cells instead of the basement membrane.

- The authors report that in myosin knockout mice, the placode is less deep than in controls (Fig. 4B), but that the shape of cells is not different in controls and in myosin knockout mice (Fig. 4c). How is this possible? The authors seems to conclude from the previously presented data and the simulations, that changes in myosin (wetting coefficient) would change cell shapes, which in turn would cause changes in the placode depth. It is unclear to me how to make these observations consistent with each other. Can the authors please explain it?

We thank the reviewer for this question as it seems that we have not been sufficiently clear on the mechanism of invagination. First, it is critical to emphasize that myosin II activity is particularly high in the suprabasal layers that do not elongate (see Fig. 1b and 2e), and the Keratin14-Cre that we use will also delete myosin IIA in these cells. Second, we observe that placode invagination has two components: 1. Apical and suprabasal contractility of the epidermis generates force to push the placode into the dermis generate invagination. 2. Extrinsic in plane compression that deforms the placode cells and constrains the placed while its budding. By deleting epidermal myosin II, we are impacting the first but not the second mechanism. Thus, the cells will elongate but will not be efficiently pushed against the dermis and the

dermal condensate by the suprabasal cells, resulting in less invagination and embedding of the dermal condensate inside the placode at later stages in the absence of substantial effect on basal cell shapes.

- The myosin inhibition experiments in fibroblasts, as well as global chemical inhibition with blebbistatin are very interesting, and also puzzling. Specific inhibition of Myh9 in fibroblasts is a natural test for some of the predictions of the model 2, as the authors point out. There seems to be some changes associated with the loss of Myh9 in fibroblasts, but the perturbation cannot stop placode formation of Sox9 expression, indicating that likely there is some other mechanical driver of placode formation. Can the authors perform radially-oriented laser ablation cuts in the observed vimentin ring to directly remove tension in case there is some there? The global chemical inhibition of myosin with blebbistatin is very interesting because it is the perturbation that leads to a stronger phenotype, with placodes not forming. The fact that this inhibition really perturbs placode formation suggests that other myosin-dependent processes not considered in the model can be causing the forces behind placode formation. Myosin inhibition makes Sox9 expression more diffuse in the tissue, but cells keep expressing Sox9. Since in the presence of myosin the placode cells seem to converge and define a higher density region (Fig. 4I), the compartmentalization of Sox9 seems to be more associated with the changes in placode morphogenesis than with any effect of force on Sox9 expression. Can the authors please check Sox9 expression after performing laser ablations to release tension in the tissue?

As discussed in response to previous points, we have now directly demonstrated tension in the fibroblast ring using laser ablation as suggested by the reviewer.

Regarding the Sox9 expression, we realize that we were not sufficiently clear in describing our results and conclusions as we did not intend to convey that tension is required to induce Sox9 expression as evident from explant experiments where blebbistatin treatment does not block Sox9 expression; Fig 4h, j). What we demonstrate is that confinement enhances Sox9 expression (now in Supplementary 5o, p) and is thereby required to specifically compartmentalize Sox9 expression into the placode. This conclusion was supported by the results in the blebbistatin-treated explants as well as in the Myh9-fibroblast KO where Sox9 fails to be restricted to the placode. We have performed additional imaging to increase the quality of the data (new Fig 4f, j) as well as edited the manuscript to clarify this conclusion.

We acknowledge that monitoring Sox9 upon release of tension such as post laser ablation would be an interesting experiment. However, it is not technically feasible for multiple reasons. First, detecting differences in protein levels would require both changes in expression as well as degradation of the existing proteins, with the half-life of Sox9 been shown to be around 5h. These time scales are not compatible with the laser ablation experiments which release tension within seconds, after which repair is initiated. Second, to be able to monitor Sox9 protein expression before and after ablation, we would further need a live Sox9-GFP fusion protein reporter mouse, which to our knowledge does not exist and this mouse would have to be crossed to a Td-Tomato or other cell membrane marker mouse to facilitate the ablation. Setting up such experiments would require 1-2 years of additional work and even then it

would be uncertain if monitoring Sox9 after ablation would be possible (to our knowledge this has not been done). Thus, we feel that this experiment is out of the scope of the current manuscript.

We have now taken several steps to experimentally strengthen this conclusion. First, we have improved the quality of the images and clarified the conclusions of the in vivo data from the Myh9-fibroblast knockout mice and blebbistatin-treated explants. What is evident from the imaging analysis is that Sox9 becomes compartmentalized into a ring of cells at the placode-epidermis interface, which is the zone of high tension compared to placode interior (Fig. 4f and 4j). This compartmentalization does not occur properly in Myh9-fibroblasts KOs or blebbistatin-treated explants (Fig. 4f and 4j).

To experimentally test if this is a simple downstream effect of abnormal tissue development or if Sox9 is mechanically regulated at this tension interface, we have now performed new experiments where we microfabricate PDMS gels to recapitulate the interface between the flat epidermis and the invaginating placode. We then plate freshly isolated epidermal cells onto these microfabricated patterns. In striking resemblance to the in vivo scenario, we observe high Sox9 expression specifically at the interface of the engineered invagination where the cells also show the highest deformation (new Fig. 4l, m). This, together with the confinement experiment, conclusively demonstrates the role of mechanical forces in tuning Sox9 expression and compartmentalization of these cells. We are grateful for the critical feedback of the reviewer that helped us improve the experimental design and strengthen our conclusions.

- Following up on the previous point, the authors claim that Sox9 increases upon application of pressure to epidermal progenitor cells in culture conditions. While there seems to be an effect in Fig. 4N, the quantification in Fig. 4O shows a very small change in expression. Because of the small change, I am not convinced by these experiments. However, performing laser cuts in the tissues and measuring Sox9 expression levels should give a clear answer to this question.

As discussed in the previous response, we acknowledge the reviewers point but the suggested laser cut experiments are technically extremely challenging and likely not technically feasible, at least in the revision timeline.

As mentioned above, to experimentally test if this is a simple downstream effect of abnormal tissue development or if Sox9 expression is under mechanical regulation at this tension interface we have now performed new experiments where we microfabricate PDMS gels to recapitulate the interface between the flat epidermis and the invaginated placode. We then plate freshly isolated epidermal cells onto these microfabricated patterns. In striking resemblance to the in vivo scenario, we observe high Sox9 expression specifically at the interface of the engineered invagination where the cells also show the highest deformation (new Fig. 4l, m).

In addition, we agree with the reviewer that the quantification of the compression experiments did not look particularly striking. We were initially puzzled to why the quantifications did not fully recapitulate the robust upregulation of Sox9 that we saw when we were carefully examining the raw data and realized that there was experiment to experiment variability in the intensity of the Sox9 staining and thereby pooling the results across experiments flattened out the differences. We have now replotted the quantifications as means of the individual biological replicates to better represent the results of these experiments (new Supplementary Fig. 5p).

This re-analysis together with the new microfabrication experiments conclusively demonstrates the role of mechanical forces in tuning Sox9 expression and compartmentalization of these cells. We are grateful for the critical feedback of the reviewer that helped us improve the experimental design and strengthen our conclusions.

- The observations presented on the pattern of cell divisions are very interesting. The complete switch in the proliferation pattern between E14.5 and E15.5 is remarkable. I think the fact that the authors saw a near completely disruption of placode formation in the presence of blebbistatin is not because of the mechanism they proposed, but rather because blebbistatin is affecting cell division. Can the authors measure the cell division pattern in the presence of blebbistatin to test this possibility?

We agree that this is an important point, but it is important to emphasize that we used low doses of blebbistatin (10 nM) to avoid dramatic effects on tissue viability and proliferation. We have now included quantifications of this showing that blebbistatin does not impact the pattern of cell divisions (new Supplementary Fig, 5k).

- The experiments to determine spatial changes in the mechanics of the basement membrane are also very interesting. However, I do not understand the AFM measurements. It is notoriously difficult to measure the mechanics of basement membranes in vivo. Did the authors isolate basement membranes or the entire tissue? If they measure the stiffness of the tissue, including the cells, the measurements do not reflect changes in the mechanics of the basement membrane because the sample is only indented by 0.5 microns, which would test the mechanics of the cells in contact with the AFM bead, but not the basement membrane. It is unclear from the methods section if the authors used isolated basement membranes and how such experiments were done. Can the authors please provide more details on these experiments? What was the size of the bead used for indentation?

We thank the reviewer for bringing up this point as it seems that we did not describe the AFM measurements in sufficient detail. The force indentation spectroscopy that we perform with the AFM to approximate the elastic modulus of the basement membrane is a contact-based method, so the cantilever needs to be in direct contact with the material that is being measured. Thus, the only way to be able to

access non-flat structures such as the basement membrane at the bottom of the placode is to snap freeze the tissue, freshly cut thick cryosections (20 μm), optically identify the basement membrane and perform force mapping of the cross-sectioned basement membrane, while cells are not being measured.

AFM force mapping of freshly cut cross-sections of snap frozen tissue is widely used technique frequently employed by us and others in the field (see for example Stashko et al., Nat Commun 2023 doi.org/10.1038/s41467-023-39085-1 and Fiore et al., Nature 2020 doi.org/10.1038/s41586-020-2695-9). To ensure that the fresh frozen tissue maintains the mechanical properties of the basement membrane, we have initially carefully calibrated these measurements by direct comparisons with measurements of non-frozen, freshly isolated tissue where the basement membrane has been measured by carefully decellularizing the epidermis with mild detergent to expose the basement membrane without cryofreezing (for description of the method and controls see for example our previous study Koester et al., Nat Cell Biol 2021). Elastic moduli of this freshly isolated basement membrane of the interfollicular epidermis are directly comparable to the measurements from the cryosections in the interfollicular epidermis. Thus, we are confident about these measurements. We have now edited the methods section to provide more detail. We used a pyramidal tip, not a spherical cantilever, to ensure that we have the resolution to specifically measure the basement membrane.

Fig.1 for the reviewer. Comparison of force indentation spectroscopy to estimate elastic moduli of basement membranes from freshly decellularized skin explants and fast frozen cryopreserved tissue cross sections.

- The experiments inhibiting MMPs are beautiful. The observed phenotype upon chemical inhibition of MMPs shows that clearly the ECM (perhaps not only basement membrane) plays an important role in placode formation.

We thank the reviewer for this positive comment!

Specific comments:

- The sketch in Figure 1A should be improved to include many of the terms and cell types that are described in the text. For instance, the mesenchymal cells are missing in the sketch and many steps of cell state changes are not described. I would strongly suggest to make a sketch for the different steps that defines the different cell states and main morphological features.

We thank the reviewer for this excellent suggestion, we have now improved the schematics in Fig. 1a.

- The term “isotropic elongation” (line 107) is rather unusual. Elongation is anisotropic, so it is quite unclear what is meant by isotropic elongation. Perhaps the authors can find a different term?

We intended to convey lack apical constriction in the elongated cells but agree that the term “isotropic elongation” is confusing. We now just say “elongation”.

- It seems quite difficult to identify FGF20 positive cells from the images in Fig. 1B, especially at E13.5. Can the authors please provide supplementary figures showing the good representative images and how they are analyzed? Also, the definition of cell geometry with basal and apical lengths can obscure the results because cell lengths on both apical and basal polygonal cell shapes are different in different directions by about 30% or so. Perhaps a proper definition of what length is measured would help identify relevant morphological changes.

We apologize for the suboptimal quality of the images, resulting from the technical limitation that good FGF20 are not available and we utilized is a beta-gal reporter allele. Detection of beta-gal was not as straight forward as classic immunofluorescence. To improve the clarity of these images, we have now moved the FGF20 images to the supplementary and use Edar staining to detect the early placode cell state and to display the morphology of these cells more clearly (new Fig. 1b)

Three cell shape parameters have been measured from skin whole mounts; apical and basal cell surfaces and the cell height in the z-direction. Only the cells in contact with the basement membrane, the basal cells, have been quantified. To perform these measurements, placode cells were identified using FGF20 beta-gal or Edar. Phalloidin staining was then used to determine the cell contours. To be able to visualize the top and bottom of the cells, images were taken with high z-resolution (0.5 μ m), which were then resliced in Image J software and the appropriate stack; (stack where the cell has the longest z-axis), has been selected individually for each cell. The measurements were then manually carried out for each single

cells. Importantly, only the basal cells with clear contours have been quantified. The same method has been applied for all stages and conditions. This is an extremely labor-intensive process and we have taken great care to obtain accurate measurements and are confident about this data. We have added more detailed description of these measurements in the materials and methods section.

- FGF20 positive cells at E14.4 and E15.5 (Fig. 1b) look morphologically very different to me, but the morphological quantification in Fig. 1D shows essentially no significant differences in basal and apical length between these two stages. Why? Can the authors please benchmark their analysis to be sure that the quantification is adequate? Can they provide the error associated with these measurements (not the ensemble error of different replicates, but rather the error in the segmentation and shape analysis) to be sure that the quantification is meaningful?

We thank the reviewer for raising this important point. As accurate segmentation is critical for our study, we had taken several steps to benchmark our manual segmentation approach in the x-z cross-sections shown in Fig 1b (as mentioned above, we have now also replaced this panel with higher quality images). This segmentation could not be automated due to the complexity of the overall tissue structure. However, to ensure that the measurements for the apical and basal lengths were accurate, we performed the same analysis in the x-y images where we could computationally segment all cells using Tissue Analyzer, followed by manual correction. We segmented cells at E13.5 for the same region that was analyzed for the x-z cross-section, first segmenting the apical and then the basal side from the confocal stacks. The area of the cells (A) in both planes were computed and $L = \sqrt{A}$ was employed to define the characteristic length scales of the cells in the two planes. We determined the mean characteristic length scale of the cells in the apical and basal planes as 8.50 and 8.60 μm respectively. The characteristic length scale difference here is 0.10 μm which is $< (0.72 \mu\text{m})$ or 3 pixels - the quintessential threshold in Tissue Analyzer to define cell-cell boundaries. Thus, we consider this error to be minimal and not affecting the biological conclusions. Importantly, our manual segmentation results in Fig 1b at E13.5 are in the exact same range as results obtained using the Tissue Analyzer-segmented images.

Reviewer #2:

Authors demonstrated (i) an elongation in cell morphology between E13.5-E15.5 (i.e., when Sox9 expression is observed in future HFSCs) and (ii) contractile oscillations around placode neck on E14 (placode stage) being driven by a ring of fibroblasts. The authors laid out the argument that this fibroblast-driven process is mediated by myosin IIA. Furthermore, Sox9 expression was more robust w/ increased compression. The ideas are novel, and the experiments seem appropriate. However, there are some major issues that may be attributing to significant biases in the study.

We thank the reviewer of the positive assessment on the significance and quality of our study, as well as the expert critical comments that helped us to further improve it.

1. Lack of an objective placode marker:

Fgf20 β Gal embryos were used to generate the data regarding placode cell morphology. Subsequent live imaging experiments utilized membrane-targeted Tomato reporter mice. Based on the figures in the paper, it appears that placodes were identified based on morphology. As a result, the lines drawn to demarcate the placode seem quite heterogeneous with some lines containing too much and some containing too little. This represents a significant source of bias for the measurement of placode depth and length. I recommend that the authors provide data using immunostaining for FGF20 for at least some of these experiments to validate their findings.

We thank the reviewer for this comment, which we agree is an important point. There are unfortunately no good FGF20 antibodies available, we were therefore using FGF20 beta- gal reporter but crossing these mice into all the other mutations would be an extensive endeavor time and resource-wise. To address this, we have now included Edar as an additional marker for the placode to facilitate unbiased quantifications (new Fig. 1b-e, 4a-c, 4d and Supplementary Fig. 5f).

2. Myosin IIA data:

Page 7, line 232 describes myosin IIA as “the major myosin isoform responsible for keratinocyte actomyosin contractility”. The authors claim that myosin IIA is essential for contraction by the ring of fibroblasts.

Was there a predetermined fluorescence threshold used by the authors for the vimentin staining analysis? The ring width of vimentin for Figure 2d looks almost identical between the control group and the Myh9-dKO group (maybe just a little blurrier in the Myh9 group). However, the arrows are smaller for the dKO group. Again, the methods need to be clarified or the data needs to be analyzed in a more objective manner.

The claim of myosin IIA as the major myosin is based on our previous work (Le et al., 2016) where we show that the Myh9KO cells completely fail to establish a contractile actomyosin cytoskeleton, indicating that the other myosins do not compensate. These epidermal Myh9-deficient mice also die in the first days after birth, indicating lack of compensation by myosin IIB, a knockout of which results in viable offspring. This is consistent with abundant literature showing that the IIB and IIC isoforms, of which IIB, is expressed in the epidermis, have more specialized roles.

To address this point we have now improved the images of the fibroblast ring to better indicate the clear phenotype of these mice (new Fig 4d; Supplementary Fig. 5b).

Furthermore, there is a lot of functional redundancy between NMII isoforms, especially NMIIA and IIB. This is supported by the lower # of placodes in the blebbistatin-treated group (inhibition of NMIIA and IIB) vs. Myh9-dKO group (inhibition of NMIIA only). It also looks like placodes still form w/ slightly

reduced invagination. The claims regarding the “essential” role of myosin IIA either needs to be attenuated or be revised to state that it is possible that the contractions are being mediated by other NMII isoforms.

Given the perinatal lethality of NMIIA-knockout mice (Le et al., Nat Cell Biol 2016), there is clearly limited compensation. Nevertheless, we fully agree that we cannot exclude some compensation from other myosin isoforms on placode development and have altered this conclusion to indicate it. We now state on pages 9-10 of the manuscript “The above genetic experiments implicate a major role for myosin IIA in this process, but additional or compensatory effects from myosin IIB are also possible (Sumigray et al, 2012. PMID: **23091070**).”

3. AFM data:

AFM was performed to measure the stiffness of the basement membrane. Prior studies that performed a similar experiment removed the surrounding cells in a systematic manner (J Struct Biol. 2009 Jul;167(1):19-24.) to ensure that the measurements were taken accurately. Was this performed? If not, how can you be certain that measurements were not taken from surrounding tissue?

We thank the reviewer for bringing up this point as it seems that we did not describe the AFM measurements in sufficient detail. The force indentation spectroscopy that we perform with the AFM to approximate the elastic modulus of the basement membrane is a contact-based method, so the cantilever needs to be in direct contact with the material that is being measured. Thus, the only way to be able to access non-flat structures such as the basement membrane at the bottom of the placode is to snap freeze the tissue, freshly cut thick cryosections, optically identify the basement membrane and perform force mapping of the cross-sectioned basement membrane, while cells are not being measured.

AFM force mapping of freshly cut cross-sections of snap frozen tissue is widely used technique that is frequently employed by us and others in the field (see for example Stashko et al., Nat Commun 2023 doi.org/10.1038/s41467-023-39085-1 and Fiore et al., Nature 2020 doi.org/10.1038/s41586-020-2695-9). To ensure that the fresh frozen tissue maintains the mechanical properties of the basement membrane, we have initially carefully calibrated these measurements by direct comparisons with measurements of non-frozen, freshly isolated tissue where the basement membrane has been measured by carefully decellularizing the epidermis with mild detergent to expose the basement membrane without cryofreezing (for description of the method and controls see for example our earlier study Koester et al., Nat Cell Biol 2021). Elastic moduli of this freshly isolated basement membrane of the interfollicular epidermis are directly comparable to the measurements from the cryosections in the interfollicular epidermis. Thus we are confident about these measurements. We have now edited the methods section to provide more detail.

Fig.2 for the reviewer. Comparison of force indentation spectroscopy to estimate elastic moduli of basement membranes from freshly decellularized skin explants and fast frozen cryopreserved tissue cross sections.

Some additional questions:

Initially there was more cell proliferation in the surrounding epidermis than in the placodes. Is it possible that the expansion of the surrounding epidermis helped "push" the less proliferative placodes downwards?

This is a very interesting point and we have also considered that this might be the case based on our finding that Mitomycin treatment of E14 explants impairs placode formation although there is no proliferation in the placode at this time point (Fig. 3 for the reviewer). However, we interpret this result with caution as we do not have technical means to block proliferation in exclusively the epidermis and cannot fully exclude effects within the placode. The strong phenotype of the fibroblast Myh9 KO mice further indicates that the forces from the dermis are the main source of extrinsic forces (Fig. 4d).

Fig.3 for the reviewer. E13.5 Skin- explants treated with DMSO or Mitomycin for 3 hours and subsequently cultured for 24h. Data show a significant reduction of placode development, characterized by a decrease of placode diameter and depth. (n= 12 (DMSO) and 11 (Mitomycin C) placodes from 3 mice; Student's test (placode diameter) and Mann-Whitney (placode depth).

If Sox9 expression is dependent on mechanical tension, then why does the blebbistatin treated tissue still have high levels of Sox9 expression?

Regarding the Sox9 expression, we realize that we were not sufficiently clear in describing our results and conclusions as we did not intend to convey that tension is required to induce Sox9 expression as evident from explant experiments where blebbistatin treatment does not block Sox9 expression; Fig 4h, j). What we demonstrate is that confinement enhances Sox9 expression (Supplementary Fig. 5o, p) and is thereby required to specifically compartmentalize Sox9 expression into the placode. This conclusion was supported by the results in the blebbistatin-treated explants as well as in the Myh9-fibroblast KO where Sox9 fails to be restricted to the placode. We have performed additional images to increase the quality of the data (new Fig 4f, j). as well as edited the manuscript to clarify this conclusion.

Any thoughts on what caused the initial difference in apical/basal cell properties in the very early stage of placode formation?

This is an excellent but complex question. As the pattern of myosin evolves over time, we hypothesize that the pattern of apical contractility develops to counteract forces from the maturing dermis to prevent the tissue from folding or buckling. The low contractility inside the placode is likely a result of commitment of these cells to the hair follicle lineage that involves specific changes in the cytoskeleton downstream of the cell fate change in to FGF20-expressing cells.

Neither Myh9-eKO or dKO abolished the initial placode pattern formation (the KO mostly just reduced

the placode down-growth, or maybe reduced the density of the placodes). So, which myosin component was involved in the initial pattern formation, or compensated for Myh9 after its deletion?

Importantly, blebbistatin treatment which blocks NMII in both compartments, fully prevents placode formation, supporting our notion that dermal and epidermal myosin contractility collectively generate the initial placode pattern. The problem in the field is that we lack a Cre line that would delete early in epidermal development. We are using a Keratin-14-promoter driven Cre (Hafner et al. Genesis 38, 176-181, 2004) that is known in the field to be a Cre line with the earliest deletion in developmental time, but even this early Cre line produces an efficient deletion around E12-E13. Interestingly, the second wave of hair follicle morphogenesis, which is initiated at E15 is severely affected and almost entirely blocked in the dermal myosin KO mice (new Fig. 4d, e), supporting the notion that dermal myosin is essential but the deletion occurs slightly too late to fully prevent the first wave of morphogenesis.

Minor edits:

1. For the “side view” photos, a more widely cropped or a more zoomed out image would be more appropriate. For example, in Figure 1b, it is difficult to appreciate the cell morphology for the top panel (E13.5) since the top portions of the cells are not visible. It is also impossible to see the dermis in these images.

We agree and have replaced the side views in Fig. 1b.

Reviewer #3:

Remarks to the Author:

In this study, Villeneuve et al. aims to characterize the mechanical forces and morphogenetic behaviors that placode cells encounter during embryonic hair follicle development. Specifically, the authors used biophysical approaches to look at placode stage (E14.5) and hair germ stage (E15.5) and mapped out a two-step process of cellular elongation followed by downgrowth of the budding placode. They concluded that both cell-intrinsic contractility and extrinsic forces from the dermis contributed to the morphological changes in the placode. With a combination of genetic perturbation experiments and drug treatment, they disrupted the contractility in the epidermal and dermal compartment and showed both resulted in defects in placode formation and/or invagination. Mechanistically, they linked placode invagination at E15.5 with changes in extrinsic mechanical stress and resultant cell cycle reentry experienced by placode cells. They further attributed the release of mechanical force to the softening of the basement membrane during placode invagination. The biophysical experiments are methodical and provided new insights into the biomechanical forces that regulate epithelial morphogenesis. However, the manuscript can be made stronger if the authors could reconcile their findings and models with previous knowledge in the hair follicle biology field.

We thank the reviewer of the overall positive assessment our study, as well as the expert critical comments that helped us to further improve it.

Major comments:

1. It appears that the definition and the biological meaning of a ‘fibroblast ring’ is not very clear. The vimentin-rich status in the fibroblast ring does not necessarily mean that those fibroblasts exert extrinsic force on the placode that causes the placode invagination. It could be that the placode or other nearby epidermal cells exert forces on the cells in the fibroblast ring as it invaginates down. For example, it could be that epidermal cells neighboring the placode push the placode inward and downward and then the placode pushes on the fibroblasts, which would be in line with their laser ablation experiment. Therefore, the extrinsic force within the authors’ model could also come from neighboring epidermal cells.

We agree that this is a critical point and have we now carried out a panel of additional experiments to demonstrate that the fibroblasts indeed exert forces on the placode.

1. We now provide higher resolution images and careful quantifications of the alignment of the fibroblasts to strengthen this point (new Fig. 2h).
2. We perform laser ablations to show that the ring is contractile and exerts tension on the placode as demonstrated by the recoil of the placode towards the ablated area in the ring (new Fig. 2i and new Supplementary Movie 6).
3. We perform laser ablations within the placode to show that these cells are under low tension, consistent with their low myosin activity (new Fig. 2d-g), excluding the possibility that the placode cells exert force on the fibroblasts rather than vice versa.

These new data together with the previous findings showing that deletion of myosin specifically in the fibroblasts prevents placode formation and strongly impacts assembly of the fibroblasts around the placode now more clearly demonstrate the existence and relevance of this contractile “fibroblast ring”. We thank the reviewer for the constructive criticism that helped us to strengthen this conclusion.

Additionally, within their model, the authors could also take into account the potential forces exerted by the DC, which lies beneath the placode. The authors should incorporate this into their 3D vertex modeling. In this study, the center of the placode is assumed to not experience any force, even though it is in contact with the DC.

We agree that the DC is an important structure for the hair follicle. However, our analysis of myosin activity as well as myosin expression have shown that the DC cells are non-contractile and this unlikely to exert tensile forces on the placode, in contrast to the fibroblasts surrounding the placode (Fig. 4 for reviewer). This is consistent with the findings that deleting epidermal myosin leads to the DC being embedded inside the placode (Supplementary Fig. 4e), indicating that the DC does not exert tensile forces on the epidermis, but rather represents a stiff, passive resistance on these cells. Given these findings, we focused on the contractile cell types for our force analyses.

Fig.4 for the reviewer. Confocal images from of E15.5 skin whole-mount. The white dashed line demarcates the dermal condensate, beta-gal staining (magenta) marks the FGF20-beta-gal-expressing placode. The picture shows a low pMLC2 levels in the dermal condensate. Scale bars; 20um.

2. The authors showed that Sox2⁺ DCs exist in the Myh9-eKO, which suggested that the defect in placode invagination mostly resulted from loss of contractility from the epidermis instead of other defects (e.g. loss of DC). The authors do not show the presence or absence of Sox2 DCs in the Myh9-dKO mutant (Figure 4a), the Blebbistatin-treated explants (Figure 4j), or the MMP-inhibitor-treated explants (Figure 5h). It would be helpful if the authors could also show the DC status in these experiments to show that the defects are not due to potential effects on other dermal populations (e.g. DC) that led to absence of HF induction.

As suggested, we now show status of the DC in the Myh9- eKO and dKO mutants (new Supplementary Fig. 5c), the Blebbistatin- and MMP-inhibitor-treated explants (new Supplementary Fig. 5j and new Fig. 5h respectively) to demonstrate that the DC is normally formed in all of the manipulations, as expected.

3. The authors concluded that E15.5 placodes undergo downgrowth due to increased nuclear YAP signaling and the resulting cell cycle reentry in basal placode cells, suggested by Figure 5b/c. 1) what is the DC status in Mitomycin-C treated explants vs control? 2) Can the authors comment on where the proliferation and YAP signaling occurs in the hair germ? For example Ouspenskaia et al. showed that Wnt-high basal cells are slow-cycling, so it would be surprising to see similar proliferation in the basal cells and suprabasal cells of the hair germ. Also, the authors could incorporate work published by Zhang et al 2010 PNAS, which examined the role of YAP in embryonic epithelial/hair follicle development.

As suggested, we have now analyzed the status of the DC in the mitomycin-treated explants and observe that it is still correctly positioned and morphologically unaffected (new Supplementary Fig. 6c).

Proliferation is seen in both layers with some more proliferation in the basal layer (Fig. 5 for the reviewer), which is largely consistent with Ouspenskaia et al. who show that cell divisions are equally distributed at the placode stage and the bias towards suprabasal divisions only emerges at the hair germ stage (E16-E17), which we don't analyze in the current study.

We now also refer to the study by Zhang et al 2020 PNAS that showed that YAP promotes proliferation in the hair follicle, consistent with our observations (p11 in the text). We thank the reviewer for pointing out the omission of this citation.

Fig.5 for the reviewer. Distribution of proliferative cells ($Ki67^+$ cells) between the suprabasal ($Keratin10^+$) versus the basal layer at E15.5 from tissue sections ($n= 11$ placodes from 3 mice; Student's test).

4. Although this paper is timely for the field, many of the findings are largely descriptive in nature. Although possibly beyond the scope of this paper, it would be nice if there were more *in vivo* functional experiments to test their model.

We are not sure how the reviewer defines descriptive findings. In this study we utilize 2 different genetic deletion systems as well as a number of specific inhibitor treatments to disrupt the cellular and molecular mechanisms that were predicted by the model. In response to this criticism, we have now added additional functional experiments to test our model:

1. We laser ablate the fibroblast ring to show that it exerts tension on the placode, as predicted by the model
2. We make an additional prediction with the model in that when the interfacial tension balance between the epidermis and the dermis would be inverted (lower in the apical/suprabasal planes of the epidermis and low on the basal /dermal side) the placode curvature should become inverted. We test this prediction by analyzing a later developmental stage of the *Myh9-eKO* mice, where the apical interfacial tension is very low, whereas the basal interfacial tension is high (due to

basement membrane stiffness) and demonstrate that indeed the placode now evaginates upward instead of invaginating downward (new Supplementary Fig. 4h, i).

Minor comments:

1. What was the control in Figure 4n? Was the control no compression at 0h, 3h or 16h?

The control was fixed together with the first compression time point to exclude effects from culture time. We have now clarified this in the methods section. In our broader experience with epidermal cell in vitro cultures Sox9 is relatively stable over time.

Decision Letter, first revision:

*Please delete the link to your author homepage if you wish to forward this email to co-authors.

Dear Professor Wickström, dear Sara,

Your manuscript, "Mechanical forces across compartments coordinate cell shape and fate transitions to generate tissue architecture", has now been seen by 3 of our original referees, who are experts in mechanobiology and biophysics (referee 1); hair follicles (referee 2); and hair follicle and scRNAseq (referee 3). As you will see from their comments (attached below) they find this work of interest, but have raised some important points. Although we are also very interested in this study, we believe that their concerns should be addressed before we can consider publication in Nature Cell Biology.

Nature Cell Biology editors discuss the referee reports in detail within the editorial team, including the chief editor, to identify key referee points that should be addressed with priority, and requests that are overruled as being beyond the scope of the current study. To guide the scope of the revisions, I have listed these points below. We are committed to providing a fair and constructive peer-review process, so please feel free to contact me if you would like to discuss any of the referee comments further.

We are willing to allow one more round of revision to address the referees' concerns. Please be aware that further reconsideration of this manuscript (in the event of re-review) will be conditional on the referees being fully satisfied with the extent of the revisions, as we generally do not encourage multiple review rounds at this journal

In particular, it would be essential to:

A) Assess tension in tissue surrounding the placode with laser ablation (Reviewer #1)

B) While we would not necessarily require new experiments, we would ask explicit discussion and justifications of boundary conditions and other parameters within your biophysical model (Reviewer

#1)

C) Similarly, while we would not necessarily require new experiments, we would require discussion on potential caveats of effects of mechanics on Sox9 and of proteolytic remodelling of the basement membrane based on your AFM measurements (reviewer #1), and toning down claims on effects on the cell cycle (Reviewer #3)

D) All other referee concerns pertaining to strengthening existing data, providing controls, methodological details, clarifications and textual changes, should also be addressed.

E) Finally please pay close attention to our guidelines on statistical and methodological reporting (listed below) as failure to do so may delay the reconsideration of the revised manuscript. In particular please provide:

While we would welcome data on these points, we would not necessarily require periodic laser ablation nor the use of additional Sox9-reporter lines as otherwise suggested by Reviewer #1, but would at least require discussion on these points as mentioned above.

We therefore invite you to take these points into account when revising the manuscript. In addition, when preparing the revision please:

- ensure that it conforms to our format instructions and publication policies (see below and <https://www.nature.com/nature/for-authors>).

- provide a point-by-point rebuttal to the full referee reports verbatim, as provided at the end of this letter.

- provide the completed Reporting Summary (found here <https://www.nature.com/documents/nr-reporting-summary.pdf>). This is essential for reconsideration of the manuscript and will be available to editors and referees in the event of peer review. For more information see <http://www.nature.com/authors/policies/availability.html> or contact me.

When submitting the revised version of your manuscript, please pay close attention to our [href="https://www.nature.com/nature-portfolio/editorial-policies/image-integrity">Digital Image Integrity Guidelines](https://www.nature.com/nature-portfolio/editorial-policies/image-integrity). and to the following points below:

- that unprocessed scans are clearly labelled and match the gels and western blots presented in

figures.

- that control panels for gels and western blots are appropriately described as loading on sample processing controls
- all images in the paper are checked for duplication of panels and for splicing of gel lanes.

Nature Cell Biology is committed to improving transparency in authorship. As part of our efforts in this direction, we are now requesting that all authors identified as 'corresponding author' on published papers create and link their Open Researcher and Contributor Identifier (ORCID) with their account on the Manuscript Tracking System (MTS), prior to acceptance. ORCID helps the scientific community achieve unambiguous attribution of all scholarly contributions. You can create and link your ORCID from the home page of the MTS by clicking on 'Modify my Springer Nature account'. For more information please visit www.springernature.com/orcid.

This journal strongly supports public availability of data. Please place the data used in your paper into a public data repository, or alternatively, present the data as Supplementary Information. If data can only be shared on request, please explain why in your Data Availability Statement, and also in the correspondence with your editor. Please note that for some data types, deposition in a public repository is mandatory - more information on our data deposition policies and available repositories appears below.

[Redacted]

We would like to receive the revision within four weeks. If submitted within this time period, reconsideration of the revised manuscript will not be affected by related studies published elsewhere, or accepted for publication in Nature Cell Biology in the meantime. We would be happy to consider a revision even after this timeframe, but in that case we will consider the published literature at the time of resubmission when assessing the file.

We hope that you will find our referees' comments, and editorial guidance helpful. Please do not hesitate to contact me if there is anything you would like to discuss.

Best wishes,

Daryl

Daryl Jason Verzosa David, PhD

Senior Editor, Nature Cell Biology
Nature Portfolio

Heidelberger Platz 3, 14197 Berlin, Germany
Email: daryl.david@nature.com
ORCID: <https://orcid.org/0000-0002-9253-4805>

Reviewers' Comments:

Reviewer #1:

Remarks to the Author:

I sincerely thank the authors for addressing my comments. To facilitate review at this point, I will not go over all the details, but rather focus on the main claims (as indicated in the abstract) and I will comment on whether or not I believe there is enough support for them after these initial revisions, and why.

Overall, I think the authors have addressed a lot of the questions convincingly, while some other claims remain, in my opinion, not sufficiently supported by data.

(1) The authors claim in the abstract that “we identify a key role for coordinated mechanical forces stemming from contractile, proliferative and proteolytic activities across the epithelial and mesenchymal compartments in generating the placode structure”

I somehow agree with this statement. The authors now provide more convincing data of the mechanical aspects of placode morphogenesis. In particular, the new data provide more evidence that there is indeed ‘ring of fibroblasts’ that is under tension. The new laser ablation experiments indicate that there is indeed tension in the ring of fibroblasts. However, no control of this experiment (by either cutting in the perpendicular direction or in the mesenchyme far away from the ring) is shown. So, I agree there is tension in the ring, but it is unclear still if there is more tension in the ring than in the surrounding tissue, as would be expected in the model presented by the authors. The authors should check that this is indeed the case by doing laser ablation in the mesenchyme away from the ring and confirming less tension far away.

Beyond this, the authors provided new evidence that there is more tension in the epidermis than in the placode. In my opinion, these data now justify the statements regarding the elongation of the placode.

(2) The authors claim in the abstract that “these mechanical stresses further enhance and compartmentalize Sox9 expression to promote stem cell positioning”

I thank the authors for their efforts to strengthen their claims on this respect. I agree with the authors that confinement seems to be a factor in Sox9 expression, and that confinement seems correlated with the extrinsic forces from the mesenchymal cells ring. However, it is unclear to me if it is the confinement of cells caused by the extrinsic forces, or the actual forces themselves that direct the pattern of Sox9 expression. The data provided only shows correlations in this respect, so I am not sure the authors can distinguish between Sox9 expression increasing because of mechanical constriction or because of the cells being more spatially confined as a result of morphogenetic changes. Mechanical induction of Sox9 was previously reported by the Gros lab (Parada et al., *Dev Cell*, 2021). Gros et al showed that internal mechanics in the tissue drives Sox9 specification during digit formation in mouse. Therefore, the authors may well be correct and mechanics may cause Sox9 expression in the placode too, but at this point I only see correlations.

The authors mention in their reply that they cannot assess the role of mechanics on Sox9 directly because there is no Sox9-GFP reporter mouse. This is not accurate, see for instance Parada et al., *Dev Cell*, 2021. Moreover, laser ablation can be repeated cyclically (periodic ablation) to prevent the tension recovery after ablation. I feel these experiments are not very complicated, but I understand that there are other indirect ways to address this question, albeit likely less conclusive.

The new experiments where isolated epidermal progenitor cells are placed into microwells are a great idea. However, I fail to see in the images any substantial increase in Sox9 at the boundary (Fig. 4I), as the authors claim. The experiments where epidermal progenitors are cultured in hydrogels are very nice too, and here there seems to be an increase of Sox9. However, this is *in vitro* and extrapolation *in vivo* is complex. I would like to point out that it is well known that ectopic compression drives Sox9 expression in mouse limb bud chondrogenic progenitors – see e.g., Takahashi et al., *J Cell Sci*, 1998.

I think the authors provide enough evidence that contractile extrinsic forces somehow, directly or indirectly, affect Sox9 expression. The authors should clarify that confinement does not induce Sox9 expression (as it is observed when myosin is inhibited), but rather enhances it upon confinement. It is also important that the previous literature on mechanical induction of Sox9 (the papers mentioned above and several others) is properly cited, and also the results reported here discussed in the light of previous works.

(3) The authors claim in the abstract that “proteolytic remodeling locally softens the basement membrane”

The authors provided some evidence that this is the case. The AFM experiments are not fully convincing because they are done on sections in which the mechanics of the basement membrane could be altered (both in cryosections and in freshly cut sections; the sectioning is the issue here). Also, it is unclear to me how these measurements isolate the basement membrane stiffness from the mechanical properties of the cells surrounding the basement membrane in the cryosectioned tissue, since AFM probes both simultaneously. Moreover, previous works have shown marked changes of basement membrane components at the site of budding that have been linked to changes in basement membrane stiffness (see e.g. paper by Ken Yamada on lung budding), so it is a bit surprising not to see changes in basement membrane composition at the invaginating site. However, both the single cell sequencing data and MMP inhibition experiments are convincing, and seem to indicate that basement membrane mechanics is likely involved in the invagination. As I said in my previous review, the MMP inhibition experiments are great and they clearly indicate that the basement membrane plays an

important role. If this role is mechanical or not, I am not sure, but it is possible.

Finally, regarding the simulations, the authors have not convinced me of their relevance in this work. I completely agree with the authors that these simulations are state-of-the-art at a technical level. We all agree that 3D vertex models are technically very challenging. Moreover, I agree that combining multiple tissues and matrix in the simulation is new. My issue is not with the technical aspects of the model, but rather the fact that there are many experimental aspects that are unknown and could affect the model. In other words, there are many assumptions when one develops a vertex model and we do not know if these assumptions are correct in this system. As an example, the authors did not include proliferation or dynamics of forces, etc. All these aspects could change the predictions of the model. Moreover, I strongly disagree with the authors comment that vertex models are 'extraordinarily well-vetted'. Vertex models have rarely been tested experimentally with direct mechanical measurements even in 2D tissues. Instead, researchers have used the models to fit the data and interpret the experimental observations from the viewpoint of the model. If the model is correct, that leads to correct interpretations, but if the assumptions are incorrect (because, for instance, cell proliferation or some other factor was neglected), the interpretation is flawed. In my opinion, the authors do not attempt to carefully test experimentally the model, but rather they interpret the experimental observations from the model assuming it is correct. In the best-case scenario, the model is correct and the data properly interpreted. In the worst-case scenario, the model is incorrect but roughly fits the data in some cases, leading to the belief that the interpretation of the data is correct.

As an example, the new data shows that inhibition of Myosin IIA in the epidermis leads to evagination (instead of invagination) of the placode. The authors claim this is the case in the simulations (Supplementary Fig. 4g), but I do not see any evagination in the simulations results that comes even close to the data in Supplementary Fig. 4h.

Reviewer #2:

Remarks to the Author:

I believe the authors have sufficiently addressed the concerns of the reviewers. The manuscript is well written and presented. The findings are interesting and advance our understanding of hair follicle formation and the cellular forces involved.

Reviewer #3:

Remarks to the Author:

In this study, Villeneuve et al. aims to characterize the mechanical forces and morphogenetic behaviors that placode cells encounter during embryonic hair follicle development. Specifically, the authors used biophysical approaches to look at placode stage (E14.5) and hair germ stage (E15.5) and mapped out a two-step process of cellular elongation followed by downgrowth of the budding placode. They concluded that both cell-intrinsic contractility and extrinsic forces from the dermis contributed to the morphological changes in the placode. With a combination of genetic experiments and drug treatment, they disrupted the contractility in the epidermal and dermal compartment and showed both resulted in defects in placode formation and/or invagination. Mechanistically, they linked

placode invagination at E15.5 with decreased extrinsic mechanical stress and resultant cell cycle reentry experienced by placode cells. They further attributed the release of mechanical force to the softening of the basement membrane during placode invagination.

Overall, the authors addressed my comments sufficiently. They used ablation experiments and displacement measurement to ablate the fibroblast ring and show the contractile force exerted on the placode by the fibroblast ring, indicated by the recoil of placode after the ablation (Figure 2i). Although beyond the scope of this work, it would be nice to see the effect of ablating the DC specifically as well. Placode tension was also measured (Figure 2d-g) to address the alternative model wherein the placode pushes fibroblasts downward. Additionally, Sox2+ DC's were shown to form normally in the new Figure 5h, suggesting the failure in placode invagination isn't an indirect effect from loss of DC. However, it is notable that the Sox2+ dermal cells do not appear to be clustered in Figure 5j, suggesting that blebbistatin affected the DC, which is not only defined by Sox2 expression. The confinement assay (Figure 4l) was done to show Sox9 expression is highest at the invagination interface where there was highest tension.

One remaining issue is related to the mechanistic link to cell cycle. For this study, the cell cycle aspect does not appear to add much, because the link between cell cycle regulation and mechanical forces is somewhat weak and disconnected and the data are not convincing (Figure 5a-d). The mitomycin-C treatment (Figure 5e-f) also does not necessarily establish that the defect of placode formation upon cell cycle inhibition is downstream of mechanical forces, which the authors seem to suggest in the manuscript. Also, the authors stated that Blebbistatin treatment didn't have a significant effect on cell cycle. The authors may consider toning down the conclusions on the cell cycle aspect of this beautiful study.

GUIDELINES FOR SUBMISSION OF NATURE CELL BIOLOGY ARTICLES

ARTICLE FORMAT

ABSTRACT – should not exceed 150 words and should be unreferenced. This paragraph is the most visible part of the paper and should briefly outline the background and rationale for the work, and accurately summarize the main results and conclusions. Key genes, proteins and organisms should be specified to ensure discoverability of the paper in online searches.

TEXT – the main text consists of the Introduction, Results, and Discussion sections and must not exceed 3500 words including the abstract. The Introduction should expand on the background relating to the work. The Results should be divided in subsections with subheadings, and should provide a concise and accurate description of the experimental findings. The Discussion should expand on the findings and their implications. All relevant primary literature should be cited, in particular when discussing the background and specific findings.

REFERENCES – are limited to a total of 70 in the main text and Methods combined,. They must be numbered sequentially as they appear in the main text, tables and figure legends and Methods and must follow the precise style of Nature Cell Biology references. References only cited in the Methods should be numbered consecutively following the last reference cited in the main text. References only associated with Supplementary Information (e.g. in supplementary legends) do not count toward the total reference limit and do not need to be cited in numerical continuity with references in the main text. Only published papers can be cited, and each publication cited should be included in the numbered reference list, which should include the manuscript titles. Footnotes are not permitted.

Methods should be written concisely, but should contain all elements necessary to allow interpretation

and replication of the results. As a guideline, Methods sections typically do not exceed 3,000 words. The Methods should be divided into subsections listing reagents and techniques. When citing previous methods, accurate references should be provided and any alterations should be noted. Information must be provided about: antibody dilutions, company names, catalogue numbers and clone numbers for monoclonal antibodies; sequences of RNAi and cDNA probes/primers or company names and catalogue numbers if reagents are commercial; cell line names, sources and information on cell line identity and authentication. Animal studies and experiments involving human subjects must be reported in detail, identifying the committees approving the protocols. For studies involving human subjects/samples, a statement must be included confirming that informed consent was obtained. Statistical analyses and information on the reproducibility of experimental results should be provided in a section titled "Statistics and Reproducibility".

All Nature Cell Biology manuscripts submitted on or after March 21 2016, must include a Data availability statement as a separate section after Methods but before references, under the heading "Data Availability". For Springer Nature policies on data availability see <http://www.nature.com/authors/policies/availability.html>; for more information on this particular policy see <http://www.nature.com/authors/policies/data/data-availability-statements-data-citations.pdf>. The Data availability statement should include:

- Accession codes for primary datasets (generated during the study under consideration and designated as "primary accessions") and secondary datasets (published datasets reanalysed during the study under consideration, designated as "referenced accessions"). For primary accessions data should be made public to coincide with publication of the manuscript. A list of data types for which submission to community-endorsed public repositories is mandated (including sequence, structure, microarray, deep sequencing data) can be found here <http://www.nature.com/authors/policies/availability.html#data>.
- Unique identifiers (accession codes, DOIs or other unique persistent identifier) and hyperlinks for datasets deposited in an approved repository, but for which data deposition is not mandated (see here for details <http://www.nature.com/sdata/data-policies/repositories>).
- At a minimum, please include a statement confirming that all relevant data are available from the authors, and/or are included with the manuscript (e.g. as source data or supplementary information), listing which data are included (e.g. by figure panels and data types) and mentioning any restrictions on availability.
- If a dataset has a Digital Object Identifier (DOI) as its unique identifier, we strongly encourage including this in the Reference list and citing the dataset in the Methods.

We recommend that you upload the step-by-step protocols used in this manuscript to the Protocol Exchange. More details can found at www.nature.com/protocolexchange/about.

DISPLAY ITEMS – main display items are limited to 6-8 main figures and/or main tables. For Supplementary Information see below.

FIGURES – Colour figure publication costs \$395 per colour figure. All panels of a multi-panel figure must be logically connected and arranged as they would appear in the final version. Unnecessary

figures and figure panels should be avoided (e.g. data presented in small tables could be stated briefly in the text instead).

All imaging data should be accompanied by scale bars, which should be defined in the legend. Cropped images of gels/blots are acceptable, but need to be accompanied by size markers, and to retain visible background signal within the linear range (i.e. should not be saturated). The boundaries of panels with low background have to be demarked with black lines. Splicing of panels should only be considered if unavoidable, and must be clearly marked on the figure, and noted in the legend with a statement on whether the samples were obtained and processed simultaneously. Quantitative comparisons between samples on different gels/blots are discouraged; if this is unavoidable, it has to be performed for samples derived from the same experiment with gels/blots were processed in parallel, which needs to be stated in the legend.

Regardless of format, all figures must be vector graphic compatible files, not supplied in a flattened raster/bitmap graphics format, but should be fully editable, allowing us to highlight/copy/paste all text and move individual parts of the figures (i.e. arrows, lines, x and y axes, graphs, tick marks, scale bars etc). The only parts of the figure that should be in pixel raster/bitmap format are photographic

images or 3D rendered graphics/complex technical illustrations.

Unprocessed scans of all key data generated through electrophoretic separation techniques need to be presented in a supplementary figure that should be labeled and numbered as the final supplementary figure, and should be mentioned in every relevant figure legend. This figure does not count towards the total number of figures and is the only figure that can be displayed over multiple pages, but should be provided as a single file, in PDF or TIFF format. Data in this figure can be displayed in a relatively informal style, but size markers and the figures panels corresponding to the presented data must be indicated.

The total number of Supplementary Figures (not including the “unprocessed scans” Supplementary Figure) should not exceed the number of main display items (figures and/or tables (see our Guide to Authors and March 2012 editorial <http://www.nature.com/ncb/authors/submit/index.html#suppinfo>; <http://www.nature.com/ncb/journal/v14/n3/index.html#ed>). No restrictions apply to Supplementary Tables or Videos, but we advise authors to be selective in including supplemental data.

Each Supplementary Figure should be provided as a single page and as an individual file in one of our accepted figure formats and should be presented according to our figure guidelines (see above). Supplementary Tables should be provided as individual Excel files. Supplementary Videos should be provided as .avi or .mov files up to 50 MB in size. Supplementary Figures, Tables and Videos must be

accompanied by a separate Word document including titles and legends.

GUIDELINES FOR EXPERIMENTAL AND STATISTICAL REPORTING

REPORTING REQUIREMENTS – We ask authors to complete a Reporting Summary that collects information on experimental design and reagents. We hope this will aid in your evaluation of the paper. The Reporting Summary can be found here <https://www.nature.com/documents/nr-reporting-summary.pdf>) Please note that these forms are dynamic 'smart pdfs' and must therefore be downloaded and completed in Adobe Reader. We will then flatten them for ease of use. If you would like to reference the guidance text as you complete the template, please access these flattened versions at <http://www.nature.com/authors/policies/availability.html>.

Author Rebuttal, first revision:

Point-by-point response to reviewer comments:

Reviewer #1:

Remarks to the Author:

I sincerely thank the authors for addressing my comments. To facilitate review at this point, I will not go over all the details, but rather focus on the main claims (as indicated in the abstract) and I will comment on whether or not I believe there is enough support for them after these initial revisions, and why.

Overall, I think the authors have addressed a lot of the questions convincingly, while some other claims remain, in my opinion, not sufficiently supported by data.

We thank the reviewer for the overall positive assessment and acknowledging our efforts to address the reviewer's concerns. We have now addressed the few remaining points by additional experimentation as well as careful editing of the text.

(1) The authors claim in the abstract that “we identify a key role for coordinated mechanical forces stemming from contractile, proliferative and proteolytic activities across the epithelial and mesenchymal compartments in generating the placode structure”

I somehow agree with this statement. The authors now provide more convincing data of the mechanical aspects of placode morphogenesis. In particular, the new data provide more evidence that there is indeed ‘ring of fibroblasts’ that is under tension. The new laser ablation experiments indicate that there is indeed tension in the ring of fibroblasts. However, no control of this experiment (by either cutting in the perpendicular direction or in the mesenchyme far away from the ring) is shown. So, I agree there is tension in the ring, but it is unclear still if there is more tension in the ring than in the surrounding tissue, as would be expected in the model presented by the authors. The authors should check that this is indeed the case by doing laser ablation in the mesenchyme away from the ring and confirming less tension far away.

Beyond this, the authors provided new evidence that there is more tension in the epidermis than in the placode. In my opinion, these data now justify the statements regarding the elongation of the placode.

We appreciate that the reviewer acknowledges that the data on the fibroblast ring is now convincing. While comparing this ring to surrounding, dispersed fibroblasts is a complex comparison, we have now carried out this additional ablation experiment: radial ablations in the ring of fibroblasts compared with ablations further away in the dermis. These new data show that the ring of fibroblasts surrounding the placode displays higher magnitudes of tension than the surrounding fibroblasts (new Supplementary Figure 2h, i), further validating our previous conclusions.

(2) The authors claim in the abstract that “these mechanical stresses further enhance and compartmentalize Sox9 expression to promote stem cell positioning”

I thank the authors for their efforts to strengthen their claims on this respect. I agree with the authors that confinement seems to be a factor in Sox9 expression, and that confinement seems correlated with the extrinsic forces from the mesenchymal cells ring. However, it is unclear to me if it is the confinement of cells caused by the extrinsic forces, or the actual forces themselves that direct the pattern of Sox9 expression. The data provided only shows correlations in this respect, so I am not sure the authors can distinguish between Sox9 expression increasing because of mechanical constriction or because of the cells being more spatially confined as a result of morphogenetic changes. Mechanical induction of Sox9 was previously reported by the Gros lab (Parada et al., Dev Cell, 2021). Gros et al showed that internal mechanics in the tissue drives Sox9 specification during digit formation in mouse. Therefore, the authors may well be correct and mechanics may cause Sox9 expression in the placode too, but at this point I only have correlations.

The authors mention in their reply that they cannot assess the role of mechanics on Sox9 directly because there is no Sox9-GFP reporter mouse. This is not accurate, see for instance Parada et al., Dev Cell, 2021. Moreover, laser ablation can be repeated cyclically (periodic ablation) to prevent the tension recovery after ablation. I feel these experiments are not very complicated, but I understand that there are other indirect ways to address this question, albeit likely less conclusive.

We are happy to see that the reviewer agrees with our conclusion that confinement is a factor regulating Sox9 expression. We also are grateful for pointing out the important reference from the Gros lab that we have accidentally omitted and which reports a similar mechanical effect on Sox9 in a different system. We have now included this important reference (p.8).

Regarding the suggested repeated laser ablation experiments to monitor Sox9 expression in an intact embryo, we still maintain that such experiments are not feasible for several reasons.

1- As we emphasized in the previous point-by-point response, to our knowledge there is no SOX9-GFP fusion protein mouse line that would allow following levels of Sox9 protein levels in real time in living animals. The mouse model referred to by the reviewer and used in the Parada et al. study (which we also have in the laboratory) is a transcriptional reporter not reporting Sox9 protein expression but Sox9 transcription, where Sox9 and EGFP are separated by an IRES. Consequently, once the protein is translated, EGFP is fully independent of Sox9 and following decay of the fluorescence means following the lifetime of EGFP, not Sox9. Hence this mouse line is not suitable for the experiments suggested by the reviewer where release of tension is expected to result in decrease in Sox9 expression, which would be read out as decreased EGFP.

2- It should be emphasized that we perform laser ablation on an entire late stage mouse embryo, freshly extracted from the mother. This makes it possible, for the first time, to measure tissue tension at an advanced embryonic stage in a completely intact animal. Unlike cell culture systems, this is an extremely challenging method where maintaining tissue viability is challenging and of utmost importance to be able to draw conclusions of tissue dynamics. Given that laser ablation will generate tissue damage, we do not

see how we could be confident about protein expression level changes post inducing this damage, let alone monitor changes post multiple ablations.

Thus, while we appreciate this suggestion, we do not see how this experiment would be feasible.

The new experiments where isolated epidermal progenitor cells are placed into microwells are a great idea. However, I fail to see in the images any substantial increase in Sox9 at the boundary (Fig. 4I), as the authors claim. The experiments where epidermal progenitors are cultured in hydrogels are very nice too, and here there seems to be an increase of Sox9. However, this is in vitro and extrapolation in vivo is complex. I would like to point out that it is well known that ectopic compression drives Sox9 expression in mouse limb bud chondrogenic progenitors – see e.g., Takahashi et al., J Cell Sci, 1998.

I think the authors provide enough evidence that contractile extrinsic forces somehow, directly or indirectly, affect Sox9 expression. The authors should clarify that confinement does not induce Sox9 expression (as it is observed when myosin is inhibited), but rather enhances it upon confinement. It is also important that the previous literature on mechanical induction of Sox9 (the papers mentioned above and several others) is properly cited, and also the results reported here discussed in the light of previous works.

We thank the reviewer for the positive comment on the PDMS microwell experiment. The fluorescence intensity of Sox9 has been carefully quantified across multiple independent experiments that consistently show a strictly localized increase in Sox9 expression at the boundary between the PDMS well and the monolayer. Although we agree with the reviewer that extrapolation between in vitro and in vivo is complex, we would like to emphasize that we now show the effect of confinement both in vivo using Myh9-dermalKO), ex vivo (skin explant cultures with Blebbistatin to inhibit myosin2A) and in vitro using 2 different approaches. These orthogonal approaches that all point to the role of confinement in promoting Sox9, together with the previous literature that the reviewer also refers to, showing similar effects in very different systems make us very confident about these results.

We thank the reviewer for helping us to strengthen this conclusion and for pointing out the two prior studies that we have now cite in the text (p 10).

(3) The authors claim in the abstract that “proteolytic remodeling locally softens the basement membrane”

The authors provided some evidence that this is the case. The AFM experiments are not fully convincing because they are done on sections in which the mechanics of the basement membrane could be altered (both in cryosections and in freshly cut sections; the sectioning is the issue here). Also, it is unclear to me how these measurements isolate the basement membrane stiffness from the mechanical properties of the cells surrounding the basement membrane in the cryosectioned tissue, since AFM probes both

simultaneously. Moreover, previous works have shown marked changes of basement membrane components at the site of budding that have been linked to changes in basement membrane stiffness (see eg paper by Ken Yamada on lung budding), so it is a bit surprising not to see changes in basement membrane composition at the invaginating site. However, both the single cell sequencing data and MMP inhibition experiments are convincing, and seem to indicate that basement membrane mechanics is likely involved in the invagination. As I said in my previous review, the MMP inhibition experiments are great and they clearly indicate that the basement membrane plays an important role. If this role is mechanical or not, I am not sure, but it is possible.

Initially, we were also surprised not to observe significant differences in BM composition in our immunofluorescence analysis of tissue sections of the three main components of the basement membrane (Collagen IV, Laminin-322 and Lamin-511). We therefore hypothesized that the softening of the basement membrane was due to active remodeling-associated turnover rather than specific alterations in molecular composition. This hypothesis was confirmed by analysis of the single cell sequencing data and by the MMP inhibition experiments.

We might also have not been sufficiently clear about how we benchmarked the AFM measurements in our previous reply. The comparison was between full, unsectioned, freshly isolated mouse tissue that had been decellularized to be able to directly access the basement membrane with the cantilever and freshly cut cryosections. Thus, we excluded exactly the aspect that the reviewer is concerned about – the impact of sectioning. The freshly isolated, unsectioned tissue gave comparable values than the freshly cut cryosection. Therefore we are confident about this methodology which is also widely used in the field.

Finally, regarding the simulations, the authors have not convinced me of their relevance in this work. I completely agree with the authors that these simulations are state-of-the-art at a technical level. We all agree that 3D vertex models are technically very challenging. Moreover, I agree that combining multiple tissues and matrix in the simulation is new. My issue is not with the technical aspects of the model, but rather the fact that there are many experimental aspects that are unknown and could affect the model. In other words, there are many assumptions when one develops a vertex model and we do not know if these assumptions are correct in this system. As an example, the authors did not include proliferation or dynamics of forces, etc. All these aspects could change the predictions of the model. Moreover, I strongly disagree with the authors comment that vertex models are ‘extraordinarily well-vetted’. Vertex models have rarely been tested experimentally with direct mechanical measurements even in 2D tissues. Instead, researchers have used the models to fit the data and interpret the experimental observations from the viewpoint of the model. If the model is correct, that leads to correct interpretations, but if the assumptions are incorrect (because, for instance, cell proliferation or some other factor was neglected), the interpretation is flawed. In my opinion, the authors do not attempt to carefully test experimentally the model, but rather they interpret the experimental observations from the model assuming it is correct. In the best-case scenario, the model is correct and the data properly interpreted. In the worst-case scenario, the model is incorrect but roughly fits the data in some cases, leading to the belief that the interpretation of the data is correct.

As an example, the new data shows that inhibition of Myosin IIA in the epidermis leads to evagination (instead of invagination) of the placode. The authors claim this is the case in the simulations (Supplementary Fig. 4g), but I do not see any evagination in the simulations results that comes even close to the data in Supplementary Fig. 4h.

We thank the reviewer for this more detailed explanation of concerns about the modeling. First, we will address the example of evagination. As highlighted in the caption, the images shown in Supplementary Fig. 4h are at E16.5. This is later time point in development than both the modeling and experimental data shown in the main text, which focuses on E14.5 and E15.5, and so the degree of invagination is larger at E16.5, as expected. The reason we showed the E16.5 data is because we were focused here on the **direction** of the invagination (upward or downward), and this is easier to see at E16.5.

Beyond this particular example, it is important to emphasize what we (and we believe most of the community) realistically expect the modeling to be able to do. Let us first address the comment “*Vertex models have rarely been tested experimentally with direct mechanical measurements even in 2D tissues. Instead, researchers have used the models to fit the data and interpret the experimental observations from the viewpoint of the model.*” We disagree with this characterization. Although we fully agree with the reviewer that direct mechanical measurements at the cellular scale are difficult with existing technology (though several groups are working on this), the existing modeling results are not post-observation fits to data. For example, in Wang PNAS 2020 the vertex model makes a quantitative, a priori prediction with **no fit parameters** (black line in Fig 4D, reproduced below), of where in observed cell shape space (parameterized by a perimeter to area ratio called the shape index and a cell alignment parameter) the tissue should be fluid-like with regular cell rearrangements vs. solid like with rare rearrangements. The data from fruit fly germband extension is overlaid, demonstrating the excellent quantitative predictive power of the model, and validating the underlying physical model equations.

We agree do agree that vertex model equations have a significant number of parameters, some of which (like cell shape) are well-constrained by data, and a few

parameters (like the heterotypic interfacial tensions) that are not well constrained because the tools to measure them in a non-invasive way are still being developed. The power, then, of vertex models is being able to vary those parameters within experimental uncertainty, and explore the entire range of possible morphology changes given that uncertainty. That is our approach in Fig 3; The blue and red regions in Fig 3 e correspond to a range of possible parameters for interfacial tensions that are not well-constrained by our experimental data. There are, of course, some constraints on those parameters, as values outside the range we report lead to unphysical cell shapes (e.g. unphysically elongated cells) or unphysical dynamics (e.g. basal cell regularly delaminating even in the presence of very small fluctuations). In addition, there are other parameters that are not well constrained (e.g. magnitude of fluctuations driven by active cell contractility) but which we have varied significantly in simulations and checked that the parameter has no impact on the results reported here.

Taken together, the reviewer is correct in that there are model parameters that are not well constrained, but we have either validated that they do not have an impact or we have transparently reported the entire range of possibilities that are consistent with observations. Thus, we are confident that the model is doing exactly what we are asking it to do: under the assumption that the underlying physical equation is correct (and that **has** been validated with no fit parameters in multiple previous experiments) we are asking it to predict the space of possible outcomes given the **parameter** uncertainty, which it does well. This allows us to draw reasonable conclusions about which mechanisms can be acting at which timepoints.

One final important point where we agree with the referee is that the underlying physical equations do not include additional processes like cell division or highly persistent force generation (e.g. beyond fluctuations), and that those could be contributing to features of our data. In fact, we do think it is likely that the large invaginations seen in E16.5 are driven by cell divisions. We choose to not attempt to include them here because it is not necessary to recapitulate the placode morphology we observe in the earlier stages (E14.5 to E15.5), and including it in the model is both technically very challenging and introduces a whole host of under constrained additional parameters. Therefore, we have now edited the manuscript to emphasize the caveat that the model is still lacking some potentially important features, and including cell divisions and force dynamics is the next (albeit technically very challenging) step. We would like to stress that while we use the model to predict experiments, in the end the experimental data stands alone and does not rely on the model for interpretation.

Reviewer #2:

Remarks to the Author:

I believe the authors have sufficiently addressed the concerns of the reviewers. The manuscript is well written and presented. The findings are interesting and advance our understanding of hair follicle formation and the cellular forces involved.

We thank the reviewer for the very positive assessment of our study.

Reviewer #3:

Remarks to the Author:

In this study, Villeneuve et al. aims to characterize the mechanical forces and morphogenetic behaviors that placode cells encounter during embryonic hair follicle development. Specifically, the authors used biophysical approaches to look at placode stage (E14.5) and hair germ stage (E15.5) and mapped out a two-step process of cellular elongation followed by downgrowth of the budding placode. They concluded that both cell-intrinsic contractility and extrinsic forces from the dermis contributed to the morphological changes in the placode. With a combination of genetic experiments and drug treatment, they disrupted the contractility in the epidermal and dermal compartment and showed both resulted in defects in placode formation and/or invagination. Mechanistically, they linked placode invagination at E15.5 with decreased extrinsic mechanical stress and resultant cell cycle reentry experienced by placode cells. They further attributed the release of mechanical force to the softening of the basement membrane during placode invagination.

Overall, the authors addressed my comments sufficiently. They used ablation experiments and displacement measurement to ablate the fibroblast ring and show the contractile force exerted on the placode by the fibroblast ring, indicated by the recoil of placode after the ablation (Figure 2i).

We thank the reviewer for the positive assessment on our study as well as for the overall very positive feedback regarding our efforts in answering the reviewer concerns.

Although beyond the scope of this work, it would be nice to see the effect of ablating the DC specifically as well. Placode tension was also measured (Figure 2d-g) to address the alternative model wherein the placode pushes fibroblasts downward. Additionally, Sox2+ DC's were shown to form normally in the new Figure 5h, suggesting the failure in placode invagination isn't an indirect effect from loss of DC. However, it is notable that the Sox2+ dermal cells do not appear to be clustered in Figure 5j, suggesting that blebbistatin affected the DC, which is not only defined by Sox2 expression. The confinement assay (Figure 4l) was done to show Sox9 expression is highest at the invagination interface where there was highest tension.

We fully agree with the reviewer that it will be interesting to measure the tension within the DC and to understand the mechanisms of DC formation, which remains unknown at present. These are indeed ongoing long-term projects in the lab and are currently addressing these exciting questions in separate follow-up studies.

One remaining issue is related to the mechanistic link to cell cycle. For this study, the cell cycle aspect does not appear to add much, because the link between cell cycle regulation and mechanical forces is somewhat weak and disconnected and the data are not convincing (Figure 5a-d). The mitomycin-C treatment (Figure 5e-f) also does not necessarily establish that the defect of placode formation upon cell cycle inhibition is downstream of mechanical forces, which the authors seem to suggest in the manuscript.

Also, the authors stated that Blebbistatin treatment didn't have a significant effect on cell cycle. The authors may consider toning down the conclusions on the cell cycle aspect of this beautiful study.

We thank the reviewer for the critical comments and for describing our study as “beautiful”.

Regarding the role cell division and mechanics, we have shown the following:

1. Laser ablation experiments show that placode cells are confined by the contractile fibroblast ring.
2. Quantitative immunofluorescence shows activation of the mechanosensitive transcription factor YAP as well as upregulation of Ki67 as a marker for proliferation at E15.5, coinciding with the timepoint when basement membrane starts to degrade (allowing release of pressure).
3. Live imaging confirms enhanced cell divisions at the base of the placode at E15.5, coinciding with downward flow.
4. Inhibition of YAP reduces keratinocyte proliferation.
5. Inhibition of cell divisions attenuate placode downward flow.

Collectively these experiments demonstrate a link between confinement, the activation of the mechanosensitive transcription factor YAP, cell divisions and downward growth. Having said this we fully agree with the reviewer that it is important not to overstate conclusions. To avoid this, we now write:

“Collectively, these data indicated that confinement of the placode at E14.5 promotes nuclear exclusion of YAP, contributing to reducing cell divisions, while the subsequent folding/buckling of the epithelium at E15.5 releases compressive stress on the placode, facilitating YAP activation, cell cycle re-entry and mitoses, which then enhance efficient budding of the placode.”

Decision Letter, second revision:

Our ref: NCB-A50186B

26th October 2023

Dear Dr. Wickström, dear Sara,

Thank you for submitting your revised manuscript "Mechanical forces across compartments coordinate cell shape and fate transitions to generate tissue architecture" (NCB-A50186B). It has now been seen by the original referees and their comments are below. The reviewers find that the paper has improved in revision, and therefore we'll be happy in principle to publish it in Nature Cell Biology, pending minor revisions to satisfy the referees' final requests and to comply with our editorial and formatting guidelines.

Thank you again for your interest in Nature Cell Biology Please do not hesitate to contact me if you have any questions.

Sincerely,

Daryl

Daryl Jason Verzosa David, PhD

Senior Editor, Nature Cell Biology
Nature Portfolio

Heidelberger Platz 3, 14197 Berlin, Germany
Email: daryl.david@nature.com
ORCID: <https://orcid.org/0000-0002-9253-4805>

Reviewer #1 (Remarks to the Author):

The authors have now addressed all my concerns regarding the experimental part of the work. I think the conclusions are well supported by the data and the work represents an important advance in cell and developmental biology.

Decision Letter, final checks:

Our ref: NCB-A50186B

14th November 2023

Dear Dr. Wickström,

Thank you for your patience as we've prepared the guidelines for final submission of your Nature Cell Biology manuscript, "Mechanical forces across compartments coordinate cell shape and fate transitions to generate tissue architecture" (NCB-A50186B). Please carefully follow the step-by-step instructions provided in the attached file, and add a response in each row of the table to indicate the changes that you have made. Please also check and comment on any additional marked-up edits we have proposed within the text. Ensuring that each point is addressed will help to ensure that your revised manuscript can be swiftly handed over to our production team.

In recognition of the time and expertise our reviewers provide to Nature Cell Biology's editorial process, we would like to formally acknowledge their contribution to the external peer review of your manuscript entitled "Mechanical forces across compartments coordinate cell shape and fate transitions to generate tissue architecture". For those reviewers who give their assent, we will be publishing their names alongside the published article.

Nature Cell Biology offers a Transparent Peer Review option for new original research manuscripts submitted after December 1st, 2019. As part of this initiative, we encourage our authors to support increased transparency into the peer review process by agreeing to have the reviewer comments, author rebuttal letters, and editorial decision letters published as a Supplementary item. When you submit your final files please clearly state in your cover letter whether or not you would like to participate in this initiative. Please note that failure to state your preference will result in delays in accepting your manuscript for publication.

Cover suggestions

COVER ARTWORK: We welcome submissions of artwork for consideration for our cover. For more information, please see our guide for cover artwork.

Nature Cell Biology has now transitioned to a unified Rights Collection system which will allow our Author Services team to quickly and easily collect the rights and permissions required to publish your work. Approximately 10 days after your paper is formally accepted, you will receive an email in providing you with a link to complete the grant of rights. If your paper is eligible for Open Access, our Author Services team will also be in touch regarding any additional information that may be required to arrange payment for your article.

Please note that *Nature Cell Biology* is a Transformative Journal (TJ). Authors may publish their research with us through the traditional subscription access route or make their paper immediately open access through payment of an article-processing charge (APC). Authors will not be required to make a final decision about access to their article until it has been accepted. Find out more about Transformative Journals

Please use the following link for uploading these materials:
[Redacted]

Best regards,

Kendra Donahue
Staff
Nature Cell Biology

On behalf of

Daryl Jason Verzosa David, PhD

Senior Editor, Nature Cell Biology
Nature Portfolio

Heidelberger Platz 3, 14197 Berlin, Germany

Email: daryl.david@nature.com
ORCID: <https://orcid.org/0000-0002-9253-4805>

Reviewer #1:

Remarks to the Author:

The authors have now addressed all my concerns regarding the experimental part of the work. I think the conclusions are well supported by the data and the work represents an important advance in cell and developmental biology.

Final Decision Letter:

Dear Dr Wickström,

I am pleased to inform you that your manuscript, "Mechanical forces across compartments coordinate cell shape and fate transitions to generate tissue architecture", has now been accepted for publication in Nature Cell Biology.

Please note that *Nature Cell Biology* is a Transformative Journal (TJ). Authors may publish their research with us through the traditional subscription access route or make their paper immediately open access through payment of an article-processing charge (APC). Authors will not be required to make a final decision about access to their article until it has been accepted. Find out more about Transformative Journals

If you have not already done so, we strongly recommend that you upload the step-by-step protocols used in this manuscript to the Protocol Exchange (www.nature.com/protocolexchange), an open online resource established by Nature Protocols that allows researchers to share their detailed experimental know-how. All uploaded protocols are made freely available, assigned DOIs for ease of citation and are fully searchable through nature.com. Protocols and Nature Portfolio journal papers in which they are used can be linked to one another, and this link is clearly and prominently visible in the online versions of both papers. Authors who performed the specific experiments can act as primary authors for the Protocol as they will be best placed to share the methodology details, but the Corresponding Author of the present research paper should be included as one of the authors. By uploading your Protocols to Protocol Exchange, you are enabling researchers to more readily reproduce or adapt the methodology you use, as well as increasing the visibility of your protocols and papers. You can also establish a dedicated page to collect your lab Protocols. Further information can be found at www.nature.com/protocolexchange/about

You can use a single sign-on for all your accounts, view the status of all your manuscript submissions and reviews, access usage statistics for your published articles and download a record of your

refereeing activity for the Nature Portfolio.

With kind regards,

Daryl

Daryl Jason Verzosa David, PhD

Senior Editor, Nature Cell Biology
Nature Portfolio

Heidelberger Platz 3, 14197 Berlin, Germany
Email: daryl.david@nature.com
ORCID: <https://orcid.org/0000-0002-9253-4805>
